# Global connections between El Nino and landslide impacts

Robert Emberson [1,2,3 ✉], Dalia Kirschbaum[1] & Thomas Stanley [1,2,3]

El Nino is a critical part of global inter-annual climate variability, and the intensity of El Nino has major implications for rainfall-induced natural hazards in many vulnerable countries. The impact of landslides triggered by rainfall is likely to be modulated by the strength of El Nino, but the nature of this connection and the places where it is most relevant remains unconstrained. Here we combine new satellite rainfall data with a global landslide exposure model to show that El Nino has far-reaching effects on landslide impacts to people and infrastructure. We find that the impact of El Nino on landslide exposure can be greater in parts of Southeast Asia and Latin America than that due to seasonal rainfall variability. These findings improve our understanding of hazard variability around the world and can assist disaster mitigation efforts on seasonal timescales.

[1] Hydrological Sciences Laboratory, NASA Goddard Space Flight Center, 8800 Greenbelt Road, Greenbelt, MD, USA. [2] Universities Space Research Association, Columbia, MD, USA. [3] Goddard Earth Sciences Technology and Research, Columbia, MD, USA. ✉email: robert.a.emberson@nasa.gov

Landslides and associated debris flows result in devastating impacts around the world every year, causing thousands of fatalities[1,2]. Extreme, intense rainfall is a common trigger for landslide hazards, and it is therefore crucial to understand the changes in rainfall patterns that may drive the damage from these hazards. Intense rainfall occurring during tropical storms or localised convective thunderstorms can result in widespread landsliding[3,4], the incidence of which can vary on seasonal to multi-annual timescales[5]. Existing work has highlighted that during the Northern hemisphere summer, the average landslide fatalities increase[2,6], particularly in association with the peak of the Indian Summer Monsoon and tropical cyclone seasons. However, the degree to which landslide impacts are driven by rainfall seasonality in many countries remains unconstrained. The effect of multi-year changes in rainfall patterns—crucially the El Nino Southern Oscillation (ENSO)—on landslide impacts at a global extent is largely unknown. This is especially relevant since ENSO drives significant changes in rainfall intensity in a range of countries where landslides cause fatalities, including India, Indonesia, Colombia, and the Philippines[7,8]. Interdecadal variability of extreme rainfall events in South America is also influenced by the Interdecadal Pacific Oscillation[9]. ENSO variability brings major and diverse changes to seasonal weather patterns in many countries around the world due to teleconnections with the equatorial Pacific Ocean. The changes in rainfall patterns in particular can lead to highly variable agricultural outcomes[10], increases in the incidence of flooding[11,12], and increased occurrence of fires[13,14], among other impacts.

Variability in the ocean-atmosphere system occurs over several different temporal scales. The fluctuations in sea surface temperatures and related changes in terrestrial rainfall and temperature are highly complex, but we make the simplifying assumption that the associated changes relevant to landslides are solely due to changes in rainfall. To illustrate the impact of ENSO on rainfall patterns, in Fig. 1 we show the difference between the peak El Nino and La Nina conditions for NASA's IMERG (Integrated Multi-satellitE Retrievals for GPM[15]) satellite rainfall record, which stretches from June 2000-present. We show difference maps of the rainfall patterns during peak El Nino and peak La Nina conditions over this period, for both Northern Hemisphere summer and winter. ENSO observations are derived from the Multivariate ENSO Index version 2 (MVEI v2)[16]. ENSO changes rainfall patterns chiefly around the Pacific, but also further afield (Fig. 1). The sign of the ENSO index changes over timescales on the order of 2-5 years[17]. Much of the change in rainfall occurs over the ocean, but major changes are also observed in Latin America, the Caribbean, SE Asia, and India. Many countries in these areas have high numbers of landslide-related fatalities[2,18]. Prior global studies have indicated that ENSO has significant impacts on global patterns of total rainfall, as well as the incidence of extreme rainfall[19,20].

It is important to note that the ENSO system exhibits significant variability, with some studies showing that in addition to the conventional 'Cold-Tongue' El Nino, an atypical form may exist, variously referred to as 'Warm-Pool' or 'Central Pacific' El Nino events[21]. While these two types of event are both well characterised by the MVEI product[16], they may lead to differences in precipitation variability[21]. In this study, we do not differentiate between different forms of El Nino event, partly because there are only a handful of ENSO cycles in the period of observation, limiting the availability of data. In addition, since the existence of this non-standard El Nino form remains debated[22] and no widely accepted index for its strength exists, we suggest that our initial findings can lay the foundations for future work to explore the differences in landslide impact due to El Nino variability when more data is available. The North Atlantic

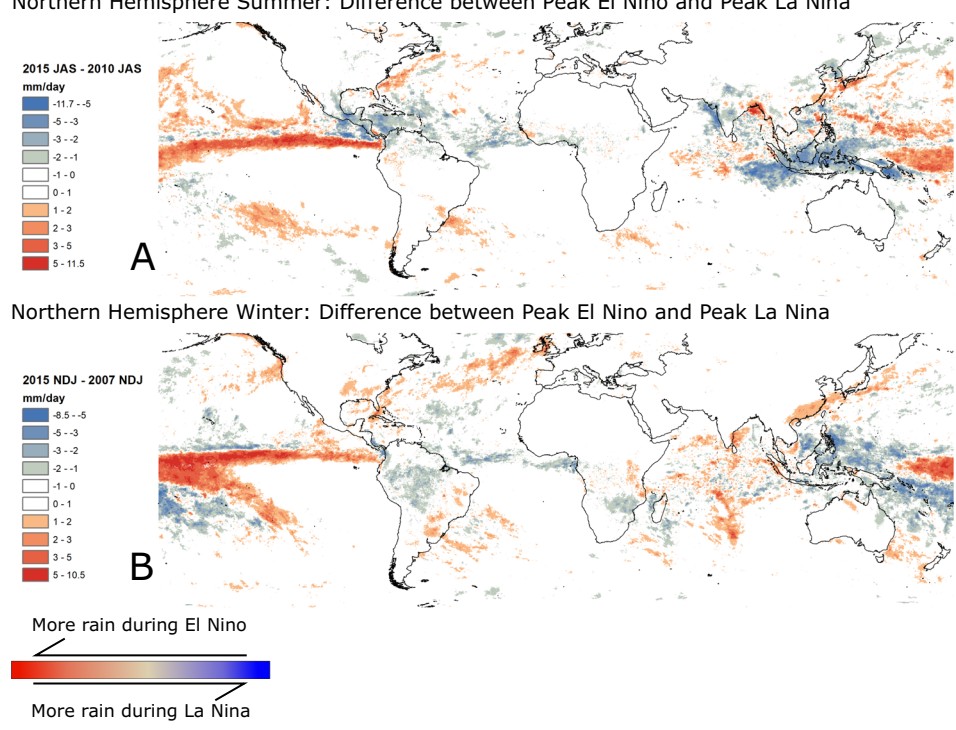

**Fig. 1 Difference between rainfall during El Nino and La Nina conditions, in Northern Hemisphere summer and winter.** Rainfall values calculated from the IMERG (Integrated Multi-satellitE Retrievals for GPM) v06B rainfall product. Values shown represent the difference in average daily rainfall (mm/day) for July–August–September (**A**) and November–December–January (**B**). The differences are calculated for the strongest El Nino and La Nina conditions on record during the IMERG v06B record (2001-present)—peak summer El Nino is 2015, and winter is 2015–2016. Peak summer La Nina is 2010, and winter is 2007. The pair of figures together illustrate the main regional differences in rainfall patterns resulting from the El Nino Southern Oscillation (ENSO).

Oscillation and Pacific Decadal Oscillation are decadal-scale changes in the ocean-atmosphere system, with less significant effects on rainfall in landslide-prone regions. During the positive phase of the Pacific Decadal Oscillation, rainfall increases over the Alaskan coastal range as well as the South-Western US and Mexico, while decreases are observed in Canada, East Siberia, Australia and during the South Asian Summer Monsoon[23], with the inverse observed during the negative phase. PDO cycles are much longer than those of ENSO, on the order of 20-30 years[24]. PDO can also change the frequency of extreme rainfall events in South America[9]. Increases in the strength of the North Atlantic Oscillation leads to rainfall increases over northern Europe, the eastern USA, and parts of Scandinavia, and decreases in the northwest Atlantic and Mediterranean regions. NAO cycles are also on the order of decades or more[25].

Connecting global climate patterns with the highly localised impacts of landslide hazards has remained challenging, in part due to limited inventories of landslide events, particularly in developing countries. This precludes consistent modelling of seasonal landslide patterns around the world. Since empirical data can be difficult to collect, model-based estimates of rainfall-triggered landsliding can help fill in gaps[26]. Now that close to two decades of consistent global satellite rainfall data at moderate resolution are available, we can explore at a local level where landslide exposure is most strongly modulated by the extreme rainfall caused by multi-year climate oscillations.

In this study, we exploit this recently available data, pairing an updated dataset of global exposure to landslide hazard with empirical observations of several ocean-atmosphere teleconnections, including ENSO, the North Atlantic Oscillation (NAO), and the Pacific Decadal Oscillation (PDO). We find that in many countries there are significant shifts in the exposure of people and infrastructure to rainfall-triggered landslide hazards depending on the current state of ENSO. Connecting climatological changes at the continental scale with the exposure of people and infrastructure to landslides represents a more complete picture of landslide hazard and exposure. Although we observe some changes in exposure with respect to PDO and NAO, their cycles have long timescales that are not captured by the 18-year rainfall record, meaning no conclusions can be drawn as to the effect of PDO and NAO on landslide exposure. We include these results in the supplementary information. Our model estimates provide for the first time indications of connections between the strength of the ENSO system and the exposure of people and infrastructure to landslide hazards.

## Results

As demonstrated by other studies, the seasonal variability of landslides is primarily set by the annual variability of dominant rainfall regimes such as the monsoon regimes and the tropical cyclone seasons around the world[2,6]. For each of the 38,257 admin-2 districts, we calculate the seasonality of exposure to landslides for population, roads and critical infrastructure (see Fig. 2 for examples). In order to test how strongly ENSO (or PDO or NAO) controls exposure, we first remove the annual variability. For each month in the data, we calculate a 12-month moving average of exposure with a window starting 12 months prior to the month in question. The moving window starts 12 months prior to the month in question, and includes that month. We also calculate the 12-month moving average of ENSO for that same month. We can then plot the smoothed ENSO index against the smoothed exposure data. The exposure output matches the temporal frequency of the MVEI data, but since it depends on daily rainfall data it captures the short duration, intense rainfall events most likely to trigger landslides.

Comparing these two moving averages, we can assess both the strength of the correlation, as well as the slope of the fit to assess the importance of ENSO in modulating landslide exposure. We suggest that a strong correlation indicates that ENSO is an important control on exposure to landslides, and a steeper slope of the fit indicates a greater overall shift in exposure with each incremental change in ENSO. Figure 2 illustrates several examples of this.

In the locations illustrated in Fig. 2, it is clear that ENSO can have a significant effect on the expected exposure of people to landslide hazard. The next step is to map this influence globally, and illustrate the regions where there is a significant ENSO driven effect, as well as the magnitude of that effect. We have calculated the p-value of the relationship between 12-month smoothed MVEI values and the modelled exposure to assess the significance of the relationship. In Fig. 3, we show the p-value for the relationship between total monthly rainfall (Fig. 3A) and exposure (Fig. 3B) and MVEI. In both cases, this is shown for admin-2 level regions where the p-value is below a 0.05 threshold. We find that there are many regions with extremely low p-values, indicating a significant relationship between MVEI and the total rainfall, as well as the modelled exposure.

We have also calculated the p-value for the relationship between MVEI and total monthly rainfall, number of days with extreme rainfall (>95th percentile) per month, and number of hazard 'nowcasts' issued by the LHASA model per month. This allows us to assess where MVEI is most significantly associated with extreme rainfall, hazard, and exposure, to highlight where each of these model components are most important. Figures illustrating the global distribution of p-values for these model outputs are shown in the supplementary information. We find that total rainfall and extreme rainfall are significantly associated with MVEI values in many parts of the world, with the most significant relationships in Northern South America, Southern Brazil and Uruguay, Central Asia, Northern Australia and South East Asia (Fig. 3A and Supplementary Fig. 1, respectively). The areas where landslide hazard is significantly associated with MVEI are similar, although nowcasts and exposure have smaller p-values in mountainous regions (Supplementary Figs. 3 and 4).

A significant relationship between MVEI and the model outputs does not necessarily indicate that MVEI creates large changes. Simply put, even small changes can have significant relationships with MVEI if other climatological trends do not occlude the relationship. However, we are interested in the areas where ENSO causes the largest changes in exposure. As such, we must look at the slope of the relationship between MVEI and exposure—see, for example, the middle set of figures in Fig. 2, where a unit increase in MVEI results in a reduction of 100,000 person days per month.

In Fig. 4, we show the magnitude of the change driven by a unit shift in MVEI index. We calculate this based on linear least-squared regression. We choose to fit a linear relationship to all of the districts to ensure consistency, and because it seems to describe the relationship between MVEI and model outputs in areas where ENSO is known to be important (e.g., Fig. 2). Figure 4 therefore shows the areas where we model a statistically significant impact of MVEI that leads to major changes in the fraction of population exposed to landslides. By looking at the fraction of the population exposed, we can consistently compare areas with varying population density.

Since there are only a handful of isolated areas where ENSO has an effect that is both significant and of a large magnitude, in Fig. 4 we show two of the key locations—SE Asia, and the northern part of Latin American.

Using these two key regions to illustrate our findings, we can see that as the ENSO shifts to an El Nino state, it is likely to

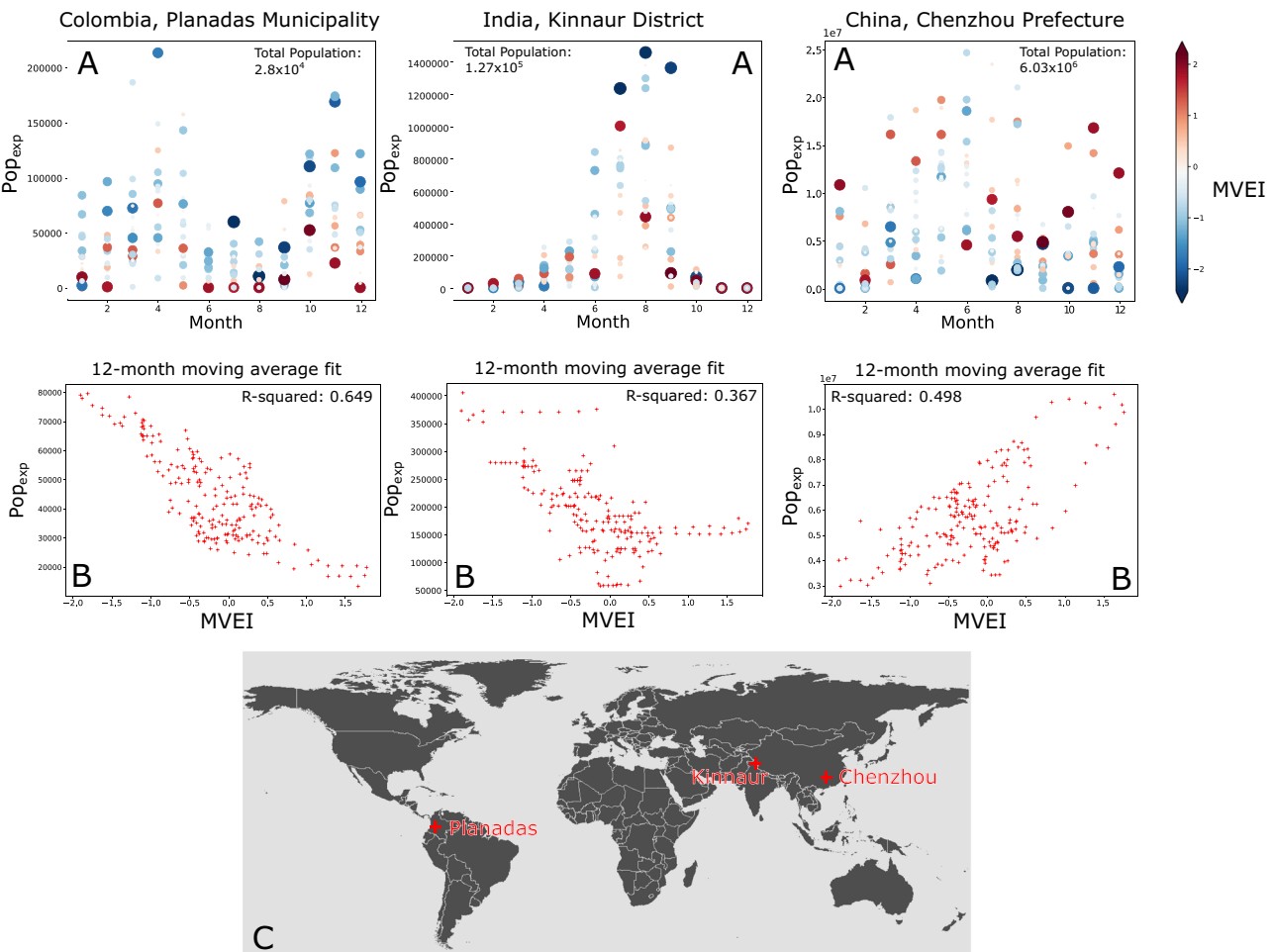

**Fig. 2 Landslide exposure for 3 illustrative administrative districts for the period 2001-2019.** The series of plots labelled A show the sum total of $Pop_{exp}$ for each month in the administrative district. This is the total number of people exposed to elevated landslide hazard in that district for each month to show seasonality. The blue-to-red colour scheme shows the corresponding value for the MultiVariate El Nino Southern Oscillation (ENSO) Index version 2 (MVEI), with red colours indicating stronger El Nino, and blue showing La Nina conditions. The size of the point also corresponds to the intensity of either El Nino or La Nina conditions, to highlight the major differences between the two end-members. The series of plots labelled B show the relationship between the 12-month moving average of MVEI and landslide exposure, which removes the annual precipitation cycle. Each of the three regions illustrate differences in the strength of seasonal impacts, as well as the sign of the relationship between MVEI and $Pop_{exp}$. The map labelled C shows the location of each of the administrative districts. It is important to note the differences in seasonal variability: Kinnaur has a singular rainy season peak, and almost no other landslide activity; Planadas has a spring and autumn peak, while Chenzhou has much more limited seasonal variability, suggesting ENSO is the primary control.

reduce the exposure to landslides in Colombia, the Philippines, and Indonesia to a major extent. In the Planadas district of Colombia (Fig. 2), for example, the change in exposure can be nearly as large as the seasonal variations in exposure (by comparing the top and bottom figures for Planadas in Fig. 2, we can see that ENSO-driven changes are on the order of 60,000 change in $Pop_{exp}$, with seasonal variability on the order of 100,000 change in $Pop_{exp}$). The Philippines, Indonesia and Colombia are countries where rainfall-triggered landslides cause many fatalities[18,27]. In Fig. 5, we show the relative change in exposure due to a unit change in MVEI relative to the monthly average exposure; the areas where ENSO leads to large magnitude changes also show large relative changes, but there are also parts of Mexico, Southern Brazil, and Central Asia that show large relative increases during El Nino periods. The results shown here provide the first quantitative model to connect ENSO variability and landslide exposure over large areas, showing that a more positive El Nino condition leads to lower landslide exposure in these areas. In South-East China, we show that the converse

relationship exists; a stronger El Nino leads to greater landslide exposure, again in line with rainfall patterns observed in Fig. 1.

## Discussion

These results raise the question of why there are such strong relationships between ENSO and exposure in the highlighted locations. We can explore the impact of each part of the model by contrasting the significance of the relationships between MVEI value and each of: total rainfall, extreme rainfall, hazard nowcasts and exposure. In Fig. 3A, we show the p-values for each district for the relationship between total rainfall and MVEI. It is clear that there are already strong relationships between ENSO and total rainfall in locations where we model large changes in exposure due to ENSO (i.e., South East Asia and Central America and northern South America). To determine whether considering only the extreme rainfall leads to stronger relationships, we compare the p-values for total rainfall and extreme rainfall relationships with MVEI; this is shown below in Fig. 6. In some

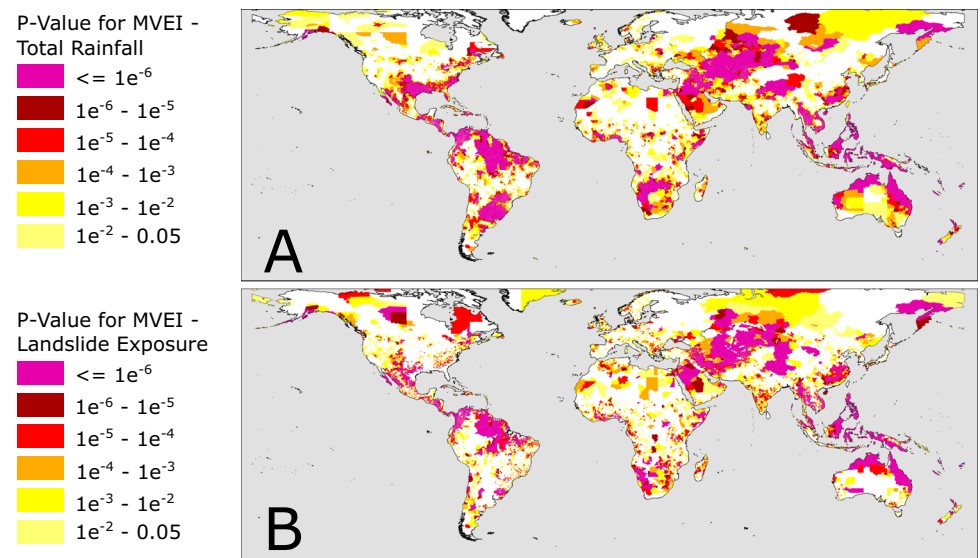

**Fig. 3 Map of the connection between El Nino Southern Oscillation (ENSO) and satellite rainfall data.** These two maps show admin-2 level districts around the world, coloured by the *p*-value of the relationship between monthly Multivariate ENSO Index and total rainfall (**A**) and modelled population exposure (**B**) (both smoothed using a 12-month moving average). Areas shown in white have *p*-values of greater than 0.05, and are not deemed significant. More intense red and pink colours indicate the areas where changes in ENSO are more strongly correlated with changes in total rainfall (**A**) and frequency of extreme rainfall events (**B**).

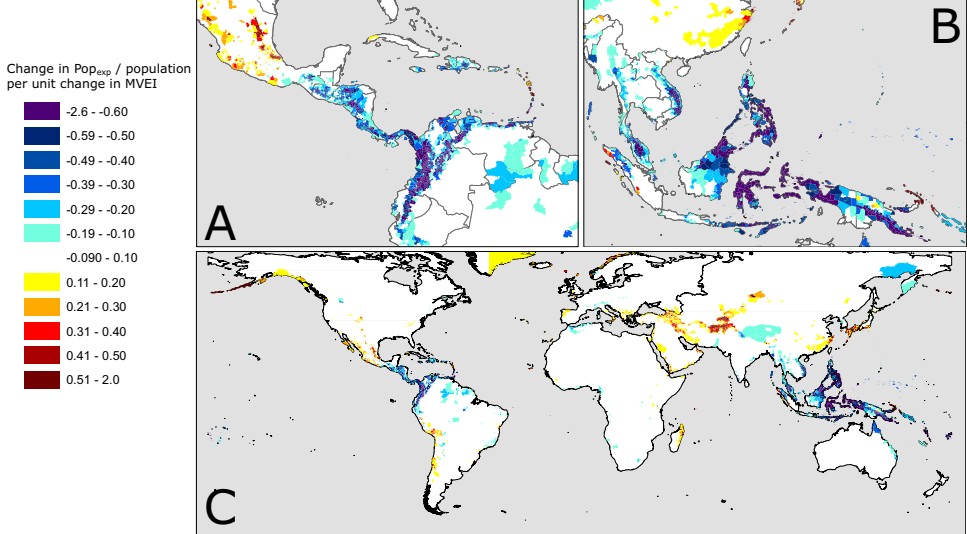

**Fig. 4 Global map of where El Nino Southern Oscillation (ENSO) most strongly impacts population exposure to landslide hazards.** The global map (**C**) of admin-2 level regions shows the shift in exposure for a unit shift in Multi-variate ENSO index. The inset figures highlight areas of Latin-America/ Caribbean (**A**) and SE Asia (**B**) where the ENSO—population exposure relationship is significant (*p*-value <0.05), coloured by the gradient of that relationship. Blue colours indicate that the La Nina state leads to more landslide exposure, and red colours that the El Nino state leads to greater landslide exposure. The units here are changes in average number of days each person is exposed to a hazard nowcast per month for a unit change in multi-variate ENSO index.

geographical areas including parts of Central Asia, Mexico, Iran, China, Luzon, and parts of Thailand, Cambodia and Vietnam, the relationship between MVEI and extreme rainfall is stronger (extreme rainfall p-value is lower than total rainfall) suggesting that the impact of ENSO on extreme rainfall is more significant than on total rainfall in these locations. This is in line with other studies of ENSO-induced rainfall changes[28,29], including global studies based on gauge data[19] and earlier studies using TRMM and GPCP rainfall data[20]. However, the impact of ENSO on extreme rainfall is weaker than on total rainfall in critical areas in

Colombia, Indonesia, and the Southern Philippines, all key locations where we model significant relationships between ENSO and landslide exposure. This suggests that the impact of ENSO on total rainfall, rather than extreme rainfall, may be the main driver of changes in exposure in many locations.

At the same time, ENSO-driven changes in extreme rainfall clearly play a significant role in places like Southern Brazil, Mexico, Eastern China, and parts of Central Asia. Small changes in total rainfall are observed in Central Asia (Fig. 1) but large shifts in exposure are modelled in Tajikistan and Kyrgyzstan,

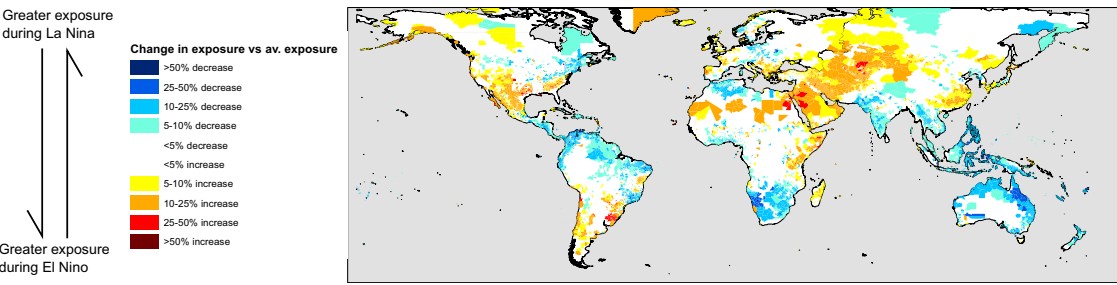

**Fig. 5 Relative change in exposure for a unit change in Multi-variate El Nino Southern Oscillation (ENSO) index value.** This figure shows how much the exposure of populations in each admin-2 district change for a shift of 1 unit in the Multi-variate ENSO index, normalised by the long-term average exposure. Yellow and red colours indicate increases in population exposure during El Nino conditions, expressed in percentage terms vs the long term monthly average exposure, while blue colours indicate increases in exposure during La Nina conditions.

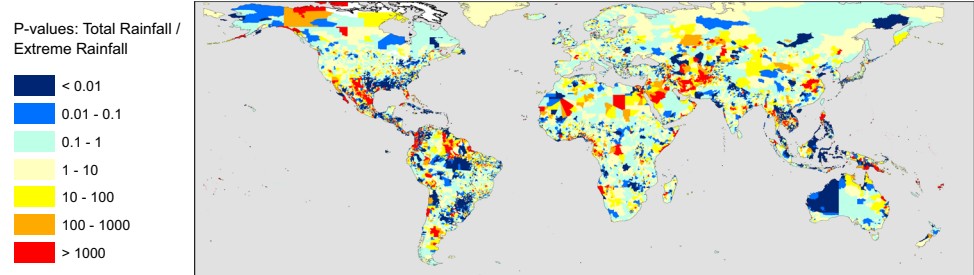

**Fig. 6 Comparison of significance for total rainfall and extreme rainfall connections to El Nino Southern Oscillation (ENSO).** Ratio of *p*-values for the relationship between Multi-variate ENSO index (MVEI) and total rainfall (see Fig. 3A) and MVEI and extreme rainfall (Supplementary Figure 2). Areas with yellow, orange and red colours indicate that extreme rainfall is more strongly correlated with MVEI than total rainfall.

likely linked to changes in extreme rainfall (Fig. 6); this is in line with other studies of ENSO-induced rainfall changes[28,29]. Interestingly, these are the areas where El Nino conditions lead to greater modelled landslide exposure (highlighted most clearly in Fig. 5), while the areas where total rainfall seems to be more relevant are those where La Nina conditions lead to greater impacts (parts of northern South America and Indonesia, highlighted in Fig. 4). This suggests that the two modes of the ENSO system may lead to changes in landslide impact due to differing effects.

We have also compared the strength of relationship for hazard nowcasts and extreme rainfall, and exposure and hazard nowcasts, to test where adding each model input strengthens and weakens the relationship with MVEI (Supplementary Figs. 9 and 10, respectively). While there are broader geographical regions where consideration of extreme rainfall increases or decreases the strength of the relationship, adding the susceptibility and population data has smaller and less regionally consistent effects. This suggests that total rainfall, and to a lesser extent extreme rainfall, is most strongly linked to ENSO, and that the modelled relationship with landslide hazard and exposure is not significantly strengthened or weakened by the addition of susceptibility and population data in the locations we highlight in Figs. 4 and 5. Our outputs are thus relatively insensitive to LHASA model parameter choices, and supports the robustness of the findings. Changes in the strength of the relationship between MVEI and exposure and hazard differ most strongly from the relationships for extreme rainfall in the flat, low population areas of the Tibetan plateau and Amazon rainforest.

There remain some uncertainties and challenges in this type of analysis. Since LHASA model outputs depend on rainfall exceeding the historical 95th percentile, the model is only sensitive to ENSO-induced changes in the frequency of those extreme events, but not to an increase in intensity of extreme events. Studies have shown ENSO-driven changes in rainfall extreme

values in South America[5] and the United States of America;[30] as such, our results may miss some impacts where this is the case. However, other studies have shown that ENSO affects extreme rainfall frequency rather than the peak intensity[9], suggesting that any effect on extremes may vary region to region. Increases in the intensity of events exceeding the 95th percentile may lead to increased impact from landslides. Our model outputs will provide the same hazard and exposure estimate regardless of the intensity over the 95th percentile, and so if ENSO leads to an increase in peak intensity we may underestimate the impacts on population and infrastructure.

There are also other areas where it might be anticipated that ENSO would lead to increased propensity for landsliding, but where the model outputs show no effect—in particular the West coast of the USA. The Western USA has documented relationships between landsliding and El Nino[11], but no evidence of this is seen in our results. The limited satellite rainfall record also prevents us from including the 1997-1998 El Nino event, which was amongst the largest on record. The accuracy of satellites precipitation estimates for extreme rainfall is known to be limited in some scenarios including orographically-enhanced rainfall, short-duration but high-intensity events[31], or mixed rain and snow events that often dominate winter to spring landslide activity in the Northwest coast of the USA. As such there may be limitations in resolving these rainfall effects, potentially affecting the representation of ENSO-induced changes in the overall rainfall difference maps (Fig. 1). We suggest this is an important topic for future research.

Limitations of the model may also affect the scope of our results. The LHASA model works best for shallow landslide events triggered by short-duration rainfall, but it is less suited to assess slow-moving or deep-seated landslides where long-term accumulation of water, influenced by seasonal changes in rainfall, may be important. In addition, extremely short duration (sub-hourly) rainfall events that exceed local intensity-duration

thresholds that do not lead to total daily rainfall exceeding the historical 95th percentile will not be detected. However, the intensity of this kind of rainfall event necessary to trigger landsliding is likely on the order of 10–100 mm/h[4], which is greater than the daily 95th percentile of historical rainfall across most of the continental surface[26]. As such, if these events are captured by the satellite, they will likely lead to a model hazard nowcast. Given the large spatial scale of our model, we are not able to resolve the small local changes in population or infrastructure exposure that may occur within the record period. We suggest that our results provide an overview of areas in which future study could focus on these more locally specific questions. In addition, while we make the simplifying assumption that changes in ENSO impacts are solely related to rainfall, it is possible that ENSO may affect other landslide-relevant factors, including land cover[32]. We suggest that further analysis of the impact of ENSO-driven land cover changes on landslide exposure represent valuable future research topics once homogenous global month-to-month land cover datasets become available.

In addition, while the observed relationships between total rainfall and frequency of extreme rainfall and ENSO result from analysis of rainfall observations, the connections between hazard and exposure depend on model parameter choices. The parameters used in the LHASA model are not empirically defined, and so have been calibrated based on distance to perfect classification of the NASA Global Landslide Catalog[27], which gives a false positive rate of 1% and a true positive rate of between 20 and 50% depending on the time interval used for landslide analysis. The exposure and hazard connections to ENSO must therefore be viewed as model outputs rather than direct observations.

Our model results do not account for lead-lag relationships between the MVEI values and the resulting changes in rainfall patterns. A 12-month moving average for comparison of ENSO to exposure helps smooth out seasonal variability. However, it is possible that if a peak in local rainfall lags behind the peak in ENSO this may reduce the correlation between the two. Evidence for a landslide activity peaking several months after ENSO in India has been shown by previous research[33], while lead-lag relationships between ENSO and rainfall have been documented in South Africa[34]. This suggests that this kind of lead-lag behaviour may explain why no strong connections are observed in India in the current analysis. We suggest exploring lead-lag relationships represents an important avenue for future research.

While there is significant variability associated with ENSO in a number of key locations, similar effects are not clearly observed for the NAO or PDO (despite some studies showing links between landslides and NAO index[35], and others demonstrating links between extreme rainfall variability and PDO[9]). We suggest that the timespan 2001–2018 is too short to capture the full rainfall variability due to multi-decadal PDO and NAO cycles; we suggest future studies may be able to address this more fully. It is not possible to draw any conclusions about the impact of NAO or PDO on landslide exposure here. All data for ENSO, NAO and PDO exposure is available in the supplementary information.

It is important to assess whether the model outputs derived here have any reflection in observational data on landslide impacts. We are not aware of globally consistent temporal datasets of exposure to landslides that can be directly used to compare with our model outputs. However, fatalities resulting from landslides have been recorded between 2004 and 2016 by Froude and Petley[18] around the world. Froude and Petley consider that their Global Fatal Landslide Dataset (GFLD) likely captures the majority of fatal landslides, with only 15% underestimation[2]. In order to compare this data with our model outputs, we have subdivided the landslides leading to fatalities by country. Given that the GFLD contains fewer than 5000 landslide events, there is

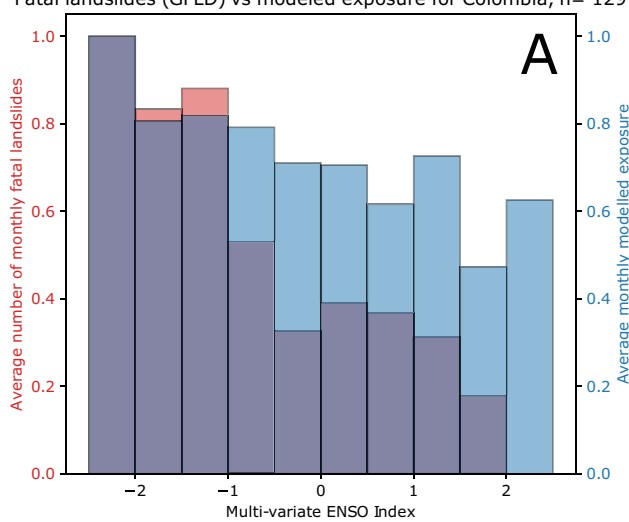

Fatal landslides (GFLD) vs modeled exposure for Colombia, n= 129

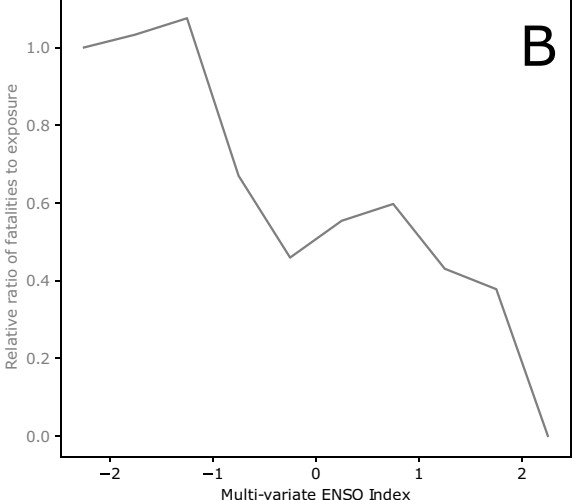

**Fig. 7 Comparison of model data to recorded fatalities in Colombia.** **A** Histogram of average fatal landslide events in Colombia split by Multi-variate El Nino Southern Oscillation (ENSO) index (MVEI) are shown in red; in blue, a histogram of the average modelled landslide exposure split by MVEI value. In the lower figure (**B**), the relative ratio of these two values is shown. A value of 1 in panel **B** indicates that the model perfectly matches the fatality data relative to the maxima of each; lower values indicate that the model over-predicts fatalities.

insufficient data to split them into admin-2 districts as we have for the model output. In each country, we compare the average frequency of fatal landslides for MVEI intervals with the average landslide exposure we model. There are only 26 countries in the GFLD with more than 30 recorded events, so we exclude other countries as we suggest data is too limited to draw conclusions. From these 26, we exclude Bhutan, Italy, Japan, Sri Lanka, Taiwan, and Turkey from further discussion as these show limited or negligible trends in observations and predictions, and further analysis would lead to spurious results using the analysis method used here. The figures for these excluded countries are provided in the supplementary information. In the remaining countries, model performance varies. An example for Colombia is shown in Fig. 7; in the top part of the figure, the histograms of modelled exposure and fatalities are shown normalised to the maximum value for each parameter. Below, the relative ratio of the two is shown; a consistent value of 1 would indicate perfect predictive performance.

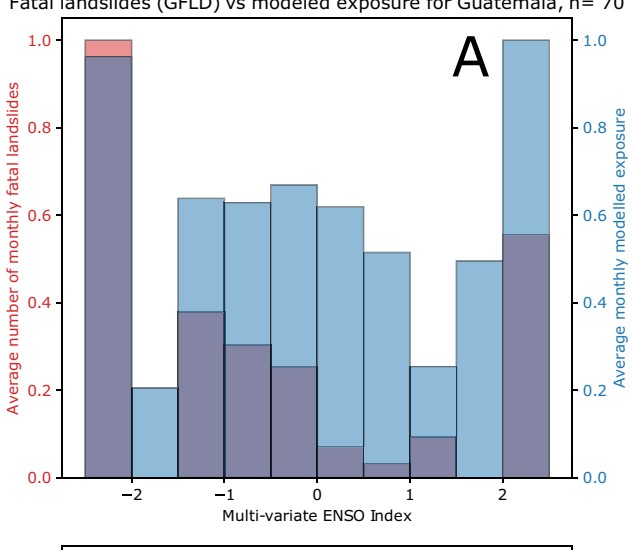

Fatal landslides (GFLD) vs modeled exposure for Guatemala, n= 70

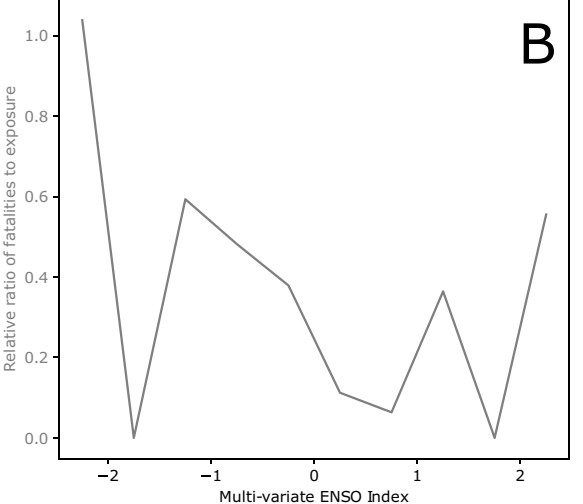

**Fig. 8 Comparison of model data to recorded fatalities in Guatemala.**
**A** Histogram of average fatal landslide events in Guatemala split by Multi-variate El Nino Southern Oscillation (ENSO) index (MVEI) are shown in red; in blue, a histogram of the average modelled landslide exposure split by MVEI value. In the lower figure (**B**), the relative ratio of these two values is shown. A value of 1 indicates that the model perfectly matches the fatality data relative to the maxima of each; lower values indicate that the model over-predicts fatalities.

For most of the analysed countries (11 of 20), the ENSO intervals where the model estimates show the highest average landslide exposure match the ENSO interval where fatal landslides were greatest, but the exposure estimates are also generally less variable than the fatality data. In Fig. 8, we show the results for Guatemala, where the fatality data is far more variable than the exposure model estimates. We suggest that with larger numbers of recorded events, the variability in fatality data may show less noise, but this also suggests that the model outputs tend toward the mean exposure to a greater extent than is reflected in reality—in other words, the model may be overly conservative in estimating the impacts, perhaps suggesting that increases in the peak rainfall intensity are being missed by the model at present. Despite this regression to the mean, the model shows strong similarity to the patterns of landslide impacts in most of the assessed countries, suggesting that our results are a useful first-order estimate of the connection between ENSO and landslide

impacts. We calculate the standard deviation of the ratio of the model estimates and fatality data (lower part of Figs. 7 and 8, see Supplementary Dataset 5 for full table) as a way to quantify how closely the data correspond; the lowest deviations are seen in Indonesia, China, Mexico, and Uganda, with the largest variability (suggesting worse predictive skill) in Thailand, Myanmar, and India (notably countries where the South Asian Monsoon plays an important role). Assessing the specific relationships within individual countries represents an important topic for future research.

Fatality data is not exactly equivalent to modelled exposure, since fatalities resulting from landslides will also depend on the vulnerability of the impacted area to loss. Since our model is calculated based on a single static set of susceptibility and exposure parameters (i.e., population and susceptibility do not change), landslide fatality data may be affected by such changes. However, by assessing the fatality data for MVEI value intervals rather than over time, temporal changes in population are less likely to introduce systematic errors. In addition, the shorter time span over which the GFLD is recorded (2004–2016) compared to our model estimates (2000-2019) does not provide a complete comparison. ENSO may lead to some changes in population vulnerability, as shown by studies in South America[36]. However, we suggest that this remains the best possible comparison of our model outputs with real data based on currently available data-sets, and in several critical countries provides a strong qualitative validation.

With the LHASA model, we can for the first time provide globally consistent estimates of where, and how strongly, ENSO may drive extreme rainfall that triggers landslides, resulting in profound and widespread impacts to people and infrastructure. Medium-term forecasting of ENSO conditions is possible with some degree of accuracy[37], allowing for 1st order estimates of landslide impact to be similarly forecasted around the world. Combined with the seasonality data generated here, this potentially provides a nuanced seasonal forecast for landslides around the world. It will be crucial for future work to validate these findings with specific data on landslide damage and fatalities. We also suggest that using retrospective data may be possible to generate a 'baseline' level of exposure, which can be used to contextualise future changes in population impacts. This may require a longer rainfall time-series than is currently available, but represents a useful future direction for analytical work. In addition, we lack data covering the major ENSO event during 1997-1998, where significant landsliding was triggered in the Americas[38]. A longer term analysis would provide a more comprehensive test of the model outputs and capture more of the variability inherent in ENSO cycles.

On longer time-scales, rainfall patterns are expected to change both in intensity and distribution under different scenarios of anthropogenic climate change[39–41]. Recent work has shown that changes in rainfall under future climate are likely to affect landsliding in the Himalayas in 2100[42], and we also expect changes in landslide patterns in other locations. Increases in global temperature due to climate change are likely to impact ENSO cycles, although modelled predictions suggest ENSO strength may wax and wane in different locations under certain climate change scenarios[40]. We stress that any study considering changes in the future impacts of landsliding due to climate change should also consider the connection with ENSO that we have highlighted here.

In this study, we have for the first time connected major ocean-atmosphere teleconnections—in particular ENSO—with the impacts associated with rainfall-triggered landslide hazards at a global scale. This is a clear demonstration of what we can learn by combining multiple different data types, from satellite rainfall

**Table 1 Explanation of exposure units derived in this study.**

| Parameter | Specific Unit | Descriptive term (shorthand used in this study) | Explanation |
|---|---|---|---|
| Population exposure | Days exposed to landslide hazard x person x. Yr$^{-1}$ / 30 × 30 arc-second cell | Pop$_{exp}$ | The exposure is estimated as number of Nowcasts (i.e. days exposed to elevated modelled hazard) per year in each 30 × 30 arc-second cell multiplied by the population in that 30 × 30 arc-second cell. |
| Road exposure | Days exposed to landslide hazard.km.yr$^{-1}$/ 30 × 30 arc-second cell | Road$_{exp}$ | Sum of Nowcasts per square km multiplied by km of road in that 30×30 arc-second cell. |
| Infrastructure exposure | Days exposed to landslide hazard.element.yr$^{-1}$/ 30 × 30 arc-second cell | Infr$_{exp}$ | Includes the following critical infrastructure categories: hospitals, schools, fuel stations, power generation and transmission |

data, ENSO index estimates, and openly available exposure datasets. Our model results indicate that La Nina conditions lead to greater landslide exposure in a diverse range of settings, with the largest increases seen in northern South America, Central America and the Caribbean, as well as Indonesia, Papua New Guinea and the Philippines. While El Nino conditions seem to lead to fewer areas with increases in exposure in comparison, there are still increases in Central Asia, parts of Eastern China, and Mexico. We suggest that future validation of these results with comparison to landslide event data will be important, and supports a need for comprehensive landslide inventories.

These results are also important when considering where landslides may cause fatalities and damage infrastructure in coming decades under climate change scenarios. In addition, by considering the current ENSO state, our findings allow a range of stakeholders, from disaster response professionals to resource management and planning experts, to qualitatively assess regions where exposure to landsliding may be higher than a typical year. We suggest that similar studies could link flood hazards with ENSO variability at a global extent to provide a more holistic consideration of global hazards in line with the Sendai Goals[43].

## Methods

To assess where ENSO affects rainfall-induced landslide impacts most, we combine landslide exposure data derived by Emberson and coauthors[44] with monthly time series of indices that provide a proxy for the strength and direction of ENSO provided by NOAA[45]. Proxy data is available in Supplementary Dataset 1. The landslide exposure data are estimated by combining an 18-year record of landslide hazard with three socioeconomic datasets to represent potential exposed elements. The landslide hazard data are derived from an updated version of the NASA Landslide Hazard Assessment for Situational Awareness (LHASA) model[26]. The updated model has revised the underlying global susceptibility map to account for recent changes in deforestation[46], (although other model parameters are unchanged) and extends the latitude range from 50°N and S to 60°N and S by making use of the GPM IMERG v06B rainfall data[15]. The susceptibility model used by the LHASA model uses a heuristic weighting of input parameters including slope, land cover, distance to road networks, and lithology. The susceptibility model has been validated using the NASA Global Landslide Catalog, with nearly 5000 recorded landslide events used for validation. The LHASA model takes the IMERG satellite-based 7-day accumulated rainfall as an input with an exponential weighting toward more recent rainfall (exponent = 2). When it exceeds the historic 95th percentile (calculated over the GPM IMERG data period, 2001-2019) and has a value greater than a minimum threshold of 6.6 mm to remove predominately arid regions, a landslide hazard 'nowcast' is output at 30 arc-second resolution at a daily time-step. Nowcasts are excluded if the susceptibility of the pixel is lower than 'moderate', using susceptibility classes defined by Stanley & Kirschbaum;[47] moderate or greater susceptibility covers approximately 20% of global land area. Several combinations of weighting coefficients and spatial windows were tested, and the best predictor of landslides was selected on the basis of distance to perfect classification. We consider these 'nowcasts' as a proxy for landslide hazard. While this binary classification of hazard is coarse on a daily basis, aggregated on monthly time-scales it provides a more nuanced perspective on changes in hazard.

The hazard outputs can be combined with exposure data for three elements: population, roads, and critical infrastructure. Population estimates are derived from the Gridded Population of the World, Version 4 (GPWv4) dataset[48], roads

from the Global Roads Inventory Project (GRIP)[49], and infrastructure from OpenStreetMap (OSM)[50]. Each of these datasets are open-access, global in extent, and recently updated. While OpenStreetMap has some variability in completeness between different countries[51], the open availability and global extent represents an excellent data source. We can account for changes in completeness by normalising the estimates of infrastructure exposure by the total infrastructural elements – i.e., the exposed fraction. We do not use dynamic values for population, roads, and infrastructure (i.e., we use a single value for each point in the raster for the entire 18-year analysis period). This means our model results can only be considered as representing changes in exposure relative to the specific time-period for these exposure datasets. We use the 2015 data for roads and population, and a snapshot of OSM data for July 2018.

To convert the linear GRIP data and point OSM data to raster density maps, we calculate the road-length in km per cell, and point density per cell respectively. This provides global rasters of population, road density, and infrastructure density that can be multiplied by the number of monthly nowcasts per cell to obtain an estimate of exposure. The units for each of these estimates are shown in Table 1.

The outputs from this assessment are daily landslide exposure estimates, which we sum up for each month to generate 216 monthly estimates (January 2001 through December 2018) of global landslide exposure at 30 arc-second grid resolution for population, roads, and critical infrastructure. We focus on population exposure in the discussion below, but all data generated are available in the supplementary information for this study. These monthly data are then comparable with the monthly MVEI values. The monthly values for each admin-2 level district are available in supplementary dataset 2. The population, road and infrastructure counts in each admin-2 district are available in supplementary dataset 3.

While climatic trends are generally considered on timescales of 30 years or greater, this 18-year exposure dataset is the longest available based on consistent satellite rainfall data. We utilise the newly reprocessed IMERG version 6B rainfall product, which merges and homogenises data from NASA's Global Precipitation Measurement (GPM) mission with its predecessor Tropical Rainfall Measurement Mission (TRMM)[52]. The use of satellite rainfall data allows a globally homogenous estimate of landslide hazard and exposure, rather than relying on recorded landslide events where spatial and temporal biases in reporting may be significant[53]. Model estimates are dependent on parameter choice, and we discuss model assumptions in more detail below. These monthly estimates of exposure to rainfall-induced landslides can be directly compared with the monthly estimates of ENSO. To establish whether the model results are a useful approximation of exposure, we have compared our results with fatality dataset derived from the dataset of Froude & Petley (2018). Comparison data with the fatality data are found in Supplementary Dataset 4.

While we calculate exposure as a 30 × 30 arc-second pixel raster, we have chosen to aggregate our estimates to administrative districts. The GPW v4 population data is presented at 1 km resolution, although in some less-developed parts of the world the census data used to generate the data is not of fine enough resolution to distinguish 1 km-scale differences[54]. We suggest that by providing model outputs at a level-2 administrative district level (e.g. county, municipality, prefecture level), we avoid drawing attention to single pixels where extreme local rainfall or extremely dense population could bias any comparison with tele-connection indices. We derive the admin-2 areas from the GADM (Global Admin - https://gadm.org/about.html) project data for administrative districts. The admin-2 level areas are generally small enough to preserve local detail; larger areas (e.g. defined by national borders) may extend across areas with diverse ENSO effects (e.g. Mexico – Fig. 1). In addition, these outputs can be easily understood by disaster mitigation policy-makers. These admin-2 level data can be easily aggregated by end-users to national level estimates. Many admin districts are not tied to topography or climate, and where admin regions include diverse landscapes (e.g., mountains and flat areas), it may be more difficult to pinpoint key locations for exposure. If exposure estimates are aggregated in such districts, a signal of a relationship between exposure and ENSO may be masked

by noise or conflicting signals from a different part of the area. Smaller admin districts are less likely to merge conflicting signals, and we suggest that the admin-2 level offers the best balance of global reproducibility and small scale to allow for a focus on local effects.

For each year, we can express the exposure estimates $E_j$ for each admin district $j$ as follows:

$$E_j = \sum_{i=1}^{n} (R_i S_i P_i)_j \qquad (1)$$

Where $R_i$ is the number of days when rainfall exceeds the historical 95th percentile in cell $i$, $S_i$ is the susceptibility of that cell (1 if the cell has susceptibility greater than medium, 0 otherwise), and $P_i$ is the population across that cell. Summed for all cells 1-$n$ in district $j$, this gives a final unit of person-days/year, expressing the average exposure for that district. This can then be normalised by the total population in region $j$ (e.g., Fig. 4) or by the long term monthly average exposure (e.g., Fig. 5) to allow for intercomparison.

## Data availability

All data associated with this study is available in the supplementary information. A shapefile containing all p-values and slope of relationships calculated here is a large file (~1GB) and at time of submission we are obtaining permission to share this in a public repository. Requests for this data can be made directly to the corresponding author and will be fulfilled as soon as possible.

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

## Acknowledgements

All material necessary to replicate these results can be found in the supplementary information. The authors have no conflicts of interest, financial or otherwise. D.K. and T.S. are supported by a NASA DISASTERS programme grant 18-DISASTER18-0022. R.E. is supported by a NASA Postdoctoral Fellowship administered by Goddard Space Flight Center.

## Author contributions

All authors (R.E., D.K., T.S.) were involved in study conceptualisation and writing of the manuscript. R.E. carried out modelling and analysis.

## Competing interests

The authors declare no competing interests.
