## [Peer Review File · Nature Communications]

Reviewers' comments:

Reviewer #1 (Remarks to the Author):

Review of Emberson et al. 2020 Nature Comms by David Milledge

This is a really nice paper addressing an important question and doing so at a global scale. It is novel in that it is, to my knowledge, the first global scale analysis of the possible influence of ENSO on the frequency of landslide triggering rainfall. At first sight the paper appears to be an entirely model-based exercise and it is true that it contains no landslide observations and relies on an existing model to convert accumulated rainfall to landslide exposure. However, it may also be one of the longest and most complete analyses of satellite rainfall data (I'm less well placed to gauge novelty here). The correlations between the frequency of landslide triggering rainfall and ENSO are startlingly good in some areas and that makes the paper particularly interesting. The global scale and the step to landslide exposure suggest that the paper is likely to be of widespread interest and of interest beyond the landslide research community.

My main concern with the paper is that at present the drivers behind the relationships that it uncovers appear somewhat unclear and I don't think they need to be. Having read the paper my understanding is that: ENSO introduces temporal clustering in extreme precipitation events (defined in relative local terms using 95th percentile of rainfall intensity) and thus alters annual frequency of these events. In landscapes that are susceptible to landslides this clustering and the resultant change in frequency of landslide triggering storms results in a strong association between annual averaged landslide hazard and ENSO. In the subset of these locations where there are many people or assets their exposure to landslide risk follows the same trends.

The key missing component from the current study is a straightforward uncensored analysis of correlation between ENSO and extreme precipitation worldwide before landslide susceptibility or exposure is introduced. This analysis could focus on landslide relevant precipitation events i.e. 7 day accumulated rainfall with a weighted kernel that accounts for drainage (as you currently use). It would also be useful to know whether the steps from correlation between ENSO and extreme precipitation on to landslide hazard then to landslide exposure result in amplification of the correlation or in damping. I strongly recommend that the authors investigate and report that behaviour since I think it would help readers both to understand what is driving the headline results and to trust the headlines themselves.

Major comments

Because this is primarily a modelling exercise more detail is needed on the model and more space needs to be devoted to explaining / understanding what is driving the model response. In the absence of a comparison with observations this is likely to be the primary way that readers gain trust in your findings.

Why is ENSO so well correlated with landslide exposure in some places? Is it simply due to the changing frequency of large storms (e.g. doubling?); or because large storms become preferentially more frequent in the most exposed or landslide susceptible locations (i.e. Amplification of the effect by the landslide risk generation process); or is the correlation to storm frequency even stronger and the landslide risk generation process damps the signal. Understanding which of these three alternatives is at work or perhaps where they are at work would be a significant contribution and you are ideally placed to make it.

If the dominant driver is changes to the annual frequency of intense rainfall under different ENSO

conditions then the literature on ENSO effects on extreme rainfall needs to feature more strongly in the paper. It is not clear to me from the introduction what and how much is already known about the relationship between ENSO and the frequency, duration and intensity of rainfall events.

The units 'nowcasts per km² per year' or 'person-nowcasts per km² per year' are difficult to interpret in absolute terms. It would be useful to explain how the reader should interpret these units and how to relate them to hazard in terms of a landslide probability. If this is not possible you should say so explicitly as this would prevent people getting distracted by trying to interpret them. In many cases you are more interested in trends anyway I think it should be possible to normalise the nowcast units so that they are more physically meaningful. I suggest trying something like the percentage oversampling of nowcasts relative to an expected number of nowcasts (which should be feasible given that you use a percentile based approach and a static binary susceptibility layer). Alternatively the approach you take in discussing the Planadas district where ENSO related change in exposure is normalised by the amplitude of the seasonal exposure change would be another good way to go.

Finally, more detail on two aspects of the modelling is required either in the paper itself and/or in a Sup Info.

1) In the methods, a little more detail on how the model is needed but also an indication of its assumptions and its parameters.

2) The paper needs a more robust treatment of the model's structural, input and parameter uncertainty and its impact on your findings. This is the primary limitation to an otherwise simple and elegant research design. You discuss uncertainty briefly in the paper but it is not clear what impact it has on your core findings. A full uncertainty analysis would be a huge endeavour. This is one of the reasons I think it would be worth demonstrating the relationships between ENSO and uncensored frequency of intense storms. This would considerably simplify the task of uncertainty estimation (though I would still be keen to know whether the findings were sensitive to the choice of threshold percentile, ARI window length and weighting factor).

Specific comments:

L45-50: The 'however' here does not seem to fit, how does the new rainfall data address the problem of limited inventories preventing modelling?

L100: the focus on ENSO, but results for NAO and PDO in the supplementary material seems a sensible approach.

L117-121: More detail is needed on how the model works, specifically:

- L117 "updated version of the NASA Landslide Hazard..." what has changed in this updated version relative to Kirschbaum and Stanley (2018)? The changes themselves need to be reported here.

- L119 "7 day accumulated rainfall" is this accumulated using the weighted kernel described in Kirschbaum and Stanley (2018)?

- L120 "historic 95th percentile" what is the timescale over which this percentile is calculated

- What is the timestep for the model? This is important for the reader to interpret the metrics since they are accumulated nowcasts per year.

- L120 "locations of moderate to high landslide susceptibility" are these the LHASA nowcast classes or the Stanley and Kirschbaum susceptibility classes? A sentence or two summarising the susceptibility model would be useful here.

- L120 what fraction of the globe is covered by the moderate to high landslide susceptibility class? This is important because it provides an indication of how similar landslide hazard would be to simply the number of times per year that the 95th percentile in precipitation is exceeded.

- L121 "we consider these 'nowcasts' as a proxy for landslide hazard" I think the number of nowcasts per year is your hazard proxy, otherwise the proxy is binary.

- Do you still use an absolute ARI threshold of 6.6 mm?

L120: A binary landslide susceptibility classification of landslides i.e they are possible or not seems a very coarse classification. It may be that it doesn't matter for your results because susceptibility is static in time but I think you need to explain more clearly that you assume this binary susceptibility, justify it, and ideally also test it (though I recognise the latter is probably out of scope).

L125: How do you know that this variability is not introducing bias? Your approach to focus on population density with the remainder in SupInfo mitigates the impact of the potential biases in road and osm data. Perhaps you could re-phrase when you introduce these datasets to make your focus on population density clearer. It may avoid people getting distracted as I did. This is only a suggestion though, fine to leave as is if you disagree.

L145: "...allows a globally homogenous estimate of landslide hazard and exposure, rather than relying on recorded landslide events where spatial and temporal biases in reporting may be significant." This is true, however it also introduces a dependence of the findings on the model and parameter uncertainty. What is the sensitivity of your findings to:

- the choice of 95th percentile in ARI for normalising
- ARI parameters: 7 day window and weighting factor of -2 (calibrated from 2007-13 landslide data but I guess that other values gave only slightly worse fits).
- The absolute ARI threshold of 6.6 mm if this is still used
- The parameters of the fuzzy overlay model that defines landslide susceptibility and therefore whether or not rainfall above 95th percentile will trigger a nowcast.
- The choice to nowcast only for areas of moderate to high landslide hazard

A full uncertainty analysis here is a huge endeavour. If you can demonstrate the relationships between ENSO and uncensored rainfall accumulation it would indicate that your results are retained in this simpler case.

L180: "(nowcasts per person)" I think this should be person-nowcasts per km² per year.

L199: "(see Figure 2) is greater than 0.2" why use Rsquared rather than a p value? Why choose this threshold? It seems quite low (20% of variance in annual nowcast frequency is explained by ENSO) it would be useful if you could explain why Rsquared of 0.2 is perhaps better than our instincts suggest in this situation.

L207: "significant relationships" this would be worth supporting with a test of the significance.

L216: "we show the magnitude of the change driven by a unit shift in MVEI index" How is this calculated? Is it the gradient of the linear least squares regression? How did you decide that a linear function was the most appropriate? The relationship looks pretty linear in Fig 2 but did you try other functional forms? It would certainly be worth remarking on the linearity in Fig 2 and confirming that this is a general property of the data so that this later step is more comfortable for the reader.

L225 Fig 4: Normalising by population makes sense but population was already in the numerator so this could be simply change in nowcasts per km² per year with changing ENSO. I wonder if it would be helpful to express this as a fraction of the nowcasts per km² per year or a percentage. This may cause problems for the linear regression but it would seem worth trying because the existing units are difficult to interpret in absolute terms.

L231: "change in exposure can be nearly as large as the seasonal shifts in exposure..." This is a very good way to express the change, could you do this more generally (e.g. for the colorscale on Fig4)?

L244: "unclear what is driving the effects seen in our model results" Because this is a modelling exercise this seems a strange statement. Your ability to explain these unexpected results is central to building trust in the model, particularly in the absence of a comparison with observations. Can you establish whether the effect is retained when susceptibility and exposure are not included? Can you identify whether the effect is related to the percentile threshold (e.g. very low) in this region or to the changing shape of the ARI distribution or both.

L251: "if ENSO makes already-intense rain storms even more intense" I struggled to understand what an already intense rain storm was. Perhaps a storm that would have been intense anyway or a storm in a location where intense rain storms are common? I think I now understand: because you record the frequency of storms above a threshold intensity you cannot account for increased intensity of storms above this threshold. I wonder if you could clarify this in the text.

L307: more could be said in the conclusions about which regions experience the strongest landslide exposure changes under ENSO, which regions experience positive and negative correlation and how those changes relate to the shorter term seasonal variability. These are all interesting and useful findings from earlier in the paper and it would be very helpful to summarise them here.

Finally, this is a very good paper that makes a valuable contribution and I would expect that the authors should be able to address all these comments fairly easily. I hope that they find the comments helpful and enable them to improve an already very good paper.

Reviewer #2 (Remarks to the Author):

Review

Global connections between El Nino and landslide impacts

General Comments

The described method of obtaining a globally homogenous estimate of landslide hazard and linking it with ENSO impacts is interesting and can be useful. I recommend acceptance after the authors have answered the comments, and met the requests/suggestions detailed in the following.

It would be informative to report if there has been any attempt at validating its results with real recorded landslide events, and what were the results of this assessment (specific comments #8 and 30). There is an issue regarding statistical significance (comments #16 and 20). Besides, there are some changes needed in the text to make it clearer and more informative. They are also detailed in the specific comments below.

I think the authors will be able to easily meet the requests, and that the paper will give a good contribution to different but connected fields.

Specific Comments

1. Pages 1, line 28: It is mentioned that the incidence of widespread landsliding can vary on seasonal to multi-annual timescales and a reference to the influence of El Niño on extreme rainfall events in South America is added. It is interesting to comment that the decadal/interdecadal modulation of ENSO events provided by IPO (or PDO, which is related to IPO) does also provide an interdecadal modulation of extreme rainfall events. A clear example is available in South America, as shown in: Grimm, A. M., N. C. Laureanti, R. B. Rodakowski and C. B. Gama, 2016: Interdecadal variability and

extreme precipitation events in South America during the monsoon season. *Climate Research*, v. 68, n. 2-3, p. 277-294. DOI: 10.3354/cr01375

In this reference, the South American main modes of precipitation interdecadal variability (which are influenced by IPO and PDO) are shown to affect significantly the frequency of extreme events in two of the regions in which the landslides are most influenced by ENSO, according to this manuscript, Colombia and South Brazil.

2. Page 2, lines 51-59: The reference above shows why shifts in exposure are observed with respect to changes in PDO state, since this oscillation, as ENSO, affects the frequency of extreme events even more significantly than the monthly or seasonal rainfall totals. It proves that this decadal variability mode affects the frequency of extreme events that can produce landslides.

3. Page 3, lines 83-84: With 'The sign of the ENSO index changes on the order of 1-5 years' are you suggesting that ENSO has periods from 1 to 5 years? Usually the shortest period attributed to ENSO is 2 years. The reference cited for this statement (16) says that the power spectrum of MEI (an ENSO index) 'shows a concentration of energy between 60 and 20 months', and evidenced two harmonics, of 26 and 50 months.

4. Page 3, lines 88-93: There are also significant effects of PDO on rainfall and its extreme events over South America, in regions that appear in this manuscript as having high landslide exposure and high ENSO-related landsliding exposure, as, for instance, Colombia and South Brazil. See the reference Grimm et al. (2016), mentioned above (comments # 1 and 2).

5. Page 5, line 119: It seems that 'where' should be replaced with 'when'.

6. Page 5, line 123: It is convenient to inform what is GPW. Therefore, in line 123, please write '...are derived from the Gridded Population of the World, Version 4 (GPWv4), roads from...'

7. Page 5, line 127: Some readers may not notice what GRIP and OSM mean. Therefore, please write in line 124: '...from the Global Roads Inventory Project (GRIP), and infrastructure from OpenStreetMap (OSM)', and use this acronym also in line 125.

8. Page 6, lines 143-145: Although I agree with your method of obtaining a globally homogenous estimate of landslide hazard, I and probably other readers would be curious to know how its results compare to real recorded landslide events. I understand that data are not homogeneous, sometimes not reliable (biases in time and space), but has there been any attempt at validating the results of this method, at least for real recorded landslide events? What were the results of the assessment? Please mention them and summarize them.

9. Page 6, lines 161-163: Why mention only the northern hemisphere monsoon season? There are also southern hemisphere monsoon systems, such as Australia, southern Africa and South America. See <http://www.clivar.org/clivar-panels/monsoons> or the Fig. 1 of a very recent paper on the global monsoons:

Wang, B. et al., 2020: Monsoon Climate Change Assessment. *Bulletin of the American Meteorological Society*. <https://doi.org/10.1175/BAMS-D-19-0335.1>

The rainfall annual cycle shows a very marked difference between winter and summer in these regions. Therefore, I would suggest rewriting these lines as: 'As demonstrated by other studies, the seasonal variability of landslides is primarily set by the dominant rainfall regimes, such as the monsoon regimes or the tropical cyclone seasons around the world.'

10. Page 6, lines 166-167: When does the 12-month moving average of exposure with a window

starting 12 months prior to the month in question end? When the window ends 12 months after the month in question?

11. Page 7, line 186: I suggest replacing `...which removes annual trends...` with `...which removes the annual precipitation cycle...`.

12. Page 7, line 187: I suggest replacing `...in the strength of annual impacts...` with `...in the strength of seasonal impacts`.

13. Page 7, lines 189-192: I suggest replacing `...the differences in seasonal variability, too; Kinnaur has a...` with `...the differences in seasonal variability: Kinnaur has a...`.

14. Page 8, line 195: I suggest replacing `In the handful of locations illustrated in...` with `In the locations illustrated in...`.

15. Page 8, line 196: I suggest replacing `The next step is to map these trends globally,...` with `The next step is to map this influence globally,...`. ENSO does not produce trends, but variations.

16. Page 8, lines 198-201: Why are you using this threshold of 0.2? I think the more correct procedure here would be to calculate the threshold for which R-squared is significantly different from zero with a chosen level of significant (for instance, 0.05 or even 0.10). It might even result to be 0.2, but you should have an acceptable reason for choosing 0.2. Or at least inform what level of significance corresponds to 0.2.

17. Page 8, line 205: It seems that `...and population exposure exceeds 0.2.` should be `...and population exposure exceeding 0.2.`

18. Page 8, lines 207-211: In the first sentence, it is said that significant relationships with ENSO are observed in Latin America, which is true. Latin America is a group of countries in Americas where the predominant language originated with the Latin language, such as Spanish, Portuguese, and French. Brazil is one of these countries. But then, in the second sentence, it is said: `There are a number of other locations, such as in central Brazil, ...`, as if central Brazil were not in Latin America. I suggest replacing this sentence with `There are a number of other locations, such as in Central Asia, and Southern Africa, where varying degrees of...`.

19. Page 8, line 220: I suggest replacing `...we show two of the key locations – SE Asia, and Latin American and the Caribbean.` with `...we show two of the key locations – SE Asia, and the northern part of Latin America.`

20. Page 9, line 225: Significant at what level of significance?

21. Pages 8 and 9, Figs. 3 and 4: The comparison of Figs. 3 and 4 is intriguing. In Fig. 3 there are extensive areas with great R-squared values (around 0.5-0.6) of the relationship between Multivariate ENSO Index and population exposure in central Amazon, which most probably are significantly different from zero. However, in Fig. 4, at the same location (lower right corner of the upper right panel, for northern Latin America) there are almost no areas with significant ENSO - population exposure relationship. What am I missing?

22. Page 9, line 232: I suggest replacing `...as large as the seasonal shifts in exposure -...` with `...as large as the seasonal variations in exposure -...`.

23. Page 9, lines 231-233: How did you calculate this? Please show with numbers that for Planadas (Colombia) the difference between Pop(exp) for La Niña and El Niño, obtained from Fig. 4, is comparable to the difference for July and April, obtained from Fig. 2. It would be a real quantitative example.

24. Page 10, lines 240-244: There are articles that show the ENSO effect on Central Asia. Below are two of them, but you can find other ones:

Mariotti, A., 2007: How ENSO impacts precipitation in southwest central Asia, *Geophys. Res. Lett.*, 34, L16706, doi:10.1029/2007GL030078.

Jialin Lin, J. and T. Qian, 2019: A New Picture of the Global Impacts of El Niño-Southern Oscillation. *Scientific Reports*, 9:17543 | <https://doi.org/10.1038/s41598-019-54090-5>

25. Page 10, lines 250-260: I do not agree with your argument in lines 250-254, and thus I think you can remove this uncertainty. There are two possible impacts of ENSO on extreme events. It can enhance the intensity of extreme events or the frequency of extreme events or both. The most common impact is of the second type, on the frequency (number in a time period) of extreme events. You seem to be considering that the first type is the most important, and this is not true, since the areas undergoing the first type of impact are more scattered than those undergoing the second type. This is shown in Grimm and Tedeschi (2009) for South America. By the way, the reference that was cited regarding USA (33, Gershunov and Barnett 1998) only shows the influence of ENSO on the frequency of extreme events in USA. If El Niño (or La Niña) increases the number of extreme events (or the number of events exceeding the 95th percentile), as in several regions in South America and USA, obviously it increases the exposure to landsliding. Therefore, this kind of uncertainty would be rarer, and you could remove it. Yet the other sources of uncertainty (lines 254-260) are more concrete: the limited satellite rainfall record and the limited accuracy of satellites precipitation estimates for extreme rainfall.

26. Page 10, lines 271-273: It seems that there is a full stop in the middle of this sentence, in line 273 (between 'ENSO' and 'this'). Remove it, or withdraw the word 'if' in line 272.

27. Page 10, lines 271-277: What do you mean by 'peak in local rainfall lags behind the peak in ENSO' or 'lead-lag relationships between ENSO and rainfall'? Is it the time lag between the largest absolute MVEI index and the largest relative impact on rainfall (percentage variation of climatological rainfall)? This lag is pretty frequent, since the teleconnections responsible for the ENSO impact depend on the basic state of the atmosphere in which they propagate. For instance, in South Brazil, where there is a relatively strong relationship ENSO-landslide exposure in your Figs. 3 and 4, the maximum relative impact happens in Oct-Nov and not in the typical height of ENSO during late Northern Hemisphere Fall/early Winter.

28. Page 11, line 279: I would not use this word 'trend', but 'variation'. Trend has a connotation of continuous increase or decrease, which is not the case of ENSO impacts.

29. Page 11, lines 279-284: The lack of signal or the weak signal showing link between landslides and PDO (or IPO, since they are related) is most possibly due to the short timespan (2001-2018) of this study preventing detection of the full rainfall variability due to interdecadal PDO (or IPO). The study by Grimm et al. (2016, mentioned in comments # 1 and 2) shows clearly that there is a very significant impact on the frequency of extreme events in several regions of South America, and therefore, on the landslide exposure.

30. Page 11, lines 291-292: I think that some validation results for the model, even if not related with ENSO, should be mentioned, as said in comment #8, since before using the model to link landslide

exposure to climate variations it is important to know its reliability in reproducing the occurrence of landslides.

Signed: Alice M. Grimm

Reviewer #3 (Remarks to the Author):

The topic addressed by this paper is of some interest. It has long been argued that changes in rainfall pattern associated with a large El Nino event might well cause changes in landslide occurrence. This hypothesis largely emerged in the aftermath of the 1997-98 El Nino event, which caused large-scale landsliding in the Americas. However, there is little validation of this idea, and indeed some evidence that La Nina conditions might be more important than El Nino on a global scale.

To that end this paper is welcome, but unfortunately I do not believe that it meets the standard needed for publication.

I would like to highlight the following particular concerns:

1. The paper makes some reference to NAO and PDO in the text, but more importantly in the supplementary information. It is simply not possible to examine the impact of decades long cycles using a dataset that does not completely cover two decades. The result is an analysis that is spurious and should be removed.
2. The authors fail to recognise that there is not a single type of El Nino (or La Nina), and that rainfall patterns vary considerably between these events. The authors should consider Kug et al (2009) (<https://doi.org/10.1175/2008JCLI2624.1>) for example. It is likely that these different types of El Nino conditions generate very different rainfall patterns. The paper should consider this carefully - thus, comparing a single peak El Nino and a single peak La Nina might have very little validity.
3. I have a concern that the analysis does not consider the types of precipitation that generates landslides. There are two types (to generalise vastly) - long duration moderate intensity - the "wet winter" scenario - and short duration, high intensity - the "cloudburst" scenario. In general, the majority of high cost landslides occur in the latter scenario, but this is barely captured by the dataset considered here. Indeed, it is in monsoon conditions entirely possible to have a drier than usual month with a record breaking rainfall event.

These are fundamental points that undermine the premise of this paper, rendering it unpublishable in my view.

I will pick up some more detailed points as well:

This should be the South Asian summer monsoon (not India). Tropical cyclone seasonality - again, this is related to TCs in the NWP, the Indian Ocean and the Atlantic, primarily

Line 51: I do not believe that the PDO and NAO results are statistically valid (see above), so this should be removed.

Line 77: this is a huge assumption that needs justifying. El Nino may drive changes in, for example, vegetation that may be really important too. Does it change vulnerability of population as well? This is not considered.

Line 125: It is not clear to me whether the exposure data was dynamic or static - in other words, for each point in the raster, was a single value used for the entire 18 years, or was this updated with time? There have been huge changes in some of these values (e.g, road density) in this period. If a single value was used, which year was considered?

Line 154: I can see the logic and strengths of aggregating to admin level, but does this cause problems? Many admin boundaries have no logic in topography or climate. Wont this approach lead to a level of confusion, for example where an admin area covers both a mountainous and a flat area of terrain?

Line 165: Is it really valid to use 12 month smoothed data in this way? If the rainfall patterns typically occur in a 3 moth period (the summer monsoon), won't the signal be lost? This approach must also miss the effect of tropical cyclones, which generate rainfall for 48-72 hours, typically. Earlier in the paper, TCs were identified as being important.

Line 202: This diagram needs further explanation. Some of the areas identified have very few landslide records in the various datasets - e.g. N. Australia, Tibet. South Asia and Taiwan, both of which are very landslide prone, show no effect. Why is this?

Line 225: This is an interesting result, but it needs explanation. What is the meaning of a shift in MVEI index? Te finding that La Nina is more important than El Nino in Central America is a surprise. This was an area that was severely affected by El nino rainfall in 1997-8, primarily because of tropical cyclone impacts.

Line 263: I agree with this analysis of the model, but the way that is being used here seems more in line with movement of large deep landslides rather than shallow, rapid ones.

Line 309: The conclusions are not reasonable, but the uncertainties in the model mean that it could not be used to anticipate the impact of future El Nino and La Nina events.

Dear Dr. Emberson,

Your manuscript entitled "Global connections between El Nino and landslide impacts" has now been seen by 3 referees, whose comments are appended below. In the light of their advice I regret to inform you that we cannot publish your manuscript in Nature Communications.

You will see that, while the reviewers find the topic of your work of interest and acknowledge, they raise substantive concerns that cast doubt on the strength of the novel conclusions that can be drawn at this stage and the robustness of the findings. Unfortunately, these reservations are sufficiently important to preclude publication of this study in Nature Communications. While we would not rule out consideration of a fully and thoroughly revised manuscript in future that presents a much stronger case in linking global landslide occurrence to El Nino (some of the key points being the analysis of correlation between ENSO and extreme precipitation; model validation with real, recorded events; the, according to referee #3, too short dataset; recognition of various rainfall patterns generated by El Nino; consideration specifically of the type of precipitation that generates landslides) it is clear that a very significant amount of additional analytical work and clarification would be necessary in order to support your main conclusions, and it is far from clear whether this is possible and whether the results will continue to appear impressive in the light of this additional work and discussion.

If you opted into the journal hosting details of a preprint version of your manuscript via a link on our dedicated website (<https://nature-research-under-consideration.nature.com>), please note that we will now remove these details as your manuscript is no longer under consideration at Nature Communications. For more information, please refer to our FAQ page at <https://nature-research-under-consideration.nature.com/posts/19641-frequently-asked-questions>

I am sorry that we cannot be more positive on this occasion and thank you for the opportunity to consider your work.

Best regards,

Sebastian

Dr Sebastian Mueller

Associate Editor

Dear Editor,

Thank you for taking the time to collect three constructive and thoughtful reviews of our study. We appreciate that the comments raised precluded publication of the first iteration of our study. The points raised by each reviewer have allowed us to revise and (we hope) significantly improve our study. We feel that none of the comments significantly undermined the central findings of our study, nor the novelty of the results. In fact, the comments have pushed us to seek out independent validation of our model results with real landslide fatality data, which we find is supportive of our model results in key locations. We feel that some of the critical comments, particularly those of reviewer 3, stem from a misapprehension of the approach we have taken, and in revision we have taken time to amend text that may have led to confusion.

We hope that you will consider the revised version of our study, as we feel that we have effectively addressed all of the comments from reviewers and improved our work in doing so. Each of the comments are addressed in the text below.

Yours faithfully,

Robert Emberson, on behalf of all authors.

Reviewers' comments:

Reviewer #1 (Remarks to the Author):

Review of Emberson et al. 2020 Nature Comms by David Milledge

This is a really nice paper addressing an important question and doing so at a global scale. It is novel in that it is, to my knowledge, the first global scale analysis of the possible influence of ENSO on the frequency of landslide triggering rainfall. At first sight the paper appears to be an entirely model-based exercise and it is true that it contains no landslide observations and relies on an existing model to convert accumulated rainfall to landslide exposure. However, it may also be one of the longest and most complete analyses of satellite rainfall data (I'm less well placed to gauge novelty here). The correlations between the frequency of landslide triggering rainfall and ENSO are startlingly good in some areas and that makes the paper particularly interesting. The global scale and the step to landslide exposure suggest that the paper is likely to be of widespread interest and of interest beyond the landslide research community.

We are grateful to the reviewer for the comprehensive and constructive review. It is clear that the reviewer put a lot of thought into providing comments, and we feel that by incorporating the feedback from the reviewer we have been able to improve our study significantly. We greatly appreciate the reviewer's interest in our findings. Below, we address the comments individually, and discuss the changes made to the paper in light of each.

My main concern with the paper is that at present the drivers behind the relationships that it uncovers appear somewhat unclear and I don't think they need to be. Having read the paper my understanding is that: ENSO introduces temporal clustering in extreme precipitation events (defined in relative local terms using 95th percentile of rainfall intensity) and thus alters annual frequency of these events. In landscapes that are susceptible to landslides this clustering and the resultant change in frequency of landslide triggering storms results in a strong association between annual averaged landslide hazard and ENSO. In the subset of these locations where there are many people or assets their exposure to landslide risk follows the same trends.

The key missing component from the current study is a straightforward uncensored analysis of correlation between ENSO and extreme precipitation worldwide before landslide susceptibility or exposure is introduced. This analysis could focus on landslide relevant precipitation events i.e. 7 day accumulated rainfall with a weighted kernel that accounts for drainage (as you currently use). It would also be useful to know whether the steps from correlation between ENSO and extreme precipitation on to landslide hazard then to landslide exposure result in amplification of the correlation or in damping. I strongly recommend that the authors investigate and report that behaviour since I think it would help readers both to understand what is driving the headline results and to trust the headlines themselves.

We thank the reviewer for making this important point. We acknowledge that in the initial version of this study it is to some extent a 'black box' model, in that our findings do not explicitly show the different variability of rainfall, landslide hazard, and landslide exposure with respect to El Nino cycles. In the revised study, we have taken the opportunity to do exactly that, and explore the different relationships between rainfall, hazard, and exposure, to build a more complete picture. Since the primary intention of the paper is to explore the relationship of climatic driver (ENSO) with impact on humans (exposure to landslides), we have placed much of the analysis relating to the in-between steps (rainfall intensity, hazard) into the supplementary material. However, the findings are summarized in the revised main text too. We describe the changes made in more detail below.

Major comments

Because this is primarily a modelling exercise more detail is needed on the model and more space needs to be devoted to explaining / understanding what is driving the model response. In the absence of a comparison with observations this is likely to be the primary way that readers gain trust in your findings.

Why is ENSO so well correlated with landslide exposure in some places? Is it simply due to the changing frequency of large storms (e.g. doubling?); or because large storms become preferentially more frequent in the most exposed or landslide susceptible locations (i.e. Amplification of the effect by the landslide risk generation process); or is the correlation to storm frequency even stronger and the landslide risk generation process damps the signal. Understanding which of these three alternatives is at work or perhaps where they are at work would be a significant contribution and you are ideally placed to make it.

If the dominant driver is changes to the annual frequency of intense rainfall under different ENSO conditions then the literature on ENSO effects on extreme rainfall needs to feature more strongly in the

paper. It is not clear to me from the introduction what and how much is already known about the relationship between ENSO and the frequency, duration and intensity of rainfall events.

We have added a new set of analyses to hopefully address this important point. Essentially, we now compare several additional variables with MVEI to assess the importance of this factor. First, we compare the simple total of rainfall for each month, to see where ENSO has the most effect on total rainfall. Next, we compare the number of days in each month where rainfall exceeds the historical 95th percentile (i.e., the threshold the model considers for 'landslide triggering rainfall'), to assess the effect of ENSO on intense rainfall events. Following on from this, we compare the number of nowcasts (i.e., where intense rainfall aligns with susceptible areas) per month with ENSO. The comparison between ENSO and Nowcasts was actually already something we had done in the first iteration of this study, but we did not discuss it in great detail. This has changed in revision in response to this set of comments. The intention of each of these steps is to build the analysis from simply rainfall to the more nuanced aspects of the model output, in line with what has been suggested by the reviewer.

The full results of this analysis can now be found in the supplemental material – broadly, we find that while the areas where significant ($p < 0.05$) relationships are observed are consistent for each of the parameters (total rainfall, extreme rainfall days, number of nowcasts, population exposure), the level of impact (i.e. the slope) of the relationship differs, with MVEI only affecting nowcasts in mountainous areas, and parts of central Asia likely more responsive to increases in number of extreme rainfall events rather than changes in sum total of rainfall. There is more detail that can be drawn from this analysis, but it is not our intention to focus on the model aspects within this study. We hope that this could provide fruitful questions for future research.

The units 'nowcasts per km² per year' or 'person-nowcasts per km² per year' are difficult to interpret in absolute terms. It would be useful to explain how the reader should interpret these units and how to relate them to hazard in terms of a landslide probability. If this is not possible you should say so explicitly as this would prevent people getting distracted by trying to interpret them. In many cases you are more interested in trends anyway I think it should be possible to normalise the nowcast units so that they are more physically meaningful. I suggest trying something like the percentage oversampling of nowcasts relative to an expected number of nowcasts (which should be feasible given that you use a percentile based approach and a static binary susceptibility layer). Alternatively the approach you take in discussing the Planadas district where ENSO related change in exposure is normalised by the amplitude of the seasonal exposure change would be another good way to go.

We appreciate this point. The units are not necessarily self-explanatory, and so in revising the paper we have tried to improve this by more clearly laying out the definition. Much of this connects to the expanded methodology discussion, which should hopefully alleviate many of the concerns. Critically, we have corrected the earlier use of km² as a unit, since the model output resolution is actually 30x30 arc-seconds (which is only approximately 1 km²). The specific unit of exposure is a multiple of people in a cell by the days exposed to elevated landslide hazard per year (in other words, a nowcast represents a day exposed to elevated landslide hazard). We hope that the new table and revised methods section is now sufficiently clear.

With respect to expressing changes in comparison to seasonal changes in exposure, we feel that this may lead to misinterpretation. In places with little to no seasonal change (i.e., year-round consistency),

even small ENSO-driven changes would look significant. This is essentially what is expressed in Figure 3 – in areas where there is a strong correlation between MVEI and exposure when both are smoothed over 12 months, we will see those areas pop out. However, expressing the exposure as a fraction of seasonal change in Figure 4 would lead to unnecessary attention being drawn to places where ENSO creates significant but small changes, when what we want to emphasise are places where ENSO leads to larger magnitude exposure and impact.

Finally, more detail on two aspects of the modelling is required either in the paper itself and/or in a Sup Info.

- 1) In the methods, a little more detail on how the model is needed but also an indication of its assumptions and its parameters.
- 2) The paper needs a more robust treatment of the model's structural, input and parameter uncertainty and its impact on your findings. This is the primary limitation to an otherwise simple and elegant research design. You discuss uncertainty briefly in the paper but it is not clear what impact it has on your core findings. A full uncertainty analysis would be a huge endeavour. This is one of the reasons I think it would be worth demonstrating the relationships between ENSO and uncensored frequency of intense storms. This would considerably simplify the task of uncertainty estimation (though I would still be keen to know whether the findings were sensitive to the choice of threshold percentile, ARI window length and weighting factor).

As detailed in the specific comments below, we have made major changes to the methodology section to hopefully clarify the parameterization and assumptions within the models.

In terms of the treatment of uncertainties, we agree that a full uncertainty analysis would indeed be a huge endeavour! Furthermore, we feel that it would be somewhat circular to conduct uncertainty analysis of the model outputs when the validation data is so limited. As can be seen in the new section on comparison with country-wide fatality data, even this validation data is at a broader regional scale and lacks enough data points to effectively calibrate or validate at a admin-2 level on a month-to-month basis. (Ironically, this lack of data is an extremely good thing, in that it means fewer fatalities.) As the reviewer has suggested, we have reanalysed the data to look at the relationships between MVEI and total rainfall, number of extreme rainfall days, and number of nowcasts – this shows that while the areas where significant relationships are observed are consistent for each of the parameters, the level of impact (i.e. the slope) of the relationship differs, with MVEI only affecting nowcasts in mountainous areas, and parts of central Asia likely more responsive to increases in number of extreme rainfall events rather than changes in sum total of rainfall. This new analysis hopefully addresses the concerns raised by the reviewer here, and allows us to demonstrate where each aspect of the model is more relevant. See the entirety of the new supplemental material for figures and detail.

Specific comments:

L45-50: The 'however' here does not seem to fit, how does the new rainfall data address the problem of limited inventories preventing modelling?

Good catch, it didn't sound right! Rephrased as follows:

"This precludes consistent modelling of seasonal landslide patterns around the world. Since empirical data can be hard to come by, model-based estimates of rainfall-triggered landsliding can help fill in gaps

(15). Now that close to two decades of consistent global satellite rainfall data at moderate resolution are available...”

L100: the focus on ENSO, but results for NAO and PDO in the supplementary material seems a sensible approach.

Thank you for the vote of confidence! We appreciate that the NAO / PDO analysis may not provide actionable information, but it does help to show that we've done it.

L117-121: More detail is needed on how the model works, specifically:

- L117 “updated version of the NASA Landslide Hazard...” what has changed in this updated version relative to Kirschbaum and Stanley (2018)? The changes themselves need be reported here.

Added several lines of text to explain the changes to the model from the original LHASA model to the current iteration – we have updated the susceptibility map with more recent deforestation data and switched from TRMM rainfall data to IMERG (i.e., expanded the latitude range).

- L119 “7 day accumulated rainfall” is this accumulated using the weighted kernel described in Kirschbaum and Stanley (2018)?

That is correct. We have added text to explain this.

- L120 “historic 95th percentile” what is the timescale over which this percentile is calculated

Added text to explain that this is calculated over the IMERG data period (2001-2019).

- What is the timestep for the model? This is important for the reader to interpret the metrics since they are accumulated nowcasts per year.

Added text to explain that these are daily outputs, aggregated to monthly data.

- L120 “locations of moderate to high landslide susceptibility” are these the LHASA nowcast classes or the Stanley and Kirschbaum susceptibility classes? A sentence or two summarising the susceptibility model would be useful here.

Add text to explain this:

“The susceptibility classes are those defined by Stanley & Kirschbaum 28; moderate or greater susceptibility covers approximately 20% of global land area.”

- L120 what fraction of the globe is covered by the moderate to high landslide susceptibility class? This is important because it provides an indication of how similar landslide hazard would be to simply the number of times per year that the 95th percentile in precipitation is exceeded.

Added in text to explain this (directly derived from original study of Stanley & Kirschbaum), moderate and greater hazard covers ~20% of land surface. This also relates to the later discussion about uncertainties.

- L121 “we consider these ‘nowcasts’ as a proxy for landslide hazard” I think the number of nowcasts per year is your hazard proxy, otherwise the proxy is binary.

Added text to clarify that this is a daily output, aggregated to a monthly output.

- Do you still use an absolute ARI threshold of 6.6 mm?

Yes, added detail to show this. Thanks for flagging.

L120: A binary landslide susceptibility classification of landslides i.e they are possible or not seems a very coarse classification. It may be that it doesn't matter for your results because susceptibility is static in time but I think you need to explain more clearly that you assume this binary susceptibility, justify it, and ideally also test it (though I recognise the latter is probably out of scope).

Added text to explain this:

"While this binary classification of hazard is coarse on a daily basis, aggregated on monthly time-scales it provides a more nuanced perspective on changes in hazard."

L125: How do you know that this variability is not introducing bias? Your approach to focus on population density with the remainder in SupInfo mitigates the impact of the potential biases in road and osm data. Perhaps you could re-phrase when you introduce these datasets to make your focus on population density clearer. It may avoid people getting distracted as I did. This is only a suggestion though, fine to leave as is if you disagree.

It is entirely possible that OSM incompleteness could introduce bias. We should have given more space to this in the first version of the text, but we can focus on ENSO-induced changes by solely looking at the fraction of exposed elements, rather than absolute numbers. We've added a sentence to point this out:

"We can account for changes in completeness by normalizing the estimates of infrastructure exposure by the total infrastructural elements – i.e., the exposed fraction."

L145: "...allows a globally homogenous estimate of landslide hazard and exposure, rather than relying on recorded landslide events where spatial and temporal biases in reporting may be significant." This is true, however it also introduces a dependence of the findings on the model and parameter uncertainty. What is the sensitivity of your findings to:

- the choice of 95th percentile in ARI for normalising
- ARI parameters: 7 day window and weighting factor of -2 (calibrated from 2007-13 landslide data but I guess that other values gave only slightly worse fits).
- The absolute ARI threshold of 6.6 mm if this is still used
- The parameters of the fuzzy overlay model that defines landslide susceptibility and therefore whether or not rainfall above 95th percentile will trigger a nowcast.
- The choice to nowcast only for areas of moderate to high landslide hazard

A full uncertainty analysis here is a huge endeavour. If you can demonstrate the relationships between ENSO and uncensored rainfall accumulation it would indicate that your results are retained in this simpler case.

These are all really good questions! We feel that the answer is quite nuanced, and there are different ways to address it. We agree that a full uncertainty analysis is a major effort, and in fact would represent a very different study. More critically, to get to a testing of different uncertainties it would be essential to have a set of robust observational data to compare the outputs against. In the revised study here, we have incorporated the fatality dataset of Froude and Petley (2018) as a useful first order comparison, but we also feel that a full sensitivity test is beyond the scope of this study. We hope that by splitting the

relationships between rainfall, extreme rainfall, nowcasts and exposure, we have in some way informed which aspects of the model are most strongly influenced by ENSO. We have added text to explain this point:

“To establish whether the model results are a useful approximation of exposure, we have compared our results with fatality dataset derived from the dataset of Froude & Petley (2018). We have not run sensitivity tests of our model for the various input parameters, as they were primarily determined in other studies that developed the LHASA model. Further sensitivity testing of our model remains an important topic for future research.”

L180: “(nowcasts per person)” I think this should be person-nowcasts per km² per year.

We have adjusted this since in revision we have corrected prior unit errors (the nowcasts are at 30-arc second grid cell resolution, which is not exactly 1km²). This now reads: “The upper-most series of plots show the sum total of Popexp for each month in the administrative district. This is the total number of people exposed to elevated landslide hazard in that district for each month to show seasonality”

Thanks for flagging!

L199: “(see Figure 2) is greater than 0.2” why use Rsquared rather than a p value? Why choose this threshold? It seems quite low (20% of variance in annual nowcast frequency is explained by ENSO) it would be useful if you could explain why Rsquared of 0.2 is perhaps better than our instincts suggest in this situation.

A great point. We’ve used this as impetus to drop the R-squared values and re-do the analysis to calculate the relevant p-values. This is now provided in data tables, and shown in the relevant figures and discussed in the text.

L207: “significant relationships” this would be worth supporting with a test of the significance.

See above point – we’ve re-done the analysis to get the p-values. Thanks for the push!

L216: “we show the magnitude of the change driven by a unit shift in MVEI index” How is this calculated? Is it the gradient of the linear least squares regression? How did you decide that a linear function was the most appropriate? The relationship looks pretty linear in Fig 2 but did you try other functional forms? It would certainly be worth remarking on the linearity in Fig 2 and confirming that this is a general property of the data so that this later step is more comfortable for the reader.

Thanks for flagging the lack of clarity! We’ve added text to this section to explain it, and the new supplementary material also provides detail. New text:

“We calculate this based on linear least-squared regression. We choose to fit a linear relationship to all of the districts to ensure consistency, and because it seems to describe the relationship between MVEI and model outputs in areas where ENSO is known to be important (e.g., Figure 2).”

L225 Fig 4: Normalising by population makes sense but population was already in the numerator so this

could be simply change in nowcasts per km² per year with changing ENSO. I wonder if it would be helpful to express this as a fraction of the nowcasts per km² per year or a percentage. This may cause problems for the linear regression but it would seem worth trying because the existing units are difficult to interpret in absolute terms.

This is an interesting point! Putting the total population in the denominator doesn't quite get back to nowcasts, since it's not always the same population exposed during each nowcast. As can be seen by comparing supplementary figures S7 and S8, it's not identical – there are parts of the US where fractional population exposure changes less than nowcast changes with respect to MVEI, indicating perhaps that more people live in less exposed areas. However, this is relatively speculative, so we refrain from drawing these conclusions in the full text. Suffice to say, it's not exactly the same.

In terms of displaying information as a fraction of seasonal change, as discussed above, we feel that expressing the data as such could lead to misinterpretation. We want to avoid drawing attention to places where seasonal changes are small, where even small ENSO-driven shifts could seem important.

L231: “change in exposure can be nearly as large as the seasonal shifts in exposure...” This is a very good way to express the change, could you do this more generally (e.g. for the colorscale on Fig4)?

See above comments. We feel that this is a useful way to contextualize some of the data, but could lead to misinterpretation if used systematically.

L244: “unclear what is driving the effects seen in our model results” Because this is a modelling exercise this seems a strange statement. Your ability to explain these unexpected results is central to building trust in the model, particularly in the absence of a comparison with observations. Can you establish whether the effect is retained when susceptibility and exposure are not included? Can you identify whether the effect is related to the percentile threshold (e.g. very low) in this region or to the changing shape of the ARI distribution or both.

Thanks for highlighting this – it was not very scientific in the first instance! Now that we have run the analysis to test the p-values and slope of the relationships between MVEI and total rainfall, extreme rainfall, and nowcasts, we are able to more carefully qualify this statement. We have extensively revised this section to reflect the new analysis, as well as explain the Central Asian trends:

“However, other studies have shown that increases in extreme rainfall intensity are less important than increases in frequency of extreme events⁹, meaning that changes in intensity of extreme rainfall events is may not be a significant systematic error for our model outputs. However, we can compare locations where MVEI is linked to changes in total rainfall, and where it drives changes in extreme rainfall, to assess whether this leads to discrepancies. In addition to assessing the slope of the relationship between population exposure and MVEI, we have also assessed the slope of the relationship between MVEI and total monthly rainfall, number of days per month where rainfall exceeds the 95th percentile, and number of days per month in which a nowcast is triggered for each admin-2 level district. These results are shown in supplementary figures S5, S6, and S7 respectively. We find that MVEI is linked to changes in extreme rainfall in a range of places; in addition to the United States and parts of South America, increases in extreme rainfall are observed in large parts of Central Asia, particularly in Tajikistan and Kyrgyzstan. This is in line with other studies of ENSO-induced rainfall changes^{39,40}. While negligible changes in total rainfall are observed in Central Asia (Fig. 1), it is likely the change in frequency of locally extreme rainfall

that drives the large magnitude changes in exposure seen in our model results (Figure 4)."

L251: "if ENSO makes already-intense rain storms even more intense" I struggled to understand what an already intense rain storm was. Perhaps a storm that would have been intense anyway or a storm in a location where intense rain storms are common? I think I now understand: because you record the frequency of storms above a threshold intensity you cannot account for increased intensity of storms above this threshold. I wonder if you could clarify this in the text.

Thank you for highlighting this. We have revised the relevant sentence:

"Since LHASA model outputs depend on rainfall exceeding the historical 95th percentile, the model is only sensitive to ENSO-induced changes in the frequency of those extreme events, but not to an increase in intensity of extreme events."

L307: more could be said in the conclusions about which regions experience the strongest landslide exposure changes under ENSO, which regions experience positive and negative correlation and how those changes relate to the shorter term seasonal variability. These are all interesting and useful findings from earlier in the paper and it would be very helpful to summarise them here.

We have taken the opportunity to reiterate the key findings of our study, and highlight the key regions where ENSO affects exposure to landslides based on our model results. New text:

"Our model results indicate that La Nina conditions lead to greater landslide exposure in a diverse range of settings, with the largest increases seen in Colombia, Ecuador, Central America and the Caribbean, as well as Indonesia, Papua New Guinea and the Philippines. While El Nino conditions seem to lead to fewer areas with increases in exposure in comparison, there are still increases in Central Asia, parts of Eastern China, and Mexico. We suggest that future validation of these results with comparison to landslide event data will be important, and supports a need for comprehensive landslide inventories."

Finally, this is a very good paper that makes a valuable contribution and I would expect that the authors should be able to address all these comments fairly easily. I hope that they find the comments helpful and enable them to improve an already very good paper.

We thank the reviewer for the supportive comments, and the thoughtful critique that we feel has served to improve our study.

Reviewer #2 (Remarks to the Author):

Review

Global connections between El Nino and landslide impacts

General Comments

The described method of obtaining a globally homogenous estimate of landslide hazard and linking it with ENSO impacts is interesting and can be useful. I recommend acceptance after the authors have

answered the comments, and met the requests/suggestions detailed in the following.

It would be informative to report if there has been any attempt at validating its results with real recorded landslide events, and what were the results of this assessment (specific comments #8 and 30). There is an issue regarding statistical significance (comments #16 and 20). Besides, there are some changes needed in the text to make it clearer and more informative. They are also detailed in the specific comments below.

I think the authors will be able to easily meet the requests, and that the paper will give a good contribution to different but connected fields.

We thank reviewer 2 for their detailed and thoughtful comments. The reviewer's expertise was greatly appreciated, and the comments provided have significantly improved the paper in revision. We appreciate the support for the conceptual advance defined in the paper. We have addressed each of the comments provided below in detail.

Specific Comments

1. Pages 1, line 28: It is mentioned that the incidence of widespread landsliding can vary on seasonal to multi-annual timescales and a reference to the influence of El Niño on extreme rainfall events in South America is added. It is interesting to comment that the decadal/interdecadal modulation of ENSO events provided by IPO (or PDO, which is related to IPO) does also provide an interdecadal modulation of extreme rainfall events. A clear example is available in South America, as shown in:

Grimm, A. M., N. C. Laureanti, R. B. Rodakovski and C. B. Gama, 2016: Interdecadal variability and extreme precipitation events in South America during the monsoon season. *Climate Research*, v. 68, n. 2-3, p. 277-294. DOI: 10.3354/cr01375

In this reference, the South American main modes of precipitation interdecadal variability (which are influenced by IPO and PDO) are shown to affect significantly the frequency of extreme events in two of the regions in which the landslides are most influenced by ENSO, according to this manuscript, Colombia and South Brazil.

We thank the reviewer for highlighting this, as it adds to the information in the introduction. We have amended the introduction accordingly to incorporate this information and cite the study the reviewer mentions:

"Interdecadal variability of extreme rainfall events in South America is also modulated by the Interdecadal Pacific Oscillation"

2. Page 2, lines 51-59: The reference above shows why shifts in exposure are observed with respect to changes in PDO state, since this oscillation, as ENSO, affects the frequency of extreme events even more significantly than the monthly or seasonal rainfall totals. It proves that this decadal variability mode affects the frequency of extreme events that can produce landslides.

We agree with the reviewer here that PDO is likely to be important in setting the frequency of extreme rainfall events. However, as discussed in our response to reviewer 3, below, we feel that the timescale over which we have assessed the rainfall is too short to effectively characterize the impact of long-period events like PDO and NAO. As discussed in our response to reviewer 3, we therefore do not draw conclusions about the nature of the relationship between PDO and NAO, even if one exists.

3. Page 3, lines 83-84: With 'The sign of the ENSO index changes on the order of 1-5 years' are you suggesting that ENSO has periods from 1 to 5 years? Usually the shortest period attributed to ENSO is 2 years. The reference cited for this statement (16) says that the power spectrum of MEI (an ENSO index) 'shows a concentration of energy between 60 and 20 months', and evidenced two harmonics, of 26 and 50 months.

Thank you for flagging – we have changed this to the more accurate '2-5 years'.

4. Page 3, lines 88-93: There are also significant effects of PDO on rainfall and its extreme events over South America, in regions that appear in this manuscript as having high landslide exposure and high ENSO-related landsliding exposure, as, for instance, Colombia and South Brazil. See the reference Grimm et al. (2016), mentioned above (comments # 1 and 2).

Added a sentence to this section to mention the change in extreme rainfall and cite the relevant reference the reviewer mentions.

5. Page 5, line 119: It seems that 'where' should be replaced with 'when'.

Changed accordingly.

6. Page 5, line 123: It is convenient to inform what is GPW. Therefore, in line 123, please write '...are derived from the Gridded Population of the World, Version 4 (GPWv4), roads from...'

Changed accordingly.

7. Page 5, line 127: Some readers may not notice what GRIP and OSM mean. Therefore, please write in line 124: '...from the Global Roads Inventory Project (GRIP), and infrastructure from OpenStreetMap (OSM)', and use this acronym also in line 125.

Added acronym descriptions where they are first used. Thank you for flagging this!

8. Page 6, lines 143-145: Although I agree with your method of obtaining a globally homogenous estimate of landslide hazard, I and probably other readers would be curious to know how its results compare to real recorded landslide events. I understand that data are not homogeneous, sometimes not reliable (biases in time and space), but has there been any attempt at validating the results of this method, at least for real recorded landslide events? What were the results of the assessment? Please mention them and summarize them.

This is a really important point, and we thank the reviewer for raising it. This has pushed us to seek out validation data for our model results. We have used the Global Fatal Landslide Database of Froude and Petley (2018) which contains data on fatalities associated with landslides around the world from 2004-2017. Fatalities are more closely tied to exposure than simply landslide occurrence (which means that the NASA Global Landslide Database is slightly less suitable here), and moreover are reported more consistently than other landslide events. We have broken down the landslides where fatalities occur by country and month, and compared the rate at which fatality-inducing landslides occur for specific intervals of MVEI values with the proportional number of months in each MVEI index interval for the period in question. This provides an approximation of 'how much more likely are fatality-inducing landslides to occur during this MVEI interval than another', allowing us to assess the extent to which La

Nina or El Nino leads to a higher rate of landsliding. There are not enough landslides in the fatality dataset to assess at an admin-2 level the changes in landslide rate, but we can look at specific countries with a greater number of landslides where we would expect changes in landslide exposure based on our model outputs. We find that in several critical countries (e.g., Colombia, Ecuador, Chile, Vietnam) there is a higher incidence of landslides than expected for La Nina conditions, as predicted by the model outputs. Less clear relationships are observed in the Philippines and Indonesia, but in broad, qualitative terms the model outputs are validated by observational data. There are caveats to this validation assessment, including the lack of vulnerability assessment in our model, but this remains a useful test. We have added extensive text to discuss the validation data and present these results. To save space, we direct the reviewer to read the revised version of the paper, rather than replicate the new text below.

9. Page 6, lines 161-163: Why mention only the northern hemisphere monsoon season? There are also southern hemisphere monsoon systems, such as Australia, southern Africa and South America. See <http://www.clivar.org/clivar-panels/monsoons> or the Fig. 1 of a very recent paper on the global monsoons:

Wang, B. et al., 2020: Monsoon Climate Change Assessment. Bulletin of the American Meteorological Society. <https://doi.org/10.1175/BAMS-D-19-0335.1>

The rainfall annual cycle shows a very marked difference between winter and summer in these regions. Therefore, I would suggest rewriting these lines as: 'As demonstrated by other studies, the seasonal variability of landslides is primarily set by the dominant rainfall regimes, such as the monsoon regimes or the tropical cyclone seasons around the world.'

Thanks for flagging this. We have changed it to the suggested phrasing.

10. Page 6, lines 166-167: When does the 12-month moving average of exposure with a window starting 12 months prior to the month in question end? When the window ends 12 months after the month in question?

Thank you for the opportunity to clarify! Added the following sentence: *"The moving window starts 12 months prior to the month in question, and includes that month"*

11. Page 7, line 186: I suggest replacing '...which removes annual trends...' with '...which removes the annual precipitation cycle...'

Thank you for the suggestion – it's much better! We have changed it accordingly.

12. Page 7, line 187: I suggest replacing '...in the strength of annual impacts...' with '...in the strength of seasonal impacts'.

Changed accordingly.

13. Page 7, lines 189-192: I suggest replacing '...the differences in seasonal variability, too; Kinnaur has a...' with '...the differences in seasonal variability: Kinnaur has a...'.

Changed accordingly.

14. Page 8, line 195: I suggest replacing 'In the handful of locations illustrated in...' with 'In the locations illustrated in...'

Changed accordingly.

15. Page 8, line 196: I suggest replacing 'The next step is to map these trends globally,...' with 'The next step is to map this influence globally,...'. ENSO does not produce trends, but variations.

Changed accordingly – thank you!

16. Page 8, lines 198-201: Why are you using this threshold of 0.2? I think the more correct procedure here would be to calculate the threshold for which R-squared is significantly different from zero with a chosen level of significant (for instance, 0.05 or even 0.10). It might even result to be 0.2, but you should have an acceptable reason for choosing 0.2. Or at least inform what level of significance corresponds to 0.2.

Thank you to both reviewer 2 and 1 for pointing out the deficiency of the R-squared assessment in the first version of the paper. In revising our study, we have instead calculated the p-values as a measure of significance of the relationship between MVEI and population exposure (and additionally the extreme rainfall, nowcasts, and total rainfall).

17. Page 8, line 205: It seems that '...and population exposure exceeds 0.2.' should be '...and population exposure exceeding 0.2.'

Since we have now changed the analysis to determine the p-values, rather than R-squared values, this text is no longer in the revised study.

18. Page 8, lines 207-211: In the first sentence, it is said that significant relationships with ENSO are observed in Latin America, which is true. Latin America is a group of countries in Americas where the predominant language originated with the Latin language, such as Spanish, Portuguese, and French. Brazil is one of these countries. But then, in the second sentence, it is said: 'There are a number of other locations, such as in central Brazil, ...', as if central Brazil were not in Latin America. I suggest replacing this sentence with 'There are a number of other locations, such as in Central Asia, and Southern Africa, where varying degrees of...'

Thank you for highlighting this, and the potential for confusion. It was not our intention to suggest Brazil was not part of Latin America. We have changed the text to the suggested wording.

19. Page 8, line 220: I suggest replacing '...we show two of the key locations – SE Asia, and Latin American and the Caribbean.' with '...we show two of the key locations – SE Asia, and the northern part of Latin America.'

Changed accordingly.

20. Page 9, line 225: Significant at what level of significance?

Thanks to reviewer 1 and 2 for highlighting the need to quantify the level of significance. We have run the analysis, and now Fig 4 shows areas with p-values lower than 0.05.

21. Pages 8 and 9, Figs. 3 and 4: The comparison of Figs. 3 and 4 is intriguing. In Fig. 3 there are extensive areas with great R-squared values (around 0.5-0.6) of the relationship between Multivariate ENSO Index and population exposure in central Amazon, which most probably are significantly different from zero. However, in Fig. 4, at the same location (lower right corner of the upper right panel, for northern Latin America) there are almost no areas with significant ENSO - population exposure relationship. What am I missing?

This is something that all three reviewers seem to have had trouble with, which definitely means we lacked clarity in the first version of the text! The key point is that there are areas where significant changes can be observed (i.e., MVEI and population exposure are correlated) but where the changes do not lead to large changes in population exposure with respect to the overall population. We can imagine a district with a large population where only a handful are impacted by ENSO driven changes – the relationship between MVEI and change in exposure could be significant, but the effect on fractional exposure of the population to landslides would be small. We have added new text which we hope clarifies this point:

“A significant relationship between MVEI and the model outputs does not necessarily indicate that MVEI creates large changes. Simply put, even small changes can have significant relationships with MVEI if other climatological trends do not occlude the relationship. However, we are interested in the areas where ENSO causes the largest changes in exposure. As such, we must look at the slope of the relationship between MVEI and exposure – see, for example, the middle set of figures in Fig 2.

In Figure 4, we show the magnitude of the change driven by a unit shift in MVEI index. We calculate this based on linear least-squared regression. We choose to fit a linear relationship to all of the districts to ensure consistency, and because it seems to describe the relationship between MVEI and model outputs in areas where ENSO is known to be important (e.g., Figure 2). Figure 4 therefore shows the areas where we model a statistically significant impact of MVEI that leads to major changes in the fraction of population exposed to landslides. By looking at the fraction of the population exposed, we can consistently compare areas with varying population density.”

22. Page 9, line 232: I suggest replacing ‘...as large as the seasonal shifts in exposure -...’ with ‘...as large as the seasonal variations in exposure -...’.

Changed accordingly.

23. Page 9, lines 231-233: How did you calculate this? Please show with numbers that for Planadas (Colombia) the difference between Pop(exp) for La Niña and El Niño, obtained from Fig. 4, is comparable to the difference for July and April, obtained from Fig. 2. It would be a real quantitative example.

Thank you for highlighting the lack of clarity. We have added the following text to help clarify:

“by comparing the top and bottom figures for Planadas in Fig 2, we can see that ENSO-driven changes are on the order of 60000 change in Popexp, with seasonal variability on the order of 100000 change in Popexp”

24. Page 10, lines 240-244: There are articles that show the ENSO effect on Central Asia. Below are two of them, but you can find other ones:

Mariotti, A., 2007: How ENSO impacts precipitation in southwest central Asia, *Geophys. Res. Lett.*, 34,

L16706, doi:10.1029/2007GL030078.

Jialin Lin, J. and T. Qian, 2019: A New Picture of the Global Impacts of El Niño-Southern Oscillation. Scientific Reports, 9:17543 | <https://doi.org/10.1038/s41598-019-54090-5>

A great point! We are grateful for the reviewer's knowledge of the relevant literature here. We have cited the studies discussed, and incorporated them into a revised discussion of the impact of extreme rainfall in central Asia.

25. Page 10, lines 250-260: I do not agree with your argument in lines 250-254, and thus I think you can remove this uncertainty. There are two possible impacts of ENSO on extreme events. It can enhance the intensity of extreme events or the frequency of extreme events or both. The most common impact is of the second type, on the frequency (number in a time period) of extreme events. You seem to be considering that the first type is the most important, and this is not true, since the areas undergoing the first type of impact are more scattered than those undergoing the second type. This is shown in Grimm and Tedeschi (2009) for South America. By the way, the reference that was cited regarding USA (33, Gershunov and Barnett 1998) only shows the influence of ENSO on the frequency of extreme events in USA. If El Niño (or La Niña) increases the number of extreme events (or the number of events exceeding the 95th percentile), as in several regions in South America and USA, obviously it increases the exposure to landsliding. Therefore, this kind of uncertainty would be rarer, and you could remove it. Yet the other sources of uncertainty (lines 254-260) are more concrete: the limited satellite rainfall record and the limited accuracy of satellites precipitation estimates for extreme rainfall.

Thank you for the detailed comment, and the information. This is helpful since it provides some supporting evidence for the validity of our model outputs. We have changed the text in the paragraph discussed as follows:

“There remain some uncertainties and challenges in this type of analysis. Since LHASA model outputs depend on rainfall exceeding the historical 95th percentile, the model is only sensitive to ENSO-induced changes in the frequency of those extreme events, but not to an increase in intensity of extreme events. Studies have shown ENSO-driven changes in rainfall extreme values in South America 5 and the United States 36; as such, our results may miss some impacts where this is the case. However, other studies have shown that increases in extreme rainfall intensity are less important than increases in frequency of extreme events 9, meaning that changes in intensity of extreme rainfall events is may not be a significant systematic error for our model outputs. The limited satellite rainfall record also prevents us from including the 1997-1998 El Niño event, which was amongst the largest on record. The accuracy of satellites precipitation estimates for extreme rainfall is known to be limited in some scenarios including orographically-enhanced rainfall or short-duration, high-intensity events 37 or mixed rain and snow events that often dominate winter to spring landslide activity in the U.S. northwest coast. As such there may be limitations in resolving rainfall effects, potentially affecting the representation of ENSO-induced changes in the overall rainfall difference maps (Figure 1). We suggest this is an important topic for future research.”

26. Page10, lines 271-273: It seems that there is a full stop in the middle of this sentence, in line 273 (between ‘ENSO’ and ‘this’). Remove it, or withdraw the word ‘if’ in line 272.

Changed to remove full stop and capture original meaning. Thank you for catching this!

27. Page 10, lines 271-277: What do you mean by ‘peak in local rainfall lags behind the peak in ENSO’ or ‘lead-lag relationships between ENSO and rainfall’? Is it the time lag between the largest absolute MVEI index and the largest relative impact on rainfall (percentage variation of climatological rainfall)? This lag is pretty frequent, since the teleconnections responsible for the ENSO impact depend on the basic state of the atmosphere in which they propagate. For instance, in South Brazil, where there is a relatively strong relationship ENSO-landslide exposure in your Figs. 3 and 4, the maximum relative impact happens in Oct-Nov and not in the typical height of ENSO during late Northern Hemisphere Fall/early Winter.

The reviewer is correct that these are the kind of lead-lag relationships to which we are referring. Our model does not account for these; we suggest that they would represent a fruitful avenue for future research. We have rephrased the sentence in question accordingly:

“Our model results do not account for lead-lag relationships between the MVEI values and the resulting changes in rainfall patterns.”

28. Page 11, line 279: I would not use this word ‘trend’, but ‘variation’. Trend has a connotation of continuous increase or decrease, which is not the case of ENSO impacts.

Changed accordingly.

29. Page 11, lines 279-284: The lack of signal or the weak signal showing link between landslides and PDO (or IPO, since they are related) is most possibly due to the short timespan (2001-2018) of this study preventing detection of the full rainfall variability due to interdecadal PDO (or IPO). The study by Grimm et al. (2016, mentioned in comments # 1 and 2) shows clearly that there is a very significant impact on the frequency of extreme events in several regions of South America, and therefore, on the landslide exposure.

This is a good point, and is useful to compare with the comments from Reviewer 3 about the use of PDO and NAO data here. We have added more text to the conclusion to hopefully capture the state of knowledge regarding long-period teleconnections and rainfall driven landslides:

“While there is significant variability associated with ENSO in a number of key locations, similar effects are not clearly observed for NAO (despite some studies showing links between landslides and NAO index 40, and others demonstrating links between extreme rainfall and PDO 9), and only to a minor degree in association with PDO. We suggest that the timespan 2001-2018 is too short to capture the full rainfall variability due to multi-decadal PDO and NAO cycles; we suggest future studies may be able to address this more fully. It is not possible to draw any conclusions about the impact of NAO or PDO on landslide exposure here. All data for ENSO, NAO and PDO exposure is available in the supplementary material.”

30. Page 11, lines 291-292: I think that some validation results for the model, even if not related with ENSO, should be mentioned, as said in comment #8, since before using the model to link landslide exposure to climate variations it is important to know its reliability in reproducing the occurrence of landslides.

A great point! We have used this and the earlier comment as motivation to compare our model outputs with real fatality data. This is described more fully above.

Signed: Alice M. Grimm

Reviewer #3 (Remarks to the Author):

The topic addressed by this paper is of some interest. It has long been argued that changes in rainfall pattern associated with a large El Nino event might well cause changes in landslide occurrence. This hypothesis largely emerged in the aftermath of the 1997-98 El Nino event, which caused large-scale landsliding in the Americas. However, there is little validation of this idea, and indeed some evidence that La Nina conditions might be more important than El Nino on a global scale.

To that end this paper is welcome, but unfortunately I do not believe that it meets the standard needed for publication.

We thank the reviewer for taking the time to provide valuable insight and comments. We appreciate that the reviewer acknowledges the interest of the subject matter. We found that many of the comments provided by the reviewer were of value to improve our study, but in some cases we respectfully disagree some of the critical points. We feel that some of the reviewer's concerns stem from a misunderstanding of the modelling approach taken, since we did not explain the model in sufficient detail in the first iteration. Ultimately, it is incumbent upon us as authors to make the method as clear as possible, so we have used this misunderstanding as impetus to revise our description to ensure clarity in the revised study. We discuss each comment in turn below.

I would like to highlight the following particular concerns:

1. The paper makes some reference to NAO and PDO in the text, but more importantly in the supplementary information. It is simply not possible to examine the impact of decades long cycles using a dataset that does not completely cover two decades. The result is an analysis that is spurious and should be removed.

We are somewhat surprised to see this comment. We have tried to make clear in the text that our analyses of PDO and NAO cycles reveals exactly what the reviewer asserts – that the length of the rainfall dataset is insufficient to assess changes due to NAO and PDO. We feel that if we removed this discussion, a reasonable reader could query whether we tested these other climatic systems. Generally, we feel it is good scientific conduct to publish findings and data even if they don't reveal connections between systems. While the reviewer suggests that the analysis of NAO and PDO cycles is 'spurious', we feel that it still tests a hypothesis – namely, that the rainfall dataset is not of sufficient length to assess NAO and PDO cycles. In revision, we have tried to make it clear that our analysis does not support any conclusion with regards NAO / PDO and landsliding – but we stand by our inclusion of these results since we feel it does not detract from the key message, and in fact provides useful supporting information. We note that reviewer 1 explicitly notes that "the focus on ENSO, but results for NAO and PDO in the supplementary material seems a sensible approach."

We have also added new text to the introduction section:

"Although we observe some changes in exposure with respect to PDO and NAO, the long timeline of NAO and PDO cycles are not captured by the 18-year rainfall record, meaning no conclusions can be drawn as

to the effect of PDO and NAO on landslide exposure. We include these results in supplementary material to ensure reporting of all findings.”

And the conclusion section:

“While there is significant variability associated with ENSO in a number of key locations, similar effects are not clearly observed for NAO (despite some studies showing links between landslides and NAO index 40, and others demonstrating links between extreme rainfall and PDO 9), and only to a minor degree in association with PDO. We suggest that the timespan 2001-2018 is too short to capture the full rainfall variability due to multi-decadal PDO and NAO cycles; we suggest future studies may be able to address this more fully. It is not possible to draw any conclusions about the impact of NAO or PDO on landslide exposure here. All data for ENSO, NAO and PDO exposure is available in the supplementary material.”

2. The authors fail to recognise that there is not a single type of El Nino (or La Nina), and that rainfall patterns vary considerably between these events. The authors should consider Kug et al (2009) (<https://doi.org/10.1175/2008JCLI2624.1>) for example. It is likely that these different types of El Nino conditions generate very different rainfall patterns. The paper should consider this carefully - thus, comparing a single peak El Nino and a single peak La Nina might have very little validity.

The reviewer raises an important point – and we wholeheartedly agree that every El Nino is not alike. This was the rationale for our choice of the Multivariate ENSO Index version 2 as the proxy we used to assess ENSO. As discussed by Wolter and Timlin in their 2011 study, MVEI v2 accounts for both forms of ENSO within the index:

“Since the MEI.ext SST loadings are high for much of the equatorial cold tongue, we believe that both types of ENSO events project reasonably well on the MEI, reducing the risk of letting any event slip by even if it were to remain confined to the eastern or western portion of that region.” [Wolter & Timlin 2011].

Naturally, using this index as a way to look for relationships between ENSO and landslide exposure means we are combining both cold-tongue and warm-pool ENSO events in our analyses. We are not drawing any distinction between the two types in our analysis, for exactly the reason the reviewer points out – there are only a handful of events between 2001 and 2018 of either type, meaning that drawing specific assertions about either type will be subject to the limited data. In revision, we have highlighted the importance of considering the variability in ENSO type, and made changes to the discussion section to better explain how these results should be contextualized against the MVEI. Given that (to our knowledge) this is the first analysis to demonstrate global relationships between ENSO and rainfall induced landslide exposure, we also suggest that defining more nuanced relationships with the different types of ENSO events is an important topic for further study.

New introductory text:

“It is important to note that the ENSO system exhibits significant variability, with some studies showing that in addition to the conventional ‘Cold-Tongue’ El Nino, an atypical form may exist, variously referred to as ‘Warm-Pool’ or ‘Central Pacific’ El Nino events (19). While these two types of event are both well characterized by the MVEI product (16), they may lead to differences in precipitation variability (19). In this study, we do not differentiate between different forms of El Nino event, partly because there are only a handful of ENSO cycles in the period of observation, limiting the availability of data. In addition,

since the existence of this non-standard El Nino form remains debated (20) and no widely accepted index for its strength exists, we suggest that our initial findings can lay the foundations for future work to explore the differences in landslide impact due to El Nino variability when more data is available."

3. I have a concern that the analysis does not consider the types of precipitation that generates landslides. There are two types (to generalise vastly) - long duration moderate intensity - the "wet winter" scenario - and short duration, high intensity - the "cloudburst" scenario. In general, the majority of high cost landslides occur in the latter scenario, but this is barely captured by the dataset considered here. Indeed, it is in monsoon conditions entirely possible to have a drier than usual month with a record breaking rainfall event.

We thank the reviewer for the comment. Unfortunately, we failed to explain the model in sufficient detail initially to provide the clarity needed to avert this. We think that the reviewer's concern stems from a misunderstanding of the model (which in the end still requires revisions to the text to improve clarity). The data is presented as a month-to-month dataset of landslide exposure vs El Nino index, but the monthly data is not the smallest time-step used in this analysis. The landslide exposure model incorporates daily rainfall intensity data from GPM IMERG to estimate a daily landslide exposure estimate. However, since MVEI values are not provided day-to-day, we only have a monthly ENSO proxy to compare with exposure. So we take the sum of exposure from each day in the month in question and compare it with the MVEI value, with the 12-month moving average used to smooth out the seasonality. Fundamentally the data is based upon daily rainfall data (technically the landslide hazard model includes a weighted total of 7-day antecedent rainfall to account for existing saturation) which does a good job of accounting for those cloudburst scenarios. If anything, we may miss the 'wet winter' scenario since the hazard model does not consider rainfall that occurred more than 7 days prior to the day under analysis. We have highlighted this more clearly in revision.

To address this, we have made changes to the methodology section to make it much more clear how we have constructed our analysis. We thank the reviewer for highlighting it, since it has allowed us to strengthen the communication of our findings.

These are fundamental points that undermine the premise of this paper, rendering it unpublishable in my view.

We are hopeful that based on the responses provided here and the changes made to the study, the reviewer will reconsider this assessment.

I will pick up some more detailed points as well:

This should be the South Asian summer monsoon (not India). Tropical cyclone seasonality - again, this is related to TCs in the NWP, the Indian Ocean and the Atlantic, primarily

Line 51: I do not believe that the PDO and NAO results are statistically valid (see above), so this should be removed.

We have tried in revision to clarify that drawing conclusions about PDO and NAO is not statistically valid, but we prefer to include these findings to avoid other researchers doing the same analysis and finding the same. Better to include negative results for the benefit of the scientific community.

Line 77: this is a huge assumption that needs justifying. El Nino may drive changes in, for example, vegetation that may be really important too. Does it change vulnerability of population as well? This is not considered.

This is an important point, and one we thank the reviewer for making. One of the main reasons we are assessing exposure to landslides rather than risk is that globally consistent datasets on population vulnerability are hard to come by in the first place, and we are not aware of any global assessment of how ENSO affects population vulnerability. It may be really significant, based on local studies. We cannot draw strong conclusions either way, unfortunately. In revision we have included real observations of fatalities, and vulnerability is important to consider when contrasting exposure with impact (i.e., fatalities); we have made that point explicitly, and emphasized that ENSO changes might lead to correlated changes in vulnerability.

In terms of vegetation changes, the reviewer is also correct to raise this important issue. The landslide susceptibility model that underlies our exposure analysis incorporates the Hansen et al. (2013) Landsat-based forest loss dataset to account for areas where deforestation has occurred; specifically, we use the 2018 data. This dataset considers anthropogenic and natural changes in forest cover. However, as it is an annual dataset, it is not clear that we can use it to determine consistent changes in vegetation index as a function of ENSO, which may cause changes on a non-annual basis. We have not attempted to estimate changes in forest cover linked to ENSO, which means we do not account for this effect. This may have some impact on our findings, and we have amended our discussion section to explain this:

“In addition, while we make the simplifying assumption that changes in ENSO impacts are solely related to rainfall, it is possible that ENSO may affect other landslide-relevant factors, including land cover (Kondo et al. 2018). We suggest that further analysis of the impact of ENSO-driven land cover changes on landslide exposure represent valuable future research topics once homogenous global month-to-month land cover datasets become available.”

Line 125: It is not clear to me whether the exposure data was dynamic or static - in other words, for each point in the raster, was a single value used for the entire 18 years, or was this updated with time? There have been huge changes in some of these values (e.g, road density) in this period. If a single value was used, which year was considered?

We thank the reviewer for raising this point. It is clear that our initial description of the input data and model was not sufficiently clear or detailed to provide the required information. We use static data on exposure from the 2015 GPWv4 population dataset, 2015 Global Roads Project data, and from mid-2018 for openstreetmap infrastructure assessment. Since our modelling study seeks to clarify the effect of ENSO on exposure, adding dynamic population or infrastructure data adds an additional element that is not controlled for. This is an additional source of variability when comparing model outputs with real data on fatalities. However, while semi-dynamic datasets on population are becoming available, the same is not true for roads, and while openstreetmap is dynamically updated, it is not clear that we can determine which infrastructural elements are newly constructed, and which are newly added. In summation, we do not feel we can effectively characterize changes in population or infrastructure at a timescale that is relevant for our analysis (i.e., month to month), and prefer to use a single static estimate. We have updated our methodology section to reflect this:

“We do not use dynamic values for population, roads, and infrastructure (i.e., we use a single value for each point in the raster for the entire 18-year analysis period). This means our model results can only be considered as representing changes in exposure relative to the specific time-period for these exposure datasets. We use the 2015 data for roads and population, and a snapshot of OSM data for July 2018.”

Line 154: I can see the logic and strengths of aggregating to admin level, but does this cause problems? Many admin boundaries have no logic in topography or climate. Won't this approach lead to a level of confusion, for example where an admin area covers both a mountainous and a flat area of terrain?

This is a good point, and we have elaborated upon this further in the text:

“Where admin regions include diverse landscapes (e.g., mountains and flat areas), it may be more difficult to pinpoint key locations for exposure, but we suggest that the admin-2 level offers the best balance of global reproducibility and small scale to allow for focus on local effects.”

Line 165: Is it really valid to use 12 month smoothed data in this way? If the rainfall patterns typically occur in a 3 month period (the summer monsoon), won't the signal be lost? This approach must also miss the effect of tropical cyclones, which generate rainfall for 48-72 hours, typically. Earlier in the paper, TCs were identified as being important.

This seems to be also stemming from the same lack of clarity about the modelling approach discussed above; namely, it seems like the reviewer thinks we are using the 12-month smoothed rainfall totals to assess exposure, but in fact we use the daily rainfall to assess exposure for each day in the time-period, and then take the sum total for each month to compare with monthly MVEI values. This is then smoothed over 12 months to remove seasonality. Since exposure is calculated for each day, we capture these short term events like tropical cyclones. We have amended the text to hopefully improve the clarity of this point (and apologise for the lack of clarity initially):

“The exposure output matches the temporal frequency of the MVEI data, but since it depends on daily rainfall data it captures the short duration, intense rainfall events most likely to trigger landslides.”

Line 202: This diagram needs further explanation. Some of the areas identified have very few landslide records in the various datasets - e.g. N. Australia, Tibet. South Asia and Taiwan, both of which are very landslide prone, show no effect. Why is this?

We regret that our explanation of Figure 3 was not clearer in the first version of this study. The regions shown in Figure 3 have a relationship between MVEI and landslides exposure that is above a level of statistical significance (R-squared greater than 0.2). This does not mean that there is a lot of landslides; it simply means that these areas exhibit a relationship between ENSO and the model output for landslide exposure. Figure 4 shows the magnitude of that relationship when compared to overall population. Those areas where low numbers of landslides occur are no longer evident in that figure since the change in landslide exposure tied to ENSO is tiny in comparison with the population (with the notable exception of Tibet, for example, where population density is very low). At the other end of the scale, places that the reviewer highlights where landslides often cause fatalities (Taiwan and South Asia) are not shown since there is not a strong relationship with ENSO – this is not to say that there is not a lot of landslides. It simply means that the variability is better explained by some other factor (e.g. South Asian monsoon). We have revised the text to better explain this point:

“A significant relationship between MVEI and the model outputs does not necessarily indicate that MVEI creates large changes. Simply put, even small changes can have significant relationships with MVEI if other climatological trends do not occlude the relationship. However, we are interested in the areas where ENSO causes the largest changes in exposure. As such, we must look at the slope of the relationship between MVEI and exposure – see, for example, the middle set of figures in Fig 2.

In Figure 4, we show the magnitude of the change driven by a unit shift in MVEI index. We calculate this based on linear least-squared regression. We choose to fit a linear relationship to all of the districts to ensure consistency, and because it seems to describe the relationship between MVEI and model outputs in areas where ENSO is known to be important (e.g., Figure 2). Figure 4 therefore shows the areas where we model a statistically significant impact of MVEI that leads to major changes in the fraction of population exposed to landslides. By looking at the fraction of the population exposed, we can consistently compare areas with varying population density.”

Line 225: This is an interesting result, but it needs explanation. What is the meaning of a shift in MVEI index? The finding that La Nina is more important than El Nino in Central America is a surprise. This was an area that was severely affected by El nino rainfall in 1997-8, primarily because of tropical cyclone impacts.

This is an important consideration. Naturally, a major challenge with the satellite rainfall data is that it does not encompass the 1997-1998 ENSO event, which makes testing this during that period a challenge. We should clarify that we do not mean that La Nina is ‘more important’ than El Nino, but instead that more landslide exposure results during La Nina events based on the rainfall data we have. In the revised text, we have included actual data on landslide induced fatalities and compared it with the MVEI values associated with the temporal occurrence (see response to reviewers 1 and 2). In central America, we find that proportionally higher fatalities are observed during La Nina in Costa Rica, Panama, and to a lesser extent Guatemala. While this does not necessarily say anything about the ‘97-98 event, it does support the validity of the model outputs in this area.

In revision, we have included a much more in-depth analysis of the comparison of model results vs real observations, and have additionally added text to explain the problems that the lack of 1997-98 data may create:

“In addition, we lack data covering the major ENSO event during 1997-1998, where significant landsliding was triggered in the Americas (Coe et al. 2004). A longer term analysis would provide a more comprehensive test of the model outputs and capture more of the variability inherent in ENSO cycles.”

Line 263: I agree with this analysis of the model, but the way that is being used here seems more in line with movement of large deep landslides rather than shallow, rapid ones.

We hope that this is a misunderstanding based on lack of clarity in the text. We suggest that this stems from the same misunderstanding as the reviewers more major comment above (comment #3). We incorporate daily rainfall data as in the original LHASA model to effectively capture the short term rainfall that triggers rapid, shallow landslides, and sum up the daily totals for exposure on a monthly basis, rather than analyse the sum total of rainfall for each month (which seems to be what the reviewer has taken from the text?). We have revised the text to address any lack of clarity to avoid any future misunderstanding.

Line 309: The conclusions are not reasonable, but the uncertainties in the model mean that it could not be used to anticipate the impact of future El Nino and La Nina events.

We respectfully note that it is not entirely clear what the reviewer means here? If the conclusions are not reasonable, then it is understandable that we should not extrapolate to future events. However, we feel that we have in revising the manuscript addressed the concerns above, and as such it is reasonable to at least qualitatively consider what future changes in ENSO may mean for landslide exposure changes. Naturally, a correlation based on past events does not imply causation nor that the same relationship would be expected in a future ENSO, but given correlations in prior events highlighted here we would suggest that a Bayesian approach to prediction would consider that a future ENSO event may lead to similar correlated changes in landslide exposure. We have made changes to the text to more clearly express that.

REVIEWER COMMENTS

Reviewer #1 (Remarks to the Author):

As I said in my first review, this is a novel paper addressing an important question at a global scale I definitely think it is worth publishing! All my previous comments about the positives of the work still stand (the surprisingly good correlations, the global scale, the step to exposure, the potential for widespread interest beyond the landslide research community). The authors have added considerable additional material to the paper to both explain the behaviour of their model and introduce empirical data. This is excellent! However, I still think that major revision of the text is needed before it can be published because: 1) the authors need to reshape the paper in the light of their additional analysis of how the model works; and 2) the empirical data while intriguing/exciting currently raise more questions than they answer. I didn't feel the paper needed the observations and I still feel that is the case. Adding them is fantastic but will require yet more work. To me this could easily be justified as a separate endeavour.

I have attached a copy of the manuscript with minor comments/typos and a review document where I revisit the three major comments from my first review and add a new comment on comparison with observations. The authors did a very good job of addressing my previous minor comments and as a result I have not included any of these or the author responses. The editor has asked that I give my opinion on the extent to which the authors have addressed the comments of reviewer 3. I do this separately at the end of this document.

My main concern in my first review was that "the drivers behind the relationships that it uncovers appear somewhat unclear and I don't think they need to be". You (the authors) have now sought to "explore the different relationships between rainfall, hazard, and exposure, to build a more complete picture." I am very pleased that you chose to do so. However you argue that "Since the primary intention of the paper is to explore the relationship of climatic driver (ENSO) with impact on humans (exposure to landslides), we have placed much of the analysis relating to the in-between steps (rainfall intensity, hazard) into the supplementary material." I do not think that it is sufficient to place this information in SI with only a summary in the main text. I think that those in-between steps are the paper's most important findings (MC1) and that they reduce (but don't remove) the need for detailed model sensitivity analysis (MC3).

The SI figures now make the drivers behind your findings clear, they are an excellent addition! ENSO shifts the frequency of intense rainfall, by different amounts in different places. These changes are predicted (from a simple model) to result in changes to landslide hazard and from there to landslide exposure. The model has a number of uncertain parameters, and some attempt to quantify sensitivity of findings to parameter uncertainty would certainly be useful. However, the rainfall changes appear to dominate the hazard and exposure predictions. As a result model structure and its parameters may play only a relatively minor role censoring areas of negligible landslide susceptibility and weighting the regional averages by population density. These steps result in relatively subtle shifts in significance (Figs S1-4) and sensitivity (Figs S5-8) and in each case they can be easily explained by the model's structure. The big finding is in the global rainfall patterns but I don't think the text of the paper reflects this. I think you need to adjust the text to make it the focus (I expand in MC1). You may disagree but if you do I think the requirements on your model uncertainty analysis become far stricter (I expand in MC3). In this case, you really do need to demonstrate very clearly through sensitivity analysis that your findings are robust to model (and particularly parameter) uncertainty.

Reviewer #2 (Remarks to the Author):

General Comments

My main concern in the first review was about the validation of the model, expressed in my previous comments # 8 and 30. I think the authors have met satisfactorily this concern, as well as the other ones I had listed.

There are only two additional minor specific comments, listed below.

Specific Comments

1. Lines 314-318: The results of Figures S5 and S6 are consistent with results of Grimm and Tedeschi (2009, reference 5 in this manuscript) for South America, showing that the ENSO impact is stronger and more extensive in the higher precipitation tail of the daily precipitation distribution, meaning that the impact of ENSO is stronger and more extensive on the 95th percentiles than on the monthly precipitation totals. This is visible comparing Figures S5 and S6.
2. Lines 436-439: The results of Figures 3 and 4 suggest that instead of mentioning just Colombia and Ecuador, you should mention more generally northern South America.

As I said in my first review, this is a novel paper addressing an important question at a global scale I definitely think it is worth publishing! All my previous comments about the positives of the work still stand (the surprisingly good correlations, the global scale, the step to exposure, the potential for widespread interest beyond the landslide research community). The authors have added considerable additional material to the paper to both explain the behaviour of their model and introduce empirical data. This is excellent! However, I still think that major revision of the text is needed before it can be published because: 1) the authors need to reshape the paper in the light of their additional analysis of how the model works; and 2) the empirical data while intriguing/exciting currently raise more questions than they answer. I didn't feel the paper needed the observations and I still feel that is the case. Adding them is fantastic but will require yet more work. To me this could easily be justified as a separate endeavour.

I have attached a copy of the manuscript with minor comments and typos. In this review I revisit the three major comments from my first review and add a new comment on comparison with observations. The authors did a very good job of addressing my previous minor comments and as a result I have not included any of these or the author responses. The editor has asked that I give my opinion on the extent to which the authors have addressed the comments of reviewer 3. I do this separately at the end of this document.

My main concern in my first review was that “the drivers behind the relationships that it uncovers appear somewhat unclear and I don't think they need to be”. You (the authors) have now sought to “explore the different relationships between rainfall, hazard, and exposure, to build a more complete picture.” I am very pleased that you chose to do so. However you argue that “Since the primary intention of the paper is to explore the relationship of climatic driver (ENSO) with impact on humans (exposure to landslides), we have placed much of the analysis relating to the in-between steps (rainfall intensity, hazard) into the supplementary material.” I do not think that it is sufficient to place this information in SI with only a summary in the main text. I think that those in-between steps are the paper's most important findings (MC1) and that they reduce (but don't remove) the need for detailed model sensitivity analysis (MC3).

The SI figures now make the drivers behind your findings clear, they are an excellent addition! ENSO shifts the frequency of intense rainfall, by different amounts in different places. These changes are predicted (from a simple model) to result in changes to landslide hazard and from there to landslide exposure. The model has a number of uncertain parameters, and some attempt to quantify sensitivity of findings to parameter uncertainty would certainly be useful. However, the rainfall changes appear to dominate the hazard and exposure predictions. As a result model structure and its parameters may play only a relatively minor role censoring areas of negligible landslide susceptibility and weighting the regional averages by population density. These steps result in relatively subtle shifts in significance (Figs S1-4) and sensitivity (Figs S5-8) and in each case they can be easily explained by the model's structure. The big finding is in the global rainfall patterns but I don't think the text of the paper reflects this. I think you need to adjust the text to make it the focus (I expand in MC1). You may disagree but if you do I think the requirements on your model uncertainty analysis become far stricter (I expand in MC3). In this case, you really do need to demonstrate very clearly through sensitivity analysis that your findings are robust to model (and particularly parameter) uncertainty.

Major comments

MC1) Refocussing the paper in the light of the new analysis

In my initial comments I said: “more space needs to be devoted to explaining / understanding what is driving the model response”. Though you “now compare several additional variables with MVEI” it is not sufficient to place the results of this analysis in sup info. The description of your results in the response is useful. “broadly, we find that while the areas where significant ($p < 0.05$) relationships are observed are consistent for each of the parameters (total rainfall, extreme rainfall days, number of nowcasts, population exposure), the level of impact (i.e. the slope) of the relationship differs, with MVEI only affecting nowcasts in mountainous areas, and parts of central Asia likely more responsive to increases in number of extreme rainfall events rather than changes in sum total of rainfall.” An expanded version of this interpretation should certainly feature in the article. You say that “(t)here is more detail that can be drawn from this analysis, but it is not our intention to focus on the model aspects within this study. We hope that this could provide fruitful questions for future research.” I don't think this can wait for future research. This to me is your main finding.

I found four comments from my first review relating to this general point that I did not feel were addressed in your response.

-“It would also be useful to know whether the steps from correlation between ENSO and extreme precipitation on to landslide hazard then to landslide exposure result in amplification of the correlation or in damping.”

-I recommended that you “investigate and report that behaviour since I think it would help readers both to understand what is driving the headline results and to trust the headlines themselves.” Your response might be either that the readers don’t need to understand this or that you feel that they understand it via another route. However, I haven’t yet seen an argument that persuades me of either of these.

- “Why is ENSO so well correlated with landslide exposure in some places?” I didn’t see an answer to this or an explanation of why this question is out of scope for the paper. Either would be fine.

- I commented that: “literature on ENSO effects on extreme rainfall needs to feature more strongly in the paper”. Though I do see closer connections to the literature in the discussion I don’t see this more in depth review early in the paper. This doesn’t need to be more than a couple of sentences but it would help to know what is already known about ENSO and extreme precipitation. For example, is this the first study to examine ENSO impacts on frequency of extreme rainfall events? Or to do so at global scale? If not what did others find? This could be done around either L43 or L87.

MC2) Modelled landslide exposure units are unclear or difficult to interpret

I said: “The units ‘nowcasts per km² per year’ or ‘person-nowcasts per km² per year’ are difficult to interpret in absolute terms. It would be useful to explain how the reader should interpret these units and how to relate them to hazard in terms of a landslide probability.” You “have tried to improve this by more clearly laying out the definition”. However, this is still a problem in the revised draft. It is not simply that the units are not clearly defined but that they need interpreting for the reader.

I still think your best chance of doing this will be to express them differently as I suggested before, suggesting that “ENSO related change in exposure is normalised by the amplitude of the seasonal exposure change” You responded that “expressing changes in comparison to seasonal changes in exposure ... may lead to misinterpretation. In places with little to no seasonal change.” This is a good point. However, the problem of unintuitive units remains. In Fig 4 I make the units (days/yr) per unit change in MVEI, but unit MVEI is not an intuitive quantity. I suggest normalising by the range in MVEI, either to express a (days/yr) per-cent change in MVEI or preferably max amplitude in (days/yr) across the full range of observed MVEI (i.e. -2 to 2). I could then read from Fig 4 that ENSO can change exposure per person by up to 10 days per year in the most ENSO sensitive areas, but leads to a change of <3 days per year across the vast majority of the globe. Histograms of these changes would be useful to aid interpretation of how important ENSO is as a driver of landslide exposure.

This normalisation makes it easier to interpret Fig 4 but doesn't account for the baseline exposure. For example, a change amplitude of 10 days per year (from ElNino to LaNina) might be 100% of the average annual exposure in one location and only 10% in another. I can understand that there are good reasons to report absolute changes but placing the changes in this context also seems important.

You addressed my comment on Fig 3 that: “(nowcasts per person), I think this should be person-nowcasts per km² per year.” The new units are explained as “the total number of people exposed to elevated landslide hazard in that district for each month to show seasonality”. This is much clearer. However, aggregating by administrative districts is not helpful for readers unless absolute values for those particular districts are the main thing that readers should take away. Different districts are different sizes so the reader loses any intuition for the values at this point.

My comment on Fig 4: “Normalising by population makes sense but population was already in the numerator so this could be simply change in nowcasts per km² per year with changing ENSO. You responded that: “Putting the total population in the denominator doesn’t quite get back to nowcasts, since it’s not always the same population exposed during each nowcast. As can be seen by comparing supplementary figures S7 and S8, it’s not identical – there are parts of the US where fractional population exposure changes less than nowcast changes with respect to MVEI, indicating perhaps that more people live in less exposed areas. However, this is relatively speculative, so we refrain from drawing these conclusions in the full text.” Thankyou for explaining this. It now makes sense. But it prompts two comments.

First, this would be easier to understand if you included the equation for exposure. I think, once normalised by population density it is something like:

$$E_j = \frac{\sum_{i=1}^n (R_i S_i P_i)}{\sum_{i=1}^n P_i}$$

where: E_j is the number of nowcast days per year for the average person in the region j . R_i is the frequency of 95th percentile weighted accumulated rainfall for cell i , S_i is its susceptibility (1 if the cell has susceptibility class > medium, 0 otherwise), P_i is its population. The units are then (days/yr) and are the average exposure to landsliding for an inhabitant of the region. This is a weighted average of H_i where population density is applied as the weighting factor.

Second, the text here using the example of parts of the US is very useful it explains what is happening within your model. I think it needs to be included in the main body of the article. It is important that regions with many inhabitants outside the susceptible zones will reduce the apparent average exposure for the region. This has implications for how the data are used. I think that this comment is connected to one of R3's concerns, see my comment on your response to R3 L154.

MC3) Some analysis of the sensitivity of the findings to model parameter uncertainty is needed

In my first review I said that detail on two aspects of the modelling was needed. First I requested a clearer explanation of the model structure, assumptions and parameters. Model setup is now clear to me though the model's assumptions could still be clearer.

Second, I argued for "a more robust treatment of the model's structural, input and parameter uncertainty and its impact on your findings" I highlighted this as "the primary limitation to an otherwise simple and elegant research design." I suggested that you: either conduct a full uncertainty analysis; or reshape the paper around relationships between ENSO and uncensored frequency of intense storms.

You chose to reanalyse the rainfall data, examining intermediate relationships within the model but did not reshape the analysis around these new results. To me this leaves the primary limitation above unaddressed. If you want to retain the focus on landslide exposure rather than landslide triggering rainfall then I think you need a more complete exploration of the sensitivity of your results to parameter uncertainty.

I do not agree that it "would be somewhat circular to conduct uncertainty analysis of the model outputs when the validation data is so limited" because such data are not required to examine the sensitivity of the findings to parameter uncertainty.

In particular, I asked: what is the sensitivity of your findings to:

- the choice of 95th percentile in ARI for normalising
- ARI parameters: 7 day window and weighting factor of -2
- The absolute ARI threshold of 6.6 mm
- The parameters of the fuzzy overlay model that defines landslide susceptibility and therefore whether or not rainfall above 95th percentile will trigger a nowcast.
- The choice to nowcast only for areas of moderate to high landslide hazard

I do not understand why/how "the answer is quite nuanced, and there are different ways to address it" and I can't trace these two arguments into the text that follows. I disagree that to test uncertainties requires "robust observational data to compare the outputs against", I don't think that is needed for sensitivity analysis (as opposed to validation/calibration). Comparison to "the fatality dataset of Froude and Petley (2018)." is certainly useful but to me is not related to the sensitivity analysis.

New comment RE comparison with observations

You now incorporated the fatality dataset of Froude and Petley (2018) and it is as you say "a useful first order comparison". However, this new analysis needs more attention. I am not convinced that you need this in the paper though I do think it will make the paper even better. If you choose to include it I think you need to deal with three issues.

- 1) It is not clear why you focus on Columbia (in Fig 5) or the other countries discussed in L386-94 but not on others. Explaining your rationale and sampling strategy would probably be enough. It looks as though you perhaps focussed on countries with significant predicted change (and perhaps also with gradient greater than x) and a sufficiently large observation set (e.g. $>n$ recorded landslide events). The quantitative thresholds for inclusion are important here if you are to argue that your choices were objective.

- 2) There is no quantitative evaluation of agreement with model predictions, Fig 5 shows only observations but model predictions are also available so a comparison is feasible. This could take the form of agreement in terms of the sign of the ENSO-exposure relationship, or be pushed further. If you expressed gradient as a function of average exposure you could see whether the gradients agreed.
- 3) It would help a lot if you could find a way to generalise the observations. Otherwise it is difficult to know what to conclude from L386-94 other than that there is good agreement in some places but not in others. That covers a very wide range of model performances.

Comments on Authors' response to Reviewer #3.

I have been asked by the editor to give comments on the authors' responses to Reviewer 3. This is somewhat difficult to do this since my perspective differs from that of Reviewer 3 but I can provide a third perspective on the discussion between the reviewer and the authors. The reviewer lists 3 particular concerns, which I will deal with in turn followed by a number of minor concerns in my view the authors have addressed almost all of these. The only outstanding concerns are the reviewer's second particular concern and their minor concern RE L154.

Particular Concerns

The reviewer's first concern is that presenting results on NAO and PDO even in the SI is spurious because the record length is too short. The authors respond that they include the analysis for completeness and because a reader could reasonably ask 'does this work for NAO and PDO'? I find this response convincing and think that they do enough to highlight that their main finding with respect to NAO and PDO is that record length is insufficient to evaluate them. I think this concern has been adequately addressed.

The reviewer's second concern is that not all ENSO events are alike and the amendments that the authors have made to clarify this in the manuscript appear to me to address this concern. Clearly there is a tradeoff here between splitting the phenomena to differentiate events that are functionally different and lumping them to enable sufficiently large number of events. I think that the authors get this balance about right and that this concern has been adequately addressed.

The reviewer's third concern is that only a subset of the rainstorms that trigger landslides can be captured within the analysis performed here. This comment may have stemmed from a misunderstanding of the model that the authors use; and their response seeks primarily to clear up this misunderstanding. However, the reviewer's point about clarifying the subset of rainstorms that trigger landslides that can be captured in their method does deserve a more detailed treatment. I think this could be done easily within the introduction or methods by highlighting the range of rainfall durations responsible for the type of rapid catastrophic landslides studied here; then comparing these rainfall durations to the resolution of the rainfall data examined here.

Minor Concerns

This comment had no response. I think that is just a straightforward oversight as it looks easy to address: This should be the South Asian summer monsoon (not India). Tropical cyclone seasonality - again, this is related to TCs in the NWP, the Indian Ocean and the Atlantic, primarily.

L51: Adequately addressed

L77: Adequately addressed. It is a big assumption but the authors clearly explain this in the paper

L125: Adequately addressed. This is now clear in the paper

L154: This remains a concern (albeit a minor one) for me. R3 said: *I can see the logic and strengths of aggregating to admin level, but does this cause problems? Many admin boundaries have no logic in topography or climate. Wont this approach lead to a level of confusion, for example where an admin area covers both a mountainous and a flat area of terrain?*

The authors responded that: *This is a good point, and we have elaborated upon this further in the text: "Where admin regions include diverse landscapes (e.g., mountains and flat areas), it may be more difficult to pinpoint key locations for exposure, but we suggest that the admin-2 level offers the best balance of global reproducibility and small scale to allow for focus on local effects."*

I think the authors could make the limitations clearer, R3 explains where this issue arises in their comment.

L165: I agree with the authors that this comment stemmed from a misunderstanding and they have now clarified the text. The authors could consider whether expressing their model structure as a flow diagram and/or as a series of

equations in SI might further insure against this type of misunderstanding. Having said that I think a careful reading of the text in its current form is sufficient to understand what you have done.

L202: Adequately addressed.

L225: Adequately addressed, this was a really useful comment/response combination.

L263: Adequately addressed. This comment stems from a misunderstanding now clarified.

L309: The changes to the text would have been useful to see here. However, the conclusions (which appear to have been the reviewer's area of concern here) don't contain problematic claims about prediction in my view. The claims of the conclusion are framed around the findings of the paper, which is reasonable. The second paragraph discusses what we can learn from these findings but I don't see any of these claims as unreasonably strong given the content of the paper.

Response to Reviews: Emberson et al. 202X, El Nino and Landslides (Nature Comms)

REVIEWER COMMENTS

Reviewer #1 (Remarks to the Author):

Second review of Emberson et al. 2020 Nature Comms by David Milledge

As I said in my first review, this is a novel paper addressing an important question at a global scale I definitely think it is worth publishing! All my previous comments about the positives of the work still stand (the surprisingly good correlations, the global scale, the step to exposure, the potential for widespread interest beyond the landslide research community). The authors have added considerable additional material to the paper to both explain the behaviour of their model and introduce empirical data. This is excellent! However, I still think that major revision of the text is needed before it can be published because: 1) the authors need to reshape the paper in the light of their additional analysis of how the model works; and 2) the empirical data while intriguing/exciting currently raise more questions than they answer. I didn't feel the paper needed the observations and I still feel that is the case. Adding them is fantastic but will require yet more work. To me this could easily be justified as a separate endeavour.

I have attached a copy of the manuscript with minor comments and typos. In this review I revisit the three major comments from my first review and add a new comment on comparison with observations. The authors did a very good job of addressing my previous minor comments and as a result I have not included any of these or the author responses. The editor has asked that I give my opinion on the extent to which the authors have addressed the comments of reviewer 3. I do this separately at the end of this document.

We thank the reviewer for another extremely thorough and detailed review. The comments provided have served to improve the paper for a second time. We are especially grateful that the reviewer took the time to go over the comments from reviewer 3; this really is going above and beyond. Below, we have provided responses to each of the comments. We feel that our responses now adequately satisfy the reviewer's concerns.

My main concern in my first review was that "the drivers behind the relationships that it uncovers appear somewhat unclear and I don't think they need to be". You (the authors) have now sought to "explore the different relationships between rainfall, hazard, and exposure, to build a more complete picture." I am very pleased that you chose to do so. However you argue that "Since the primary intention of the paper is to explore the relationship of climatic driver (ENSO) with impact on humans (exposure to landslides), we have placed much of the analysis relating to the in-between steps (rainfall intensity, hazard) into the supplementary material." I do not think that it is sufficient to place this information in SI with only a summary in the main text. I think that those in-between steps are the paper's most important findings (MC1) and that they reduce (but don't remove) the need for detailed model sensitivity analysis (MC3).

We appreciate the feedback here. This is an important point, and we apologise if we didn't capture the nuance of the earlier comment in the first round of revisions. On the one hand, we disagree with the reviewer that the in-between steps are the most important finding; our view remains that the most important take home message is shown in Figure 4 (i.e., where ENSO may most strongly drive changes in exposure). However, we can see where the reviewer is coming from and have made changes to the text

to better reflect the conceptual understanding that the in-between stages represent. See response to main comment 1, below.

The SI figures now make the drivers behind your findings clear, they are an excellent addition! ENSO shifts the frequency of intense rainfall, by different amounts in different places. These changes are predicted (from a simple model) to result in changes to landslide hazard and from there to landslide exposure. The model has a number of uncertain parameters, and some attempt to quantify sensitivity of findings to parameter uncertainty would certainly be useful.

However, the rainfall changes appear to dominate the hazard and exposure predictions. As a result model structure and its parameters may play only a relatively minor role censoring areas of negligible landslide susceptibility and weighting the regional averages by population density. These steps result in relatively subtle shifts in significance (Figs S1-4) and sensitivity (Figs S5-8) and in each case they can be easily explained by the model's structure. The big finding is in the global rainfall patterns but I don't think the text of the paper reflects this. I think you need to adjust the text to make it the focus (I expand in MC1). You may disagree but if you do I think the requirements on your model uncertainty analysis become far stricter (I expand in MC3). In this case, you really do need to demonstrate very clearly through sensitivity analysis that your findings are robust to model (and particularly parameter) uncertainty.

We apologise that we did not adequately address this in the first round of revisions. The choice of rainfall weighting coefficients was previously tested by Kirschbaum and Stanley in the 2018 paper describing the LHASA model, and the parameters associated with the susceptibility model were tested in the study that produced that model initially, and it is our mistake that we neglected to reference that again here to support the parameter choices.

We agree with the reviewer that it is clear the rainfall input is the most significant factor correlated with ENSO changes. We have significantly redrafted our text to reflect this and added more analysis of the impact of where the specific model aspects involved with rainfall (i.e., total rainfall or extreme rainfall) are most significantly correlated with ENSO. Below, we detail these changes, which we hope now address the reviewer's point.

Major comments

MC1) Refocussing the paper in the light of the new analysis

In my initial comments I said: "more space needs to be devoted to explaining / understanding what is driving the model response". Though you "now compare several additional variables with MVEI" it is not sufficient to place the results of this analysis in sup info. The description of your results in the response is useful. "broadly, we find that while the areas where significant ($p < 0.05$) relationships are observed are consistent for each of the parameters (total rainfall, extreme rainfall days, number of nowcasts, population exposure), the level of impact (i.e. the slope) of the relationship differs, with MVEI only affecting nowcasts in mountainous areas, and parts of central Asia likely more responsive to increases in number of extreme rainfall events rather than changes in sum total of rainfall." An expanded version of this interpretation should certainly feature in the article. You say that "(t)here is more detail that can be drawn from this analysis, but it is not our intention to focus on the model aspects within this study. We hope that this could provide fruitful questions for future research." I don't think this can wait for future research.

This to me is your main finding. I found four comments from my first review relating to this general point that I did not feel were addressed in your response.

“It would also be useful to know whether the steps from correlation between ENSO and extreme precipitation on to landslide hazard then to landslide exposure result in amplification of the correlation or in damping.”

As the reviewer has correctly noted, this study is essentially an exploration of how extreme rainfall changes are related to ENSO. Since the other inputs (population, susceptibility) are static, the strength of the relationship between ENSO and the modeled exposure in a given district over a time series is determined fundamentally by the changing rainfall patterns. We appreciate that this may not have been clear enough in this initial or revised text. We have added new figures to the supplementary material and main text to show the difference in the strength of relationship when each additional step of the model is introduced. There are clear spatial patterns in the differences in the strength of relationship between ENSO-total rainfall and ENSO-extreme rainfall that indicate where extreme rainfall has a stronger relationship between ENSO than simply total rainfall, but the spatial patterns are less consistent for the introduction of susceptibility and population, which supports the reviewer’s assessment that rainfall is the key determinant of changes.

Critically, the places where exposure and ENSO are linked in most cases already demonstrate significant relationships between ENSO and total rainfall. While extreme rainfall is more strongly correlated with MVEI in some settings, the key locations for exposure (e.g., Colombia, Philippines, Indonesia) do not for the most part exhibit stronger relationships when extreme rainfall, susceptibility or exposure is added. In other words, total rainfall is driving the increases in exposure where there are strong exposure relationships; however, there are other settings where extreme rainfall gives stronger relationships.

This is discussed in a new set of paragraphs, with the new Figure 6 and Supplementary Figures S9 and S10:

“These results raise the question of why there are such strong relationships between ENSO and exposure in the highlighted locations. We can explore the impact of each part of the model by contrasting the significance of the relationships between total rainfall, extreme rainfall, hazard nowcasts, and exposure and MVEI value. In Figure 3A, we show the p-values for each district for the relationship between total rainfall and MVEI. It is clear that there are already strong relationships between ENSO and total rainfall in locations where we model large changes in exposure due to ENSO (i.e., South East Asia and Central America and northern South America). To determine whether considering only the extreme rainfall leads to stronger relationships, we compare the p-values for total rainfall and extreme rainfall relationships with MVEI; this is shown below in Figure 6. In some geographical areas including parts of Central Asia, Mexico, Iran, China, Luzon, and parts of Thailand, Cambodia and Vietnam, the relationship between MVEI and extreme rainfall is stronger (extreme rainfall p-value is lower than total rainfall) suggesting that the impact of ENSO on extreme rainfall is more significant than on total rainfall in these locations. This is in line with other studies of ENSO-induced rainfall changes^{40,41}, including global studies based on gauge data²⁰ and earlier studies using TRMM and GPCP rainfall data²¹. However, the impact of ENSO on extreme rainfall is weaker than on total rainfall in critical areas in Colombia, Indonesia, and the Southern Philippines, all key locations where we model significant relationships between ENSO and landslide exposure. This suggests that the impact of ENSO on total rainfall, rather than extreme rainfall, may be the main driver of changes in exposure in many locations.

At the same time, ENSO-driven changes in extreme rainfall clearly play a significant role in places like Southern Brazil, Mexico, Eastern China, and parts of Central Asia. Small changes in total rainfall are observed in Central Asia (Figure 1) but large shifts in exposure are modelled in Tajikistan and Kyrgyzstan, likely linked to changes in extreme rainfall (Figure 6); this is in line with other studies of ENSO-induced rainfall changes^{40,41}. Interestingly, these are the areas where El Nino conditions lead to greater modelled landslide exposure (highlighted most clearly in Figure 5), while the areas where total rainfall seems to be more relevant are those where La Nina conditions lead to greater impacts (parts of northern South America and Indonesia, highlighted in Figure 4). This suggests that the two modes of the ENSO system may lead to changes in landslide impact due to differing effects.

We have also compared the strength of relationship for hazard nowcasts and extreme rainfall, and exposure and hazard nowcasts, to test where adding each model input strengthens and weakens the relationship with MVEI (Supplementary Figures S9 and S10, respectively). While there are broader geographical regions where consideration of extreme rainfall increases or decreases the strength of the relationship, adding the susceptibility and population data has smaller and less regionally consistent effects. This suggests that total rainfall, and to a lesser extent extreme rainfall, is most strongly linked to ENSO, and that the modelled relationship with landslide hazard and exposure is not significantly strengthened or weakened by the addition of susceptibility and population data in the locations we highlight in Figure 4 and 5. Changes in the strength of the relationship between MVEI and exposure and hazard differ most strongly from the relationships for extreme rainfall in the flat, low population areas of the Tibetan plateau and Amazon rainforest.

Figure 6: Ratio of p-values for the relationship between MVEI and total rainfall (see Figure 3A) and MVEI and extreme rainfall (Supplementary Figure S2). Areas with yellow, orange and red colours indicate that extreme rainfall is more strongly correlated with MVEI than total rainfall.

There remain some uncertainties and challenges in this type of analysis. Since LHASA model outputs depend on rainfall exceeding the historical 95th percentile, the model is only sensitive to ENSO-induced changes in the frequency of those extreme events, but not to an increase in intensity of extreme events. Studies have shown ENSO-driven changes in rainfall extreme values in South America⁵ and the United States³⁹; as such, our results may miss some impacts where this is the case. However, other studies have shown that ENSO affects extreme rainfall frequency rather than the peak intensity⁹, suggesting that any effect on extremes may vary region to region. Increases in the intensity of events exceeding the 95th percentile may lead to increased impact from landslides. Our model outputs will provide the same hazard and exposure estimate regardless of the intensity over the 95th percentile, and so if ENSO leads to an increase in peak intensity we may underestimate the impacts on population and infrastructure.

-I recommended that you “investigate and report that behaviour since I think it would help readers both to understand what is driving the headline results and to trust the headlines themselves.” Your response might be either that the readers don’t need to understand this or that you feel that they understand it via another route. However, I haven’t yet seen an argument that persuades me of either of these.

We thank the reviewer for this point. We feel that we have addressed this comment in the response to the prior part of this comment.

- “Why is ENSO so well correlated with landslide exposure in some places?” I didn’t see an answer to this or an explanation of why this question is out of scope for the paper. Either would be fine.

- I commented that: “literature on ENSO effects on extreme rainfall needs to feature more strongly in the paper”. Though I do see closer connections to the literature in the discussion I don’t see this more in depth review early in the paper. This doesn’t need to be more than a couple of sentences but it would help to know what is already known about ENSO and extreme precipitation. For example, is this the first study to examine ENSO impacts on frequency of extreme rainfall events? Or to do so at global scale? If not what did others find? This could be done around either L43 or L87.

We thank the reviewer for the comment. We have added a sentence in the introduction to explain that prior studies have explored the relationship between ENSO and extreme rainfall:

"Prior global studies have indicated that ENSO has significant impacts on global patterns of total rainfall, as well as the incidence of extreme rainfall^{20, 21}."

As well as a sentence in the discussion section to indicate that our findings in terms of the total rainfall and extreme rainfall relationships are consistent with these earlier studies:

"This is in line with other studies of ENSO-induced rainfall changes^{40,41}, including global studies based on gauge data²⁰ and earlier studies using TRMM and GPCP rainfall data²¹."

MC2) Modelled landslide exposure units are unclear or difficult to interpret

I said: “The units ‘nowcasts per km² per year’ or ‘person-nowcasts per km² per year’ are difficult to interpret in absolute terms. It would be useful to explain how the reader should interpret these units and how to relate them to hazard in terms of a landslide probability.” You “have tried to improve this by more clearly laying out the definition”. However, this is still a problem in the revised draft. It is not simply that the units are not clearly defined but that they need interpreting for the reader.

I still think your best chance of doing this will be to express them differently as I suggested before, suggesting that “ENSO related change in exposure is normalised by the amplitude of the seasonal exposure change” You responded that “expressing changes in comparison to seasonal changes in exposure ... may lead to misinterpretation. In places with little to no seasonal change.” This is a good point. However, the problem of unintuitive units remains. In Fig 4 I make the units (days/yr) per unit change in MVEI, but unit MVEI is not an intuitive quantity. I suggest normalising by the range in MVEI, either to express a (days/yr) per-cent change in MVEI or preferably max amplitude in (days/yr) across the full range of observed MVEI (i.e. -2 to 2). I could then read from Fig 4 that ENSO can change exposure per person by up to 10 days per year in the most ENSO sensitive areas, but leads to a change of <3 days per year across the vast majority of the globe. Histograms of these changes would be useful to aid interpretation of how important ENSO is as a driver of landslide exposure.

Thanks to the reviewer for raising this point. The MVEI index is theoretically open ended on either scale, since it depends on observations of key components including sea surface temperature, near surface air temperature, and outgoing long-wave radiation. As such, it isn’t strictly speaking feasible to normalize by a maximum value. In addition, since our record doesn’t include the intense El-Nino event of 1998-1999, to suggest that we normalize by the full range of observed values may be misleading since it would not include the even larger swings in that period. As such, we feel that the current expression of the units in Figure 4 is as simple as we can make it.

This normalisation makes it easier to interpret Fig 4 but doesn't account for the baseline exposure. For example, a change amplitude of 10 days per year (from El Niño to La Niña) might be 100% of the average annual exposure in one location and only 10% in another. I can understand that there are good reasons to report absolute changes but placing the changes in this context also seems important.

This is a really good point; we appreciate the reviewer pushing it again here. We have added a new figure 5 and text to show where the relative changes in exposure are greatest. While the same regions that show large magnitude changes are also seen in the relative changes, some other regions (including Southern Brazil and central Asia) also show up. We hope this helps clarify this point!

New text:

"In Figure 5, we show the relative change in exposure due to a unit change in MVEI relative to the monthly average exposure; the areas where ENSO leads to large magnitude changes also show large relative changes, but there are also parts of Mexico, Southern Brazil, and Central Asia that show large relative increases during El Niño periods."

You addressed my comment on Fig 3 that: "(nowcasts per person), I think this should be person-nowcasts per km² per year." The new units are explained as "the total number of people exposed to elevated landslide hazard in that district for each month to show seasonality". This is much clearer. However, aggregating by administrative districts is not helpful for readers unless absolute values for those particular districts are the main thing that readers should take away. Different districts are different sizes so the reader loses any intuition for the values at this point. My comment on Fig 4: "Normalising by population makes sense but population was already in the numerator so this could be simply change in nowcasts per km² per year with changing ENSO. You responded that: "Putting the total population in the denominator doesn't quite get back to nowcasts, since it's not always the same population exposed during each nowcast. As can be seen by comparing supplementary figures S7 and S8, it's not identical – there are parts of the US where fractional population exposure changes less than nowcast changes with respect to MVEI, indicating perhaps that more people live in less exposed areas. However, this is relatively speculative, so we refrain from drawing these conclusions in the full text." Thankyou for explaining this. It now makes sense. But it prompts two comments. First, this would be easier to understand if you included the equation for exposure. I think, once normalised by population density it is something like:

$$E_j = \frac{\sum (R_i S_i P_i)}{\sum P_i} \quad P_i = 1j$$

where: E_j is the number of nowcast days per year for the average person in the region j . R_i is the frequency of 95th percentile weighted accumulated rainfall for cell i , S_i is its susceptibility (1 if the cell has susceptibility class > medium, 0 otherwise), P_i is its population. The units are then (days/yr) and are the average exposure to landsliding for an inhabitant of the region. This is a weighted average of H_i where population density is applied as the weighting factor. Second, the text here using the example of parts of the US is very useful it explains what is happening within your model. I think it needs to be included in the main body of the article. It is important that regions with many inhabitants outside the susceptible zones will reduce the apparent average exposure for the region. This has implications for how the data are used. I think that this comment is connected to one of R3's concerns, see my comment on your response to R3 L154.

This is a really helpful comment! There are a couple of points to respond to here, so we'll break it down into a few parts.

1. Normalising by population – we suggest that by showing the relative changes in exposure relative to the longer term average (Figure 5) *and* relative to the total population (Figure 4), there are now two spatially consistent ways to express the changes due to ENSO. Although we have not normalized the pop_exp values in Figure 2, we have now also included population totals for each district to help give context to the values (since expressing as a fraction would ultimately leave the y-axis unchanged).

2. Synthesizing the units more clearly – this equation is a great expression of the units we use. If you don't object, we have used this description in the text to help clarify the data. It works really well! New text:

“We can express the exposure estimates E_j for each admin district j as follows:

$$E_j = \sum_i \left[\frac{R_i S_i}{P_i} \right] j$$

Where R_i is the number of days when rainfall exceeds the historical 95th percentile in cell i , S_i is the susceptibility of that cell (1 if the cell has susceptibility greater than medium, 0 otherwise), and P_i is the population across that cell. This gives a final unit of person-days/year, expressing the average exposure for that district. This can then be normalized by the total population in region j (e.g. Figure 4) or by the long term monthly average exposure (e.g. Figure 5) to allow for intercomparison.”

MC3) Some analysis of the sensitivity of the findings to model parameter uncertainty is needed

In my first review I said that detail on two aspects of the modelling was needed. First I requested a clearer explanation of the model structure, assumptions and parameters. Model setup is now clear to me though the model's assumptions could still be clearer.

Second, I argued for “a more robust treatment of the model's structural, input and parameter uncertainty and its impact on your findings” I highlighted this as “the primary limitation to an otherwise simple and elegant research design.” I suggested that you: either conduct a full uncertainty analysis; or reshape the paper around relationships between ENSO and uncensored frequency of intense storms. You chose to reanalyse the rainfall data, examining intermediate relationships within the model but did not reshape the analysis around these new results. To me this leaves the primary limitation above unaddressed. If you want to retain the focus on landslide exposure rather than landslide triggering rainfall then I think you need a more complete exploration of the sensitivity of your results to parameter uncertainty. I do not agree that it “would be somewhat circular to conduct uncertainty analysis of the model outputs when the validation data is so limited” because such data are not required to examine the sensitivity of the findings to parameter uncertainty.

We appreciate the comment here, it is certainly an important point. The parameters discussed by the reviewer below are each related to earlier studies that derived the susceptibility model (fuzzy overlay parameters) and LHASA nowcast model (ARI parameters and thresholds), and in those earlier studies the choice of parameters was analysed. We have clarified in revising our introductory and methods text that these parameters are part of the model design and justified based on prior research.

In particular, I asked: what is the sensitivity of your findings to:

- the choice of 95th percentile in ARI for normalising
- ARI parameters: 7 day window and weighting factor of -2
- The absolute ARI threshold of 6.6 mm

- The parameters of the fuzzy overlay model that defines landslide susceptibility and therefore whether or not rainfall above 95th percentile will trigger a nowcast.

- The choice to nowcast only for areas of moderate to high landslide hazard

I do not understand why/how “the answer is quite nuanced, and there are different ways to address it” and I can't trace these two arguments into the text that follows. I disagree that to test uncertainties requires “robust observational data to compare the outputs against”, I don't think that is needed for sensitivity analysis (as opposed to validation/calibration). Comparison to “the fatality dataset of Froude and Petley (2018).” is certainly useful but to me is not related to the sensitivity analysis.

We thank the reviewer for this comment, the logic of our prior answer was not totally sound. This is why we spent some time investigating the possibility of a sensitivity analysis, and as discussed above we feel it isn't feasible within the current scope of the paper. As such, we have tried to follow the suggestion of the reviewer to refocus around the findings related to rainfall – ENSO links. Ultimately, the outputs of the model are representative of the LHASA model parameters; the rainfall connection is based on purely observational data, but the results for hazard depend on model parameter choice. We have added the following text to the discussion to expound on this point:

“In addition, while the observed relationships between total rainfall and frequency of extreme rainfall and ENSO result from analysis of rainfall observations, the connections between hazard and exposure depend on model parameter choices. The parameters used in the LHASA model have been calibrated based on distance to perfect classification of the NASA Global Landslide Catalog38, which gives a false positive rate of 1% and a true positive rate of between 20 and 50% depending on the time interval used for landslide analysis. The exposure and hazard connections to ENSO must therefore be viewed as model outputs rather than direct observations.”

New comment RE comparison with observations

You now incorporated the fatality dataset of Froude and Petley (2018) and it is as you say “a useful first order comparison”. However, this new analysis needs more attention. I am not convinced that you need this in the paper though I do think it will make the paper even better. If you choose to include it I think you need to deal with three issues.

1) It is not clear why you focus on Columbia (in Fig 5) or the other countries discussed in L386-94 but not on others. Explaining your rationale and sampling strategy would probably be enough. It looks as though you perhaps focussed on countries with significant predicted change (and perhaps also with gradient greater than x) and a sufficiently large observation set (e.g. >n recorded landslide events). The quantitative thresholds for inclusion are important here if you are to argue that your choices were objective.

2) There is no quantitative evaluation of agreement with model predictions, Fig 5 shows only observations but model predictions are also available so a comparison is feasible. This could take the form of agreement in terms of the sign of the ENSO-exposure relationship, or be pushed further. If you expressed gradient as a function of average exposure you could see whether the gradients agreed.

3) It would help a lot if you could find a way to generalise the observations. Otherwise it is difficult to know what to conclude from L386-94 other than that there is good agreement in some places but not in others. That covers a very wide range of model performances.

We appreciate the reviewer taking the time to dig into the new parts of the paper in detail. We have used the comments as impetus to extensively revise this section, and directly compare the model

estimates for each MVEI interval with the fatality data. This provides a much more direct and quantitative analysis. We have also explicitly spelled out the reasons for excluding certain countries, and made a more complete discussion of the results in various areas. The new section is replicated below, since it addresses each of the concerns above:

“Comparison with real data

It is important to assess whether the model outputs derived here have any reflection in real data. We are not aware of globally consistent temporal datasets of exposure to landslides that can be directly used to compare with our model outputs. However, fatalities resulting from landslides have been recorded between 2004 and 2016 by Froude and Petley¹⁹ around the world. Froude & Petley consider that their Global Fatal Landslide Dataset (GFLD) likely captures the majority of fatal landslides, with only 15% underestimation ⁴⁷. In order to compare this data with our model outputs, we have subdivided the landslides leading to fatalities by country. Given that the GFLD contains fewer than 5,000 landslides events, there is insufficient data to split them into admin-2 districts as we have for the model output. In each country, we compare the average frequency of fatal landslides for MVEI intervals with the average landslide exposure we model. There are only 26 countries in the GFLD with more than 30 recorded events, so we exclude other countries as we suggest data is too limited to draw conclusions. From these 26, we exclude Bangladesh, Bhutan, Italy, Japan, Nepal, Sri Lanka, Taiwan, and Turkey from further discussion as these are in areas where our model does not suggest a strong ENSO effect. The figures for these excluded countries are still provided in the supplementary material. In the remaining countries, model performance varies. An example for Colombia is shown below in Figure 7; in the top part of the figure, the histograms of modelled exposure and fatalities are shown, and below the relative ratio of the two is shown; a consistent value of 1 would indicate perfect predictive performance. We plot the histogram of MVEI value for the month in which each landslide occurred, and compare it with the histogram of MVEI for each month in the recording period. By comparing the two histograms, we can qualitatively assess which MVEI values have more landslides than we would expect based on the null hypothesis of equal distribution across all MVEI values. An example is shown in Figure 5, for Colombia.

Figure 7. Top: Histogram of average fatal landslide events in Colombia split by MVEI index are shown in red; in blue, a histogram of the average modelled landslide exposure split by MVEI index. In the lower figure, the relative ratio of these two values is shown. A value of 1 indicates that the model perfectly matches the fatality data relative to the maxima of each; lower values indicate that the model over-predicts fatalities.

For most of the analysed countries, the ENSO intervals where the model estimates show the highest average landslide exposure match the ENSO interval where fatal landslides were greatest, but the exposure estimates are also generally less variable than the fatality data. In Figure 8, we show the results for Guatemala, where the fatality data is far more variable than the exposure model estimates. We suggest that with larger numbers of recorded events, the variability in fatality data may show less noise, but this also suggests that the model outputs tend toward the mean exposure to a greater extent than is reflected in reality. The model captures the patterns of landslide impacts in most of the assessed countries, suggesting that our results are a useful first-order estimate of the connection between ENSO and landslide impacts. We calculate the standard deviation of the ratio of the model estimates and fatality data (lower part of Figures 7 and 8, see supplementary material for full table) as a way to quantify how closely the data correspond; the lowest deviations are seen in Indonesia, China, Mexico, and Uganda, with the largest variability (suggesting worse predictive skill) in Thailand, Myanmar, and India (notably countries where the South Asian Monsoon plays an important role). Assessing the specific relationships within individual countries represents an important topic for future research.

Figure 8. Top: Histogram of average fatal landslide events in Guatemala split by MVEI index are shown in red; in blue, a histogram of the average modelled landslide exposure split by MVEI index. In the lower figure, the relative ratio of these two values is shown. A value of 1 indicates that the model perfectly matches the fatality data relative to the maxima of each; lower values indicate that the model over-predicts fatalities.”

Comments on Authors’ response to Reviewer #3.

I have been asked by the editor to give comments on the authors’ responses to Reviewer 3. This is somewhat difficult to do this since my perspective differs from that of Reviewer 3 but I can provide a third perspective on the discussion between the reviewer and the authors. The reviewer lists 3 particular concerns, which I will deal with in turn followed by a number of minor concerns in my view the authors have addressed almost all of these. The only outstanding concerns are the reviewer’s second particular concern and their minor concern RE L154.

We greatly appreciate the extra effort taken by reviewer 1 to take on the comments from the original reviewer 3. The third opinion is very welcome! We respond line-by-line below.

Particular Concerns

The reviewer’s first concern is that presenting results on NAO and PDO even in the SI is spurious because the record length is too short. The authors respond that they include the analysis for completeness and because a reader could reasonably ask ‘does this work for NAO and PDO’? I find this response convincing and think that they do enough to highlight that their main finding with respect to NAO and PDO is that record length is insufficient to evaluate them. I think this concern has been adequately addressed.

Well noted, thank you.

The reviewer’s second concern is that not all ENSO events are alike and the amendments that the authors have made to clarify this in the manuscript appear to me to address this concern. Clearly there is a tradeoff here between splitting the phenomena to differentiate events that are functionally different and lumping them to enable sufficiently large number of events. I think that the authors get this balance about right and that this concern has been adequately addressed.

Again, thank you for the support.

The reviewer’s third concern is that only a subset of the rainstorms that trigger landslides can be captured within the analysis performed here. This comment may have stemmed from a misunderstanding of the model that the authors use; and their response seeks primarily to clear up this misunderstanding. However, the reviewer’s point about clarifying the subset of rainstorms that trigger landslides that can be captured in their method does deserve a more detailed treatment. I think this could be done easily within the introduction or methods by highlighting the range of rainfall durations responsible for the type of rapid catastrophic landslides studied here; then comparing these rainfall durations to the resolution of the rainfall data examined here.

We have added text to the discussion to explain that while it is theoretically possible for short duration events (sub-hourly microburst of rain) to trigger landsliding without exceeding the daily 95th percentile, analysis of intensity duration thresholds suggests that the intensity necessary to do so will generally

exceed the 95th percentile of daily rainfall across most of the land areas around the world, suggesting that our model is sensitive to this kind of event. New text:

“In addition, extremely short duration (sub-hourly) rainfall events that exceed local intensity-duration thresholds that do not lead to total daily rainfall exceeding the historical 95th percentile will not be detected. However, the intensity of this kind of rainfall event necessary to trigger landsliding is likely on the order of 10-100mm/hr⁴, which is greater than the daily 95th percentile of historical rainfall across most of the continental surface¹⁵. As such, if these events are captured by the satellite, they will likely lead to a model hazard nowcast”

Minor Concerns

This comment had no response. I think that is just a straightforward oversight as it looks easy to address:

This should be the South Asian summer monsoon (not India). Tropical cyclone seasonality - again, this is related to TCs in the NWP, the Indian Ocean and the Atlantic, primarily.

Changed ‘Indian Summer Monsoon’ to ‘South Asian Summer Monsoon’

L51: Adequately addressed

Noted, thank you

L77: Adequately addressed. It is a big assumption but the authors clearly explain this in the paper

Noted, thank you

L125: Adequately addressed. This is now clear in the paper

Noted, thank you

L154: This remains a concern (albeit a minor one) for me. R3 said: *I can see the logic and strengths of aggregating to admin level, but does this cause problems? Many admin boundaries have no logic in topography or climate. Wont this approach lead to a level of confusion, for example where an admin area covers both a mountainous and a flat area of terrain?*

The authors responded that: *This is a good point, and we have elaborated upon this further in the text: “Where admin regions include diverse landscapes (e.g., mountains and flat areas), it may be more difficult to pinpoint key locations for exposure, but we suggest that the admin-2 level offers the best balance of global reproducibility and small scale to allow for focus on local effects.”*

I think the authors could make the limitations clearer, R3 explains where this issue arises in their comment.

We have rephrased this section slightly, to help identify the potential issue more clearly:

“Many admin districts are not tied to topography or climate, and where admin regions include diverse landscapes (e.g., mountains and flat areas), it may be more difficult to pinpoint key locations for

exposure. If exposure estimates are aggregated in such districts, a signal of a relationship between exposure and ENSO may be masked by noise or conflicting signals from a different part of the area. Smaller admin districts are less likely to merge conflicting signals, and we suggest that the admin-2 level offers the best balance of global reproducibility and small scale to allow for a focus on local effects.”

L165: I agree with the authors that this comment stemmed from a misunderstanding and they have now clarified the text. The authors could consider whether expressing their model structure as a flow diagram and/or as a series of equations in SI might further insure against this type of misunderstanding. Having said that I think a careful reading of the text in its current form is sufficient to understand what you have done.

In line with main comment #2 (above), we have added in a key explanatory equation and improvements to the text throughout to help with clarity, which hopefully helps address this point.

L202: Adequately addressed.

Noted, thank you

L225: Adequately addressed, this was a really useful comment/response combination.

Noted, thank you

L263: Adequately addressed. This comment stems from a misunderstanding now clarified.

Noted, thank you

L309: The changes to the text would have been useful to see here. However, the conclusions (which appear to have been the reviewer’s area of concern here) don’t contain problematic claims about prediction in my view. The claims of the conclusion are framed around the findings of the paper, which is reasonable. The second paragraph discusses what we can learn from these findings but I don’t see any of these claims as unreasonably strong given the content of the paper.

Noted, thank you

Reviewer 1 also provided a number of comments on a separate pdf; these were either small in line corrections of vocabulary and spelling (in which case we have made changes each time) or were highlighting places in the text relating to the main comments raised above. We have made corrections and changes to address these concerns, and as such we refrain from directly reproducing these comments below to avoid replicating our response above.

Reviewer #2 (Remarks to the Author):

General Comments

My main concern in the first review was about the validation of the model, expressed in my previous comments # 8 and 30. I think the authors have met satisfactorily this concern, as well as

the other ones I had listed.

There are only two additional minor specific comments, listed below.

We thank the reviewer for taking the time to consider our revised study and the useful comments provided.

Specific Comments

1. Lines 314-318: The results of Figures S5 and S6 are consistent with results of Grimm and Tedeschi (2009, reference 5 in this manuscript) for South America, showing that the ENSO impact is stronger and more extensive in the higher precipitation tail of the daily precipitation distribution, meaning that the impact of ENSO is stronger and more extensive on the 95th percentiles than on the monthly precipitation totals. This is visible comparing Figures S5 and S6.

We have added text that these results are consistent with those found by prior work.

2. Lines 436-439: The results of Figures 3 and 4 suggest that instead of mentioning just Colombia and Ecuador, you should mention more generally northern South America.

We have changed 'Colombia and Ecuador' to 'northern South America'

Signed: Alice M. Grimm

REVIEWERS' COMMENTS

Reviewer #1 (Remarks to the Author):

This really is an excellent paper (important, of wide scope and rigorous) definitely worth publishing in Nature Comms and I have really enjoyed reading the revised version. The authors have done a great job of addressing my comments and those of R3 (I was asked to stand in for R3 in the last iteration). In the process they have substantially improved both the content and clarity of an already very strong paper. The remaining comments are now very minor and I'd expect them to be easily achieved by small tweaks to the manuscript.

I have attached: 1) a copy of the manuscript with minor comments and typos, none of these comments require a response, they are all suggestions only; 2) the updated review-response document where I revisit the four major comments from my last review. I have left the author-reviewer interactions from previous iterations and added my comments below each point. Comments I made in previous reviews are in blue, author responses in red and my new comments in black.

Third Review of Emberson et al. 202X, El Nino and Landslides (Nature Comms) **by David Milledge**

Summary

This really is an excellent paper (important, of wide scope and rigorous) definitely worth publishing in Nature Comms and I have really enjoyed reading the revised version. The authors have done a great job of addressing my comments and those of R3 (I was asked to stand in for R3 in the last iteration). In the process they have substantially improved both the content and clarity of an already very strong paper. The remaining comments are now very minor and I'd expect them to be easily achieved by small tweaks to the manuscript. I have attached a copy of the manuscript with minor comments and typos. In this review I revisit the four major comments from my last review.

I have left the author-reviewer interactions from previous iterations and added my comments below each point. Comments I made in previous reviews are in blue, author responses in red and my new comments in black.

Major comments

MC1) Refocussing the paper in the light of the new analysis

In my initial comments I said: "more space needs to be devoted to explaining / understanding what is driving the model response". Though you "now compare several additional variables with MVEI" it is not sufficient to place the results of this analysis in sup info. The description of your results in the response is useful. "broadly, we find that while the areas where significant ($p < 0.05$) relationships are observed are consistent for each of the parameters (total rainfall, extreme rainfall days, number of nowcasts, population exposure), the level of impact (i.e. the slope) of the relationship differs, with MVEI only affecting nowcasts in mountainous areas, and parts of central Asia likely more responsive to increases in number of extreme rainfall events rather than changes in sum total of rainfall." An expanded version of this interpretation should certainly feature in the article. You say that "(t)here is more detail that can be drawn from this analysis, but it is not our intention to focus on the model aspects within this study. We hope that this could provide fruitful questions for future research." I don't think this can wait for future research. This to me is your main finding. I found four comments from my first review relating to this general point that I did not feel were addressed in your response.

"It would also be useful to know whether the steps from correlation between ENSO and extreme precipitation on to landslide hazard then to landslide exposure result in amplification of the correlation or in damping."

As the reviewer has correctly noted, this study is essentially an exploration of how extreme rainfall changes are related to ENSO. Since the other inputs (population, susceptibility) are static, the strength of the relationship between ENSO and the modeled exposure in a given district over a time series is determined fundamentally by the changing rainfall patterns. We appreciate that this may not have been clear enough in this initial or revised text. We have added new figures to the supplementary material and main text to show the difference in the strength of relationship when each additional step of the model is introduced. There are clear spatial patterns in the differences in the strength of relationship between ENSO-total rainfall and ENSO-extreme rainfall that indicate where extreme rainfall has a stronger relationship between ENSO than simply total rainfall, but the spatial patterns are less consistent for the introduction of susceptibility and population, which supports the reviewer's assessment that rainfall is the key determinant of changes.

Critically, the places where exposure and ENSO are linked in most cases already demonstrate significant relationships between ENSO and total rainfall. While extreme rainfall is more strongly correlated with MVEI in some settings, the key locations for exposure (e.g., Colombia, Philippines, Indonesia) do not for the most part exhibit stronger relationships when extreme rainfall, susceptibility or exposure is added. In other words, total rainfall is driving the increases in exposure where there are strong exposure relationships; however, there are other settings where extreme rainfall gives stronger relationships.

This is discussed in a new set of paragraphs, with the new Figure 6 and Supplementary Figures S9 and S10:

*“These results raise the question of why there are such strong relationships between ENSO and exposure in the highlighted locations. We can explore the impact of each part of the model by contrasting the significance of the relationships between total rainfall, extreme rainfall, hazard nowcasts, and exposure and MVEI value. In Figure 3A, we show the p-values for each district for the relationship between total rainfall and MVEI. It is clear that there are already strong relationships between ENSO and total rainfall in locations where we model large changes in exposure due to ENSO (i.e., South East Asia and Central America and northern South America). To determine whether considering only the extreme rainfall leads to stronger relationships, we compare the p-values for total rainfall and extreme rainfall relationships with MVEI; this is shown below in Figure 6. In some geographical areas including parts of Central Asia, Mexico, Iran, China, Luzon, and parts of Thailand, Cambodia and Vietnam, the relationship between MVEI and extreme rainfall is stronger (extreme rainfall p-value is lower than total rainfall) suggesting that the impact of ENSO on extreme rainfall is more significant than on total rainfall in these locations. This is in line with other studies of ENSO-induced rainfall changes^{40,41}, including global studies based on gauge data²⁰ and earlier studies using TRMM and GPCP rainfall data²¹. However, the impact of ENSO on extreme rainfall is weaker than on total rainfall **in critical areas** in Colombia, Indonesia, and the Southern Philippines, all key locations where we model significant relationships between ENSO and landslide exposure. This suggests that the impact of ENSO on total rainfall, rather than extreme rainfall, may be the main driver of changes in exposure in many locations.*

At the same time, ENSO-driven changes in extreme rainfall clearly play a significant role in places like Southern Brazil, Mexico, Eastern China, and parts of Central Asia. Small changes in total rainfall are observed in Central Asia (Figure 1) but large shifts in exposure are modelled in Tajikistan and Kyrgyzstan, likely linked to changes in extreme rainfall (Figure 6); this is in line with other studies of ENSO-induced rainfall changes^{40,41}. Interestingly, these are the areas where El Nino conditions lead to greater modelled landslide exposure (highlighted most clearly in Figure 5), while the areas where total rainfall seems to be more relevant are those where La Nina conditions lead to greater impacts (parts of northern South America and Indonesia, highlighted in Figure 4). This suggests that the two modes of the ENSO system may lead to changes in landslide impact due to differing effects.

We have also compared the strength of relationship for hazard nowcasts and extreme rainfall, and exposure and hazard nowcasts, to test where adding each model input strengthens and weakens the relationship with MVEI (Supplementary Figures S9 and S10, respectively). While there are broader geographical regions where consideration of extreme rainfall increases or decreases the strength of the relationship, adding the susceptibility and population data has smaller and less regionally consistent effects. This suggests that total rainfall, and to a lesser extent extreme rainfall, is most strongly linked to

ENSO, and that the modelled relationship with landslide hazard and exposure is not significantly strengthened or weakened by the addition of susceptibility and population data in the locations we highlight in Figure 4 and 5. Changes in the strength of the relationship between MVEI and exposure and hazard differ most strongly from the relationships for extreme rainfall in the flat, low population areas of the Tibetan plateau and Amazon rainforest.

Figure 6: Ratio of p-values for the relationship between MVEI and total rainfall (see Figure 3A) and MVEI and extreme rainfall (Supplementary Figure S2). Areas with yellow, orange and red colours indicate that extreme rainfall is more strongly correlated with MVEI than total rainfall.

There remain some uncertainties and challenges in this type of analysis. Since LHASA model outputs depend on rainfall exceeding the historical 95th percentile, the model is only sensitive to ENSO-induced changes in the frequency of those extreme events, but not to an increase in intensity of extreme events. Studies have shown ENSO-driven changes in rainfall extreme values in South America 5 and the United States 39; as such, our results may miss some impacts where this is the case. However, other studies have shown that ENSO affects extreme rainfall frequency rather than the peak intensity⁹, suggesting that any effect on extremes may vary region to region. Increases in the intensity of events exceeding the 95th percentile may lead to increased impact from landslides. Our model outputs will provide the same hazard and exposure estimate regardless of the intensity over the 95th percentile, and so if ENSO leads to an increase in peak intensity we may underestimate the impacts on population and infrastructure.

This is all excellent!! My concern is addressed. You might also be able point out that the observation that: *“the modelled relationship with landslide hazard and exposure is not significantly strengthened or weakened by the addition of susceptibility and population data in the locations we highlight in Figure 4 and 5.”* is good news not bad. It suggests that the model is likely fairly insensitive to LHASA parameter choices. So, the results are likely robust and we can have confidence in them. I think you could make that point in this block of text and/or when you discuss LHASA parameter uncertainty and that will allow you to step away from the difficult and time consuming problem of LHASA parameters and sensitivity testing (MC3).

I commented that: *“literature on ENSO effects on extreme rainfall needs to feature more strongly in the paper”*. Though I do see closer connections to the literature in the discussion I don't see this more in depth review early in the paper. This doesn't need to be more than a couple of sentences but it would help to know what is already known about ENSO and extreme precipitation. For example, is this the first study to examine ENSO impacts on frequency of extreme rainfall events? Or to do so at global scale? If not what did others find? This could be done around either L43 or L87.

We thank the reviewer for the comment. We have added a sentence in the introduction to explain that prior studies have explored the relationship between ENSO and extreme rainfall:

"Prior global studies have indicated that ENSO has significant impacts on global patterns of total rainfall, as well as the incidence of extreme rainfall^{20, 21.}"

This is a good addition, I was hoping for one or two more sentences summarising the findings. E.g. does positive ENSO lead to more rain / more frequent intense rain in particular places and less in others? Is it

patchy? If so what sort of length scale is it patchy over? Could you point to particularly affected places (e.g. more frequent intense rainfall over the tropics in El Niño)? Having looked at the references this is very difficult to do. If I were to pull anything out of the two papers you cite it would be:

Curtis et al., 2007: *“Seasonal precipitation over land appears to be much less affected by extreme events than over ocean, especially during El Niño. However, during La Niña the following regions are likely to experience an increased number of extreme daily precipitation events: the Philippines in January–March, the interior Maritime Continent in July–September, and the northwest coast of Australia October–December”*

And Sun et al., 2015:

“The season experiencing the greatest ENSO effect varies regionally, but in most of the ENSO-affected regions is strongest in boreal winter, during which time the 10-year precipitation for [...] a strong [ENSO] episode can be up to 50% higher or lower than for [...] a neutral phase. [The effect of ENSO on] extreme precipitation is asymmetric, with most parts of the world experiencing a significant effect only for a single ENSO phase.”

BUT, if you decide it is too difficult / clumsy to add detail from these studies that is fine by me.

As well as a sentence in the discussion section to indicate that our findings in terms of the total rainfall and extreme rainfall relationships are consistent with these earlier studies:

“This is in line with other studies of ENSO-induced rainfall changes^{40,41}, including global studies based on gauge data²⁰ and earlier studies using TRMM and GPCP rainfall data²¹.”

This is a really good addition and I think it is sufficient for the discussion.

MC2) Modelled landslide exposure units are unclear or difficult to interpret

I said: “The units ‘nowcasts per km² per year’ or ‘person-nowcasts per km² per year’ are difficult to interpret in absolute terms. It would be useful to explain how the reader should interpret these units and how to relate them to hazard in terms of a landslide probability.” You “**have tried to improve this by more clearly laying out the definition**”. However, this is still a problem in the revised draft. It is not simply that the units are not clearly defined but that they need interpreting for the reader.

I still think your best chance of doing this will be to express them differently as I suggested before, suggesting that “ENSO related change in exposure is normalised by the amplitude of the seasonal exposure change” You responded that “**expressing changes in comparison to seasonal changes in exposure ... may lead to misinterpretation. In places with little to no seasonal change.**” This is a good point. However, the problem of unintuitive units remains. In Fig 4 I make the units (days/yr) per unit change in MVEI, but unit MVEI is not an intuitive quantity. I suggest normalising by the range in MVEI, either to express a (days/yr) per-cent change in MVEI or preferably max amplitude in (days/yr) across the full range of observed MVEI (i.e. -2 to 2). I could then read from Fig 4 that ENSO can change exposure per person by up to 10 days per year in the most ENSO sensitive areas, but leads to a change of <3 days per year across the vast majority of the globe. Histograms of these changes would be useful to aid interpretation of how important ENSO is as a driver of landslide exposure.

Thanks to the reviewer for raising this point. The MVEI index is theoretically open ended on either scale, since it depends on observations of key components including sea surface temperature, near surface air temperature, and outgoing long-wave radiation. As such, it isn't strictly speaking feasible to normalize by a maximum value. In addition, since our record doesn't include the intense El-Nino event of 1998-1999, to suggest that we normalize by the full range of observed values may be misleading since it would not include the even larger swings in that period. As such, we feel that the current expression of the units in Figure 4 is as simple as we can make it.

I agree, and I now find the units clear and well described.

This normalisation makes it easier to interpret Fig 4 but doesn't account for the baseline exposure. For example, a change amplitude of 10 days per year (from ElNino to LaNina) might be 100% of the average annual exposure in one location and only 10% in another. I can understand that there are good reasons to report absolute changes but placing the changes in this context also seems important.

This is a really good point; we appreciate the reviewer pushing it again here. We have added a new figure 5 and text to show where the relative changes in exposure are greatest. While the same regions that show large magnitude changes are also seen in the relative changes, some other regions (including Southern Brazil and central Asia) also show up. We hope this helps clarify this point!

New text:

"In Figure 5, we show the relative change in exposure due to a unit change in MVEI relative to the monthly average exposure; the areas where ENSO leads to large magnitude changes also show large relative changes, but there are also parts of Mexico, Southern Brazil, and Central Asia that show large relative increases during El Nino periods."

I really like this new figure and the text around it. This is now resolved for me.

You addressed my comment on Fig 3 that: "(nowcasts per person), I think this should be personnowcasts per km² per year." The new units are explained as "the total number of people exposed to elevated landslide hazard in that district for each month to show seasonality". This is much clearer. However, aggregating by administrative districts is not helpful for readers unless absolute values for those particular districts are the main thing that readers should take away. Different districts are different sizes so the reader loses any intuition for the values at this point. My comment on Fig 4: "Normalising by population makes sense but population was already in the numerator so this could be simply change in nowcasts per km² per year with changing ENSO. You responded that: "Putting the total population in the denominator doesn't quite get back to nowcasts, since it's not always the same population exposed during each nowcast. As can be seen by comparing supplementary figures S7 and S8, it's not identical – there are parts of the US where fractional population exposure changes less than nowcast changes with respect to MVEI, indicating perhaps that more people live in less exposed areas. However, this is relatively speculative, so we refrain from drawing these conclusions in the full text." Thankyou for explaining this. It now makes sense. But it prompts two comments. First, this would be easier to understand if you included the equation for exposure. I think, once normalised by population density it is something like:

$$E_j = \sum_i (R_i S_i P_i) / \sum_i P_i$$

where: E_j is the number of nowcast days per year for the average person in the region j . R_i is the frequency of 95th percentile weighted accumulated rainfall for cell i , S_i is its susceptibility (1 if the cell has susceptibility class > medium, 0 otherwise), P_i is its population. The units are then (days/yr) and are the average exposure to landsliding for an inhabitant of the region. This is a weighted average of H_i where population density is applied as the weighting factor. Second, the text here using the example of parts of the US is very useful it explains what is happening within your model. I think it needs to be included in the main body of the article. It is important that regions with many inhabitants outside the susceptible zones will reduce the apparent average exposure for the region. This has implications for how the data are used. I think that this comment is connected to one of R3's concerns, see my comment on your response to R3 L154.

This is a really helpful comment! There are a couple of points to respond to here, so we'll break it down into a few parts.

1. Normalising by population – we suggest that by showing the relative changes in exposure relative to the longer term average (Figure 5) *and* relative to the total population (Figure 4), there are now two spatially consistent ways to express the changes due to ENSO. Although we have not normalized the pop_exp values in Figure 2, we have now also included population totals for each district to help give context to the values (since expressing as a fraction would ultimately leave the y-axis unchanged).

2. Synthesizing the units more clearly – this equation is a great expression of the units we use. If you don't object, we have used this description in the text to help clarify the data. It works really well!
New text:

"We can express the exposure estimates E_j for each admin district j as follows:

$$E_j = \sum_i [(R_i S_i) P_i] / \sum_i P_i$$

Where R_i is the number of days when rainfall exceeds the historical 95th percentile in cell i , S_i is the susceptibility of that cell (1 if the cell has susceptibility greater than medium, 0 otherwise), and P_i is the population across that cell. This gives a final unit of person-days/year, expressing the average exposure for that district. This can then be normalized by the total population in region j (e.g. Figure 4) or by the long term monthly average exposure (e.g. Figure 5) to allow for intercomparison."

No problem to use the description. The content that you have added addresses my concerns and is now clear. I don't have any more concerns here.

MC3) Some analysis of the sensitivity of the findings to model parameter uncertainty is needed

In my first review I said that detail on two aspects of the modelling was needed. First I requested a clearer explanation of the model structure, assumptions and parameters. Model setup is now clear to me though the model's assumptions could still be clearer.

Second, I argued for "a more robust treatment of the model's structural, input and parameter uncertainty and its impact on your findings" I highlighted this as "the primary limitation to an otherwise

simple and elegant research design.” I suggested that you: either conduct a full uncertainty analysis; or reshape the paper around relationships between ENSO and uncensored frequency of intense storms. You chose to reanalyse the rainfall data, examining intermediate relationships within the model but did not reshape the analysis around these new results. To me this leaves the primary limitation above unaddressed. If you want to retain the focus on landslide exposure rather than landslide triggering rainfall then I think you need a more complete exploration of the sensitivity of your results to parameter uncertainty. I do not agree that it “**would be somewhat circular to conduct uncertainty analysis of the model outputs when the validation data is so limited**” because such data are not required to examine the sensitivity of the findings to parameter uncertainty.

We appreciate the comment here, it is certainly an important point. The parameters discussed by the reviewer below are each related to earlier studies that derived the susceptibility model (fuzzy overlay parameters) and LHASA nowcast model (ARI parameters and thresholds), and in those earlier studies the choice of parameters was analysed. We have clarified in revising our introductory and methods text that these parameters are part of the model design and justified based on prior research.

In particular, I asked: what is the sensitivity of your findings to:

- the choice of 95th percentile in ARI for normalising
- ARI parameters: 7 day window and weighting factor of -2
- The absolute ARI threshold of 6.6 mm
- The parameters of the fuzzy overlay model that defines landslide susceptibility and therefore whether or not rainfall above 95th percentile will trigger a nowcast.
- The choice to nowcast only for areas of moderate to high landslide hazard

I do not understand why/how “**the answer is quite nuanced, and there are different ways to address it**” and I can’t trace these two arguments into the text that follows. I disagree that to test uncertainties requires “**robust observational data to compare the outputs against**”, I don’t think that is needed for sensitivity analysis (as opposed to validation/calibration). Comparison to “**the fatality dataset of Froude and Petley (2018).**” is certainly useful but to me is not related to the sensitivity analysis.

We thank the reviewer for this comment, the logic of our prior answer was not totally sound. This is why we spent some time investigating the possibility of a sensitivity analysis, and as discussed above we feel it isn’t feasible within the current scope of the paper. **As such, we have tried to follow the suggestion of the reviewer to refocus around the findings related to rainfall – ENSO links.** Ultimately, the outputs of the model are representative of the LHASA model parameters; the rainfall connection is based on purely observational data, but the results for hazard depend on model parameter choice. We have added the following text to the discussion to expound on this point:

“In addition, while the observed relationships between total rainfall and frequency of extreme rainfall and ENSO result from analysis of rainfall observations, the connections between hazard and exposure depend on model parameter choices. The parameters used in the LHASA model have been calibrated based on distance to perfect classification of the NASA Global Landslide Catalog38, which gives a false positive rate of 1% and a true positive rate of between 20 and 50% depending on the time interval used for landslide analysis. The exposure and hazard connections to ENSO must therefore be viewed as model outputs rather than direct observations.”

As you say above you have now refocused your analysis around rainfall (I think you've done that very nicely) and this makes LHASA uncertainty a much more minor issue. The text above is useful and worth including. I'd like to see explicit mention that the parameters are uncertain and thus must be calibrated before saying "have been calibrated based on distance... etc". But this is now only a very minor point. I also suggested above, an opportunity within your new text to make the point that **because** the landslide hazard findings are overwhelmingly driven by the pattern of rainfall (except in very obvious and understandable ways e.g. for low gradient landscapes) sensitivity of the LHASA model to uncertain parameters is less of a concern.

New comment RE comparison with observations

You now incorporated the fatality dataset of Froude and Petley (2018) and it is as you say "a useful first order comparison". However, this new analysis needs more attention. I am not convinced that you need this in the paper though I do think it will make the paper even better. If you choose to include it I think you need to deal with three issues.

- 1) It is not clear why you focus on Columbia (in Fig 5) or the other countries discussed in L386-94 but not on others. Explaining your rationale and sampling strategy would probably be enough. It looks as though you perhaps focussed on countries with significant predicted change (and perhaps also with gradient greater than x) and a sufficiently large observation set (e.g. >n recorded landslide events). The quantitative thresholds for inclusion are important here if you are to argue that your choices were objective.
- 2) There is no quantitative evaluation of agreement with model predictions, Fig 5 shows only observations but model predictions are also available so a comparison is feasible. This could take the form of agreement in terms of the sign of the ENSO-exposure relationship, or be pushed further. If you expressed gradient as a function of average exposure you could see whether the gradients agreed.
- 3) It would help a lot if you could find a way to generalise the observations. Otherwise it is difficult to know what to conclude from L386-94 other than that there is good agreement in some places but not in others. That covers a very wide range of model performances.

We appreciate the reviewer taking the time to dig into the new parts of the paper in detail. We have used the comments as impetus to extensively revise this section, and directly compare the model estimates for each MVEI interval with the fatality data. This provides a much more direct and quantitative analysis. We have also explicitly spelled out the reasons for excluding certain countries, and made a more complete discussion of the results in various areas. The new section is replicated below, since it addresses each of the concerns above:

"Comparison with real data

It is important to assess whether the model outputs derived here have any reflection in real data. We are not aware of globally consistent temporal datasets of exposure to landslides that can be directly used to compare with our model outputs. However, fatalities resulting from landslides have been recorded between 2004 and 2016 by Froude and Petley¹⁹ around the world. Froude & Petley consider that their Global Fatal Landslide Dataset (GFLD) likely captures the majority of fatal landslides, with only 15% underestimation⁴⁷. In order to compare this data with our model outputs, we have subdivided the landslides leading to fatalities by country. Given that the GFLD contains fewer than 5,000 landslides

events, there is insufficient data to split them into admin-2 districts as we have for the model output. In each country, we compare the average frequency of fatal landslides for MVEI intervals with the average landslide exposure we model. There are only 26 countries in the GFLD with more than 30 recorded events, so we exclude other countries as we suggest data is too limited to draw conclusions. From these 26, we exclude Bangladesh, Bhutan, Italy, Japan, Nepal, Sri Lanka, Taiwan, and Turkey from further discussion as these are in areas where our model does not suggest a strong ENSO effect. The figures for these excluded countries are still provided in the supplementary material. In the remaining countries, model performance varies. An example for Colombia is shown below in Figure 7; in the top part of the figure, the histograms of modelled exposure and fatalities are shown, and below the relative ratio of the two is shown; a consistent value of 1 would indicate perfect predictive performance. We plot the histogram of MVEI value for the month in which each landslide occurred, and compare it with the histogram of MVEI for each month in the recording period. By comparing the two histograms, we can qualitatively assess which MVEI values have more landslides than we would expect based on the null hypothesis of equal distribution across all MVEI values. An example is shown in Figure 5, for Colombia.

Figure 7. Top: Histogram of average fatal landslide events in Colombia split by MVEI index are shown in red; in blue, a histogram of the average modelled landslide exposure split by MVEI index. In the lower figure, the relative ratio of these two values is shown. A value of 1 indicates that the model perfectly matches the fatality data relative to the maxima of each; lower values indicate that the model overpredicts fatalities.

For most of the analysed countries, the ENSO intervals where the model estimates show the highest average landslide exposure match the ENSO interval where fatal landslides were greatest, but the exposure estimates are also generally less variable than the fatality data. In Figure 8, we show the results for Guatemala, where the fatality data is far more variable than the exposure model estimates. We suggest that with larger numbers of recorded events, the variability in fatality data may show less noise, but this also suggests that the model outputs tend toward the mean exposure to a greater extent than is reflected in reality. The model captures the patterns of landslide impacts in most of the assessed countries, suggesting that our results are a useful first-order estimate of the connection between ENSO and landslide impacts. We calculate the standard deviation of the ratio of the model estimates and fatality data (lower part of Figures 7 and 8, see supplementary material for full table) as a way to quantify how closely the data correspond; the lowest deviations are seen in Indonesia, China, Mexico, and Uganda, with the largest variability (suggesting worse predictive skill) in Thailand, Myanmar, and India (notably countries where the South Asian Monsoon plays an important role). Assessing the specific relationships within individual countries represents an important topic for future research.

Figure 8. Top: Histogram of average fatal landslide events in Guatemala split by MVEI index are shown in red; in blue, a histogram of the average modelled landslide exposure split by MVEI index. In the lower figure, the relative ratio of these two values is shown. A value of 1 indicates that the model perfectly matches the fatality data relative to the maxima of each; lower values indicate that the model overpredicts fatalities.”

This new section addresses almost all my previous concerns and is a really valuable addition to the paper. I have two minor comments.

First, in relation to the countries excluded on the basis that they do not show trend in predictions. This seems a strange reason to exclude if you are seeking to evaluate the model because it is possible that some of these countries show no trend in the model but strong trend in observations and this would be informative. Their inclusion in SI is good though. This is a minor issue and I expect you can address it either: 1) with a statement that these sites are excluded from the paper because they show neither trend in observations nor predictions and would generate spurious results using the performance evaluation metric employed here; or 2) by including them in your analysis.

Second, the information content of the bottom panel of Figs 7 and 8 is pretty small. Given that you have somewhere between 18 and 26 sites in total (depending on your response to the comment above) it seems like you could show all of these curves in one or other of Fig 7 or 8. Having said that, I've looked at the Figures in the supinfo and I don't know if those curves would be very informative (or readable) when plotted together. Perhaps it is best as you have it already. The motivation to include this data comes from L474-5, and particularly the phrase "captures the pattern" I'm not sure what you mean by this and I'd have to go and even if I did I would have to look in the SI to establish whether I agree. Can you come up with a description of what it means to capture the pattern then tell us the fraction of sites that meet that description? That might be enough rather than complicating the Figures, having everything in the SI as you do means people can go and look if they want to see the evidence.

Comments on Authors' response to Reviewer #3.

In my view all of Reviewer #3's original comments have now been adequately addressed

Particular Concerns

The reviewer's third concern is that only a subset of the rainstorms that trigger landslides can be captured within the analysis performed here. This comment may have stemmed from a misunderstanding of the model that the authors use; and their response seeks primarily to clear up this misunderstanding. However, the reviewer's point about clarifying the subset of rainstorms that trigger landslides that can be captured in their method does deserve a more detailed treatment. I think this could be done easily within the introduction or methods by highlighting the range of rainfall durations responsible for the type of rapid catastrophic landslides studied here; then comparing these rainfall durations to the resolution of the rainfall data examined here.

We have added text to the discussion to explain that while it is theoretically possible for short duration events (sub-hourly microburst of rain) to trigger landsliding without exceeding the daily 95th percentile, analysis of intensity duration thresholds suggests that the intensity necessary to do so will generally exceed the 95th percentile of daily rainfall across most of the land areas around the world, suggesting that our model is sensitive to this kind of event. New text:

"In addition, extremely short duration (sub-hourly) rainfall events that exceed local intensity-duration thresholds that do not lead to total daily rainfall exceeding the historical 95th percentile will not be detected. However, the intensity of this kind of rainfall event necessary to trigger landsliding is likely on the order of 10-100mm/hr, which is greater than the daily 95th percentile of historical rainfall across most of the continental surface¹⁵. As such, if these events are captured by the satellite, they will likely lead to a model hazard nowcast"

This new text addresses the comment.

Minor Concerns

This comment had no response. I think that is just a straightforward oversight as it looks easy to address: This should be the South Asian summer monsoon (not India). Tropical cyclone seasonality - again, this is related to TCs in the NWP, the Indian Ocean and the Atlantic, primarily.

Changed 'Indian Summer Monsoon' to 'South Asian Summer Monsoon'

Adequately addressed

L154: This remains a concern (albeit a minor one) for me. R3 said: I can see the logic and strengths of aggregating to admin level, but does this cause problems? Many admin boundaries have no logic in topography or climate. Wont this approach lead to a level of confusion, for example where an admin area covers both a mountainous and a flat area of terrain?

The authors responded that: This is a good point, and we have elaborated upon this further in the text: *"Where admin regions include diverse landscapes (e.g., mountains and flat areas), it may be more difficult to pinpoint key locations for exposure, but we suggest that the admin-2 level offers the best balance of global reproducibility and small scale to allow for focus on local effects."*

I think the authors could make the limitations clearer, R3 explains where this issue arises in their comment.

We have rephrased this section slightly, to help identify the potential issue more clearly:

“Many admin districts are not tied to topography or climate, and where admin regions include diverse landscapes (e.g., mountains and flat areas), it may be more difficult to pinpoint key locations for exposure. If exposure estimates are aggregated in such districts, a signal of a relationship between exposure and ENSO may be masked by noise or conflicting signals from a different part of the area. Smaller admin districts are less likely to merge conflicting signals, and we suggest that the admin-2 level offers the best balance of global reproducibility and small scale to allow for a focus on local effects.”

This new text addresses the comment.

L165: I agree with the authors that this comment stemmed from a misunderstanding and they have now clarified the text. The authors could consider whether expressing their model structure as a flow diagram and/or as a series of equations in SI might further insure against this type of misunderstanding. Having said that I think a careful reading of the text in its current form is sufficient to understand what you have done.

In line with main comment #2 (above), we have added in a key explanatory equation and improvements to the text throughout to help with clarity, which hopefully helps address this point.

Adequately addressed.

Reviewer 1

We thank reviewer 1 for their extensive input throughout this process! We thoroughly appreciate their thoughtful comments, which have at each step improved the final paper. Since the reviewer provided comments both as direct in-line comments for the manuscript as well as in response to the prior 'response to reviewer', we have reproduced both sets of comments below. First, we provide the response to reviewer comments, and then reproduce the in-line comments from the marked up pdf. To distinguish the prior set of comments (revision #2) from the new responses, all prior discussion is now italicized. Where the reviewer concludes that there is no need for further changes, we have not added a response.

Summary

This really is an excellent paper (important, of wide scope and rigorous) definitely worth publishing in Nature Comms and I have really enjoyed reading the revised version. The authors have done a great job of addressing my comments and those of R3 (I was asked to stand in for R3 in the last iteration). In the process they have substantially improved both the content and clarity of an already very strong paper. The remaining comments are now very minor and I'd expect them to be easily achieved by small tweaks to the manuscript. I have attached a copy of the manuscript with minor comments and typos. In this review I revisit the four major comments from my last review.

I have left the author-reviewer interactions from previous iterations and added my comments below each point. Comments I made in previous reviews are in blue, author responses in red and my new comments in black.

Major comments

MC1) Refocussing the paper in the light of the new analysis

In my initial comments I said: "more space needs to be devoted to explaining / understanding what is driving the model response". Though you "now compare several additional variables with MVEI" it is not sufficient to place the results of this analysis in sup info. The description of your results in the response is useful. "broadly, we find that while the areas where significant ($p < 0.05$) relationships are observed are consistent for each of the parameters (total rainfall, extreme rainfall days, number of nowcasts, population exposure), the level of impact (i.e. the slope) of the relationship differs, with MVEI only affecting nowcasts in mountainous areas, and parts of central Asia likely more responsive to increases in number of extreme rainfall events rather than changes in sum total of rainfall." An expanded version of this interpretation should certainly feature in the article. You say that "(t)here is more detail that can be drawn from this analysis, but it is not our intention to focus on the model aspects within this study. We hope that this could provide fruitful questions for future research." I don't think this can wait for future research. This to me is your main finding. I found four comments from my first review relating to this general point that I did not feel were addressed in your response. "It would also be useful to know whether the steps from correlation between ENSO and extreme precipitation on to landslide hazard then to landslide exposure result in amplification of the correlation or in damping."

As the reviewer has correctly noted, this study is essentially an exploration of how extreme rainfall changes are related to ENSO. Since the other inputs (population, susceptibility) are static, the strength of the relationship between ENSO and the modeled exposure in a given district over a time series is determined fundamentally by the changing rainfall patterns. We appreciate that this may not have been clear enough in this initial or revised text. We have added new figures to the supplementary material and main text to show the difference in the strength of relationship when each additional step of the model is introduced. There are clear spatial patterns in the differences in the strength of relationship between ENSO-total rainfall and ENSO-extreme rainfall that indicate where extreme rainfall has a

stronger relationship between ENSO than simply total rainfall, but the spatial patterns are less consistent for the introduction of susceptibility and population, which supports the reviewer's assessment that rainfall is the key determinant of changes.

Critically, the places where exposure and ENSO are linked in most cases already demonstrate significant relationships between ENSO and total rainfall. While extreme rainfall is more strongly correlated with MVEI in some settings, the key locations for exposure (e.g., Colombia, Philippines, Indonesia) do not for the most part exhibit stronger relationships when extreme rainfall, susceptibility or exposure is added. In other words, total rainfall is driving the increases in exposure where there are strong exposure relationships; however, there are other settings where extreme rainfall gives stronger relationships. This is discussed in a new set of paragraphs, with the new Figure 6 and Supplementary Figures S9 and S10:

"These results raise the question of why there are such strong relationships between ENSO and exposure in the highlighted locations. We can explore the impact of each part of the model by contrasting the significance of the relationships between total rainfall, extreme rainfall, hazard nowcasts, and exposure and MVEI value. In Figure 3A, we show the p-values for each district for the relationship between total rainfall and MVEI. It is clear that there are already strong relationships between ENSO and total rainfall in locations where we model large changes in exposure due to ENSO (i.e., South East Asia and Central America and northern South America). To determine whether considering only the extreme rainfall leads to stronger relationships, we compare the p-values for total rainfall and extreme rainfall relationships with MVEI; this is shown below in Figure 6. In some geographical areas including parts of Central Asia, Mexico, Iran, China, Luzon, and parts of Thailand, Cambodia and Vietnam, the relationship between MVEI and extreme rainfall is stronger (extreme rainfall p-value is lower than total rainfall) suggesting that the impact of ENSO on extreme rainfall is more significant than on total rainfall in these locations. This is in line with other studies of ENSO-induced rainfall changes^{40,41}, including global studies based on gauge data²⁰ and earlier studies using TRMM and GPCP rainfall data²¹. However, the impact of ENSO on extreme rainfall is weaker than on total rainfall **in critical areas** in Colombia, Indonesia, and the Southern Philippines, all key locations where we model significant relationships between ENSO and landslide exposure. This suggests that the impact of ENSO on total rainfall, rather than extreme rainfall, may be the main driver of changes in exposure in many locations.

At the same time, ENSO-driven changes in extreme rainfall clearly play a significant role in places like Southern Brazil, Mexico, Eastern China, and parts of Central Asia. Small changes in total rainfall are observed in Central Asia (Figure 1) but large shifts in exposure are modelled in Tajikistan and Kyrgyzstan, likely linked to changes in extreme rainfall (Figure 6); this is in line with other studies of ENSO-induced rainfall changes^{40,41}. Interestingly, these are the areas where El Nino conditions lead to greater modelled landslide exposure (highlighted most clearly in Figure 5), while the areas where total rainfall seems to be more relevant are those where La Nina conditions lead to greater impacts (parts of northern South America and Indonesia, highlighted in Figure 4). This suggests that the two modes of the ENSO system may lead to changes in landslide impact due to differing effects.

We have also compared the strength of relationship for hazard nowcasts and extreme rainfall, and exposure and hazard nowcasts, to test where adding each model input strengthens and weakens the relationship with MVEI (Supplementary Figures S9 and S10, respectively). While there are broader geographical regions where consideration of extreme rainfall increases or decreases the strength of the relationship, adding the susceptibility and population data has smaller and less regionally consistent effects. This suggests that total rainfall, and to a lesser extent extreme rainfall, is most strongly linked to ENSO, and that the modelled relationship with landslide hazard and exposure is not significantly strengthened or weakened by the addition of susceptibility and population data in the locations we highlight in Figure 4 and 5. Changes in the strength of the relationship between MVEI and exposure and hazard differ most strongly from the relationships for extreme rainfall in the flat, low population areas of the Tibetan plateau and Amazon rainforest.

Figure 6: Ratio of p-values for the relationship between MVEI and total rainfall (see Figure 3A) and MVEI and extreme rainfall (Supplementary Figure S2). Areas with yellow, orange and red colours indicate that extreme rainfall is more strongly correlated with MVEI than total rainfall.

There remain some uncertainties and challenges in this type of analysis. Since LHASA model outputs depend on rainfall exceeding the historical 95th percentile, the model is only sensitive to ENSO-induced changes in the frequency of those extreme events, but not to an increase in intensity of extreme events. Studies have shown ENSO-driven changes in rainfall extreme values in South America⁵ and the United States³⁹; as such, our results may miss some impacts where this is the case. However, other studies have shown that ENSO affects extreme rainfall frequency rather than the peak intensity⁹, suggesting that any effect on extremes may vary region to region. Increases in the intensity of events exceeding the 95th percentile may lead to increased impact from landslides. Our model outputs will provide the same hazard and exposure estimate regardless of the intensity over the 95th percentile, and so if ENSO leads to an increase in peak intensity we may underestimate the impacts on population and infrastructure.

This is all excellent!! My concern is addressed. You might also be able point out that the observation that: *“the modelled relationship with landslide hazard and exposure is not significantly strengthened or weakened by the addition of susceptibility and population data in the locations we highlight in Figure 4 and 5.”* is good news not bad. It suggests that the model is likely fairly insensitive to LHASA parameter choices. So, the results are likely robust and we can have confidence in them. I think you could make that point in this block of text and/or when you discuss LHASA parameter uncertainty and that will allow you to step away from the difficult and time consuming problem of LHASA parameters and sensitivity testing (MC3).

- We thank the reviewer for their comment. We have added the following sentence to make this point: *“Our outputs are thus relatively insensitive to LHASA model parameter choices, and supports the robustness of the findings.”* (LL 371-372)

I commented that: “literature on ENSO effects on extreme rainfall needs to feature more strongly in the paper”. Though I do see closer connections to the literature in the discussion I don’t see this more in depth review early in the paper. This doesn’t need to be more than a couple of sentences but it would help to know what is already known about ENSO and extreme precipitation. For example, is this the first study to examine ENSO impacts on frequency of extreme rainfall events? Or to do so at global scale? If not what did others find? This could be done around either L43 or L87.

We thank the reviewer for the comment. We have added a sentence in the introduction to explain that prior studies have explored the relationship between ENSO and extreme rainfall: “Prior global studies have indicated that ENSO has significant impacts on global patterns of total rainfall, as well as the incidence of extreme rainfall20, 21.”

This is a good addition, I was hoping for one or two more sentences summarising the findings. E.g. does positive ENSO lead to more rain / more frequent intense rain in particular places and less in others? Is it patchy? If so what sort of length scale is it patchy over? Could you point to particularly affected places (e.g. more frequent intense rainfall over the tropics in El Niño)? Having looked at the references this is very difficult to do. If I were to pull anything out of the two papers you cite it would be:

Curtis et al., 2007: *“Seasonal precipitation over land appears to be much less affected by extreme events than over ocean, especially during El Niño. However, during La Niña the following regions are likely to experience an increased number of extreme daily precipitation events: the Philippines in January–March, the interior Maritime Continent in July–September, and the northwest coast of Australia October–December”*

And Sun et al., 2015:

“The season experiencing the greatest ENSO effect varies regionally, but in most of the ENSO-affected regions is strongest in boreal winter, during which time the 10-year precipitation for [...] a strong [ENSO] episode can be up to 50% higher or lower than for [...] a neutral phase. [The effect of ENSO on] extreme precipitation is asymmetric, with most parts of the world experiencing a significant effect only for a single ENSO phase.”

BUT, if you decide it is too difficult / clumsy to add detail from these studies that is fine by me.

- We thank the reviewer for their comment. We agree that it is challenging to simply explain changes in total and extreme rainfall trends as a result of ENSO around the world. We prefer to keep the sentence as it is, since it avoids over-complicating the introductory section. Moreover, our own findings demonstrate where ENSO-driven changes lead to shifts in extreme and total rainfall – so the answer to this question can be found in the text below.

As well as a sentence in the discussion section to indicate that our findings in terms of the total rainfall

and extreme rainfall relationships are consistent with these earlier studies:

"This is in line with other studies of ENSO-induced rainfall changes^{40,41}, including global studies based on gauge data²⁰ and earlier studies using TRMM and GPCP rainfall data²¹."

This is a really good addition and I think it is sufficient for the discussion.

MC2) Modelled landslide exposure units are unclear or difficult to interpret

I said: "The units 'nowcasts per km² per year' or 'person-nowcasts per km² per year' are difficult to interpret in absolute terms. It would be useful to explain how the reader should interpret these units and how to relate them to hazard in terms of a landslide probability." You "have tried to improve this by more clearly laying out the definition". However, this is still a problem in the revised draft. It is not simply that the units are not clearly defined but that they need interpreting for the reader.

I still think your best chance of doing this will be to express them differently as I suggested before, suggesting that "ENSO related change in exposure is normalised by the amplitude of the seasonal exposure change" You responded that "expressing changes in comparison to seasonal changes in exposure ... may lead to misinterpretation. In places with little to no seasonal change." This is a good point. However, the problem of unintuitive units remains. In Fig 4 I make the units (days/yr) per unit change in MVEI, but unit MVEI is not an intuitive quantity. I suggest normalising by the range in MVEI, either to express a (days/yr) per-cent change in MVEI or preferably max amplitude in (days/yr) across the full range of observed MVEI (i.e. -2 to 2). I could then read from Fig 4 that ENSO can change exposure per person by up to 10 days per year in the most ENSO sensitive areas, but leads to a change of <3 days per year across the vast majority of the globe. Histograms of these changes would be useful to aid interpretation of how important ENSO is as a driver of landslide exposure.

Thanks to the reviewer for raising this point. The MVEI index is theoretically open ended on either scale, since it depends on observations of key components including sea surface temperature, near surface air temperature, and outgoing long-wave radiation. As such, it isn't strictly speaking feasible to normalize by a maximum value. In addition, since our record doesn't include the intense El-Nino event of 1998/1999, to suggest that we normalize by the full range of observed values may be misleading since it would not include the even larger swings in that period. As such, we feel that the current expression of the units in Figure 4 is as simple as we can make it.

I agree, and I now find the units clear and well described.

This normalisation makes it easier to interpret Fig 4 but doesn't account for the baseline exposure. For example, a change amplitude of 10 days per year (from ElNino to LaNina) might be 100% of the average annual exposure in one location and only 10% in another. I can understand that there are good reasons to report absolute changes but placing the changes in this context also seems important.

This is a really good point; we appreciate the reviewer pushing it again here. We have added a new figure 5 and text to show where the relative changes in exposure are greatest. While the same regions that show large magnitude changes are also seen in the relative changes, some other regions (including Southern Brazil and central Asia) also show up. We hope this helps clarify this point!

New text:

"In Figure 5, we show the relative change in exposure due to a unit change in MVEI relative to the monthly average exposure; the areas where ENSO leads to large magnitude changes also show large relative changes, but there are also parts of Mexico, Southern Brazil, and Central Asia that show large relative increases during El Nino periods."

I really like this new figure and the text around it. This is now resolved for me.

You addressed my comment on Fig 3 that: "(nowcasts per person), I think this should be person-nowcasts per km² per year." The new units are explained as "the total number of people exposed to elevated landslide hazard in that district for each month to show seasonality". This is much clearer. However, aggregating by administrative districts is not helpful for readers unless absolute values for those particular districts are the main thing that readers should take away. Different districts are different sizes so the reader loses any intuition for the values at this point. My comment on Fig 4: "Normalising by population makes sense but population was already in the numerator so this could be simply change in nowcasts per km² per year with changing ENSO. You responded that: "Putting the total population in the denominator doesn't quite get back to nowcasts, since it's not always the same population exposed during each nowcast. As can be seen by comparing supplementary figures S7 and S8, it's not identical – there are parts of the US where fractional population exposure changes less than nowcast changes with respect to MVEI, indicating perhaps that more people live in less exposed areas. However, this is relatively speculative, so we refrain from drawing these conclusions in the full text." Thankyou for

explaining this. It now makes sense. But it prompts two comments. First, this would be easier to understand if you included the equation for exposure. I think, once normalised by population density it is something like:

$$E_j = \sum_{i=1}^n (R_i S_i P_i) / \sum_{i=1}^n P_i$$

where: E_j is the number of nowcast days per year for the average person in the region j . R_i is the frequency of 95th percentile weighted accumulated rainfall for cell i , S_i is its susceptibility (1 if the cell has susceptibility class > medium, 0 otherwise), P_i is its population. The units are then (days/yr) and are the average exposure to landsliding for an inhabitant of the region. This is a weighted average of H_i where population density is applied as the weighting factor. Second, the text here using the example of parts of the US is very useful it explains what is happening within your model. I think it needs to be included in the main body of the article. It is important that regions with many inhabitants outside the susceptible zones will reduce the apparent average exposure for the region. This has implications for how the data are used. I think that this comment is connected to one of R3's concerns, see my comment on your response to R3 L154.

This is a really helpful comment! There are a couple of points to respond to here, so we'll break it down into a few parts.

1. Normalising by population – we suggest that by showing the relative changes in exposure relative to the longer term average (Figure 5) and relative to the total population (Figure 4), there are now two spatially consistent ways to express the changes due to ENSO. Although we have not normalized the pop_exp values in Figure 2, we have now also included population totals for each district to help give context to the values (since expressing as a fraction would ultimately leave the y-axis unchanged).

2. Synthesizing the units more clearly – this equation is a great expression of the units we use. If you don't object, we have used this description in the text to help clarify the data. It works really well!

New text:

"We can express the exposure estimates E_j for each admin district j as follows:

$$E_j = \frac{\sum_i [(R_i S_i) P_i]}{\sum_i P_i}$$

Where R_i is the number of days when rainfall exceeds the historical 95th percentile in cell i , S_i is the susceptibility of that cell (1 if the cell has susceptibility greater than medium, 0 otherwise), and P_i is the population across that cell. This gives a final unit of person-days/year, expressing the average exposure for that district. This can then be normalized by the total population in region j (e.g. Figure 4) or by the long term monthly average exposure (e.g. Figure 5) to allow for intercomparison."

No problem to use the description. The content that you have added addresses my concerns and is now clear. I don't have any more concerns here.

MC3) Some analysis of the sensitivity of the findings to model parameter uncertainty is needed

In my first review I said that detail on two aspects of the modelling was needed. First I requested a clearer explanation of the model structure, assumptions and parameters. Model setup is now clear to me though the model's assumptions could still be clearer.

Second, I argued for "a more robust treatment of the model's structural, input and parameter uncertainty and its impact on your findings" I highlighted this as "the primary limitation to an otherwise simple and elegant research design." I suggested that you: either conduct a full uncertainty analysis; or reshape the paper around relationships between ENSO and uncensored frequency of intense storms. You chose to reanalyse the rainfall data, examining intermediate relationships within the model but did not reshape the analysis around these new results. To me this leaves the primary limitation above unaddressed. If you want to retain the focus on landslide exposure rather than landslide triggering rainfall then I think you need a more complete exploration of the sensitivity of your results to parameter uncertainty. I do not agree that it "would be somewhat circular to conduct uncertainty analysis of the model outputs when the validation data is so limited" because such data are not required to examine the sensitivity of the findings to parameter uncertainty.

We appreciate the comment here, it is certainly an important point. The parameters discussed by the reviewer below are each related to earlier studies that derived the susceptibility model (fuzzy overlay parameters) and LHASA nowcast model (ARI parameters and thresholds), and in those earlier studies the choice of parameters was analysed. We have clarified in revising our introductory and methods text that these parameters are part of the model design and justified based on prior research.

In particular, I asked: what is the sensitivity of your findings to:

- the choice of 95th percentile in ARI for normalising*
- ARI parameters: 7 day window and weighting factor of -2*

- The absolute ARI threshold of 6.6 mm
- The parameters of the fuzzy overlay model that defines landslide susceptibility and therefore whether or not rainfall above 95th percentile will trigger a nowcast.
- The choice to nowcast only for areas of moderate to high landslide hazard

I do not understand why/how “the answer is quite nuanced, and there are different ways to address it” and I can’t trace these two arguments into the text that follows. I disagree that to test uncertainties requires “robust observational data to compare the outputs against”, I don’t think that is needed for sensitivity analysis (as opposed to validation/calibration). Comparison to “the fatality dataset of Froude and Petley (2018).” is certainly useful but to me is not related to the sensitivity analysis.

We thank the reviewer for this comment, the logic of our prior answer was not totally sound. This is why we spent some time investigating the possibility of a sensitivity analysis, and as discussed above we feel it isn’t feasible within the current scope of the paper. **As such, we have tried to follow the suggestion of the reviewer to refocus around the findings related to rainfall – ENSO links.** Ultimately, the outputs of the model are representative of the LHASA model parameters; the rainfall connection is based on purely observational data, but the results for hazard depend on model parameter choice. We have added the following text to the discussion to expound on this point:

“In addition, while the observed relationships between total rainfall and frequency of extreme rainfall and ENSO result from analysis of rainfall observations, the connections between hazard and exposure depend on model parameter choices. The parameters used in the LHASA model have been calibrated based on distance to perfect classification of the NASA Global Landslide Catalog³⁸, which gives a false positive rate of 1% and a true positive rate of between 20 and 50% depending on the time interval used for landslide analysis. The exposure and hazard connections to ENSO must therefore be viewed as model outputs rather than direct observations.”

As you say above you have now refocused your analysis around rainfall (I think you’ve done that very nicely) and this makes LHASA uncertainty a much more minor issue. The text above is useful and worth including. I’d like to see explicit mention that the parameters are uncertain and thus must be calibrated before saying “have been calibrated based on distance... etc”. But this is now only a very minor point. I also suggested above, an opportunity within your new text to make the point that **because** the landslide hazard findings are overwhelmingly driven by the pattern of rainfall (except in very obvious and understandable ways e.g. for low gradient landscapes) sensitivity of the LHASA model to uncertain parameters is less of a concern.

- We thank the reviewer again for their comments here. We feel that we have also addressed this point in our response to MC1; by adding the sentence “Our outputs are thus relatively insensitive to LHASA model parameter choices, and supports the robustness of the findings.” (LL 371-372), We have partly addressed this point.

To address the remainder of this comment, and explain that the LHASA model parameters are uncertain, we have changed the sentence at LL 424-425 to the following:

“The parameters used in the LHASA model are not empirically defined, and so have been calibrated based on distance to perfect classification of the NASA Global Landslide Catalog³⁸”

New comment RE comparison with observations

You now incorporated the fatality dataset of Froude and Petley (2018) and it is as you say “a useful first order comparison”. However, this new analysis needs more attention. I am not convinced that you need this in the paper though I do think it will make the paper even better. If you choose to include it I think you need to deal with three issues.

1) It is not clear why you focus on Columbia (in Fig 5) or the other countries discussed in L386-94 but not on others. Explaining your rationale and sampling strategy would probably be enough. It looks as though you perhaps focussed on countries with significant predicted change (and perhaps also with gradient greater than x) and a sufficiently large observation set (e.g. >n recorded landslide events). The quantitative thresholds for inclusion are important here if you are to argue that your choices were objective.

2) There is no quantitative evaluation of agreement with model predictions, Fig 5 shows only observations but model predictions are also available so a comparison is feasible. This could take the

form of agreement in terms of the sign of the ENSO-exposure relationship, or be pushed further. If you expressed gradient as a function of average exposure you could see whether the gradients agreed.

3) It would help a lot if you could find a way to generalise the observations. Otherwise it is difficult to know what to conclude from L386-94 other than that there is good agreement in some places but not in others. That covers a very wide range of model performances.

We appreciate the reviewer taking the time to dig into the new parts of the paper in detail. We have used the comments as impetus to extensively revise this section, and directly compare the model estimates for each MVEI interval with the fatality data. This provides a much more direct and quantitative analysis. We have also explicitly spelled out the reasons for excluding certain countries, and made a more complete discussion of the results in various areas. The new section is replicated below, since it addresses each of the concerns above:

“Comparison with real data

It is important to assess whether the model outputs derived here have any reflection in real data. We are not aware of globally consistent temporal datasets of exposure to landslides that can be directly used to compare with our model outputs. However, fatalities resulting from landslides have been recorded between 2004 and 2016 by Froude and Petley¹⁹ around the world. Froude & Petley consider that their Global Fatal Landslide Dataset (GFLD) likely captures the majority of fatal landslides, with only 15% underestimation⁴⁷. In order to compare this data with our model outputs, we have subdivided the landslides leading to fatalities by country. Given that the GFLD contains fewer than 5,000 landslide events, there is insufficient data to split them into admin-2 districts as we have for the model output. In each country, we compare the average frequency of fatal landslides for MVEI intervals with the average landslide exposure we model. There are only 26 countries in the GFLD with more than 30 recorded events, so we exclude other countries as we suggest data is too limited to draw conclusions. From these 26, we exclude Bangladesh, Bhutan, Italy, Japan, Nepal, Sri Lanka, Taiwan, and Turkey from further discussion as these are in areas where our model does not suggest a strong ENSO effect. The figures for these excluded countries are still provided in the supplementary material. In the remaining countries, model performance varies. An example for Colombia is shown below in Figure 7; in the top part of the figure, the histograms of modelled exposure and fatalities are shown, and below the relative ratio of the two is shown; a consistent value of 1 would indicate perfect predictive performance. We plot the histogram of MVEI value for the month in which each landslide occurred, and compare it with the histogram of MVEI for each month in the recording period. By comparing the two histograms, we can qualitatively assess which MVEI values have more landslides than we would expect based on the null hypothesis of equal distribution across all MVEI values. An example is shown in Figure 5, for Colombia. Figure 7. Top: Histogram of average fatal landslide events in Colombia split by MVEI index are shown in red; in blue, a histogram of the average modelled landslide exposure split by MVEI index. In the lower figure, the relative ratio of these two values is shown. A value of 1 indicates that the model perfectly matches the fatality data relative to the maxima of each; lower values indicate that the model overpredicts fatalities.

For most of the analysed countries, the ENSO intervals where the model estimates show the highest average landslide exposure match the ENSO interval where fatal landslides were greatest, but the exposure estimates are also generally less variable than the fatality data. In Figure 8, we show the results for Guatemala, where the fatality data is far more variable than the exposure model estimates. We suggest that with larger numbers of recorded events, the variability in fatality data may show less noise, but this also suggests that the model outputs tend toward the mean exposure to a greater extent than is reflected in reality. The model captures the patterns of landslide impacts in most of the assessed countries, suggesting that our results are a useful first-order estimate of the connection between ENSO and landslide impacts. We calculate the standard deviation of the ratio of the model estimates and fatality data (lower part of Figures 7 and 8, see supplementary material for full table) as a way to quantify how closely the data correspond; the lowest deviations are seen in Indonesia, China, Mexico, and Uganda, with the largest variability (suggesting worse predictive skill) in Thailand, Myanmar, and India (notably countries where the South Asian Monsoon plays an important role). Assessing the specific relationships within individual countries represents an important topic for future research.

Figure 8. Top: Histogram of average fatal landslide events in Guatemala split by MVEI index are shown in red; in blue, a histogram of the average modelled landslide exposure split by MVEI index. In the lower figure, the relative ratio of these two values is shown. A value of 1 indicates that the model perfectly matches the fatality data relative to the maxima of each; lower values indicate that the model overpredicts fatalities.”

This new section addresses almost all my previous concerns and is a really valuable addition to the paper. I have two minor comments.

First, in relation to the countries excluded on the basis that they do not show trend in predictions. This

seems a strange reason to exclude if you are seeking to evaluate the model because it is possible that some of these countries show no trend in the model but strong trend in observations and this would be informative. Their inclusion in SI is good though. This is a minor issue and I expect you can address it either: 1) with a statement that these sites are excluded from the paper because they show neither trend in observations nor predictions and would generate spurious results using the performance evaluation metric employed here; or 2) by including them in your analysis.

- The reviewer makes a good point here, this was not explained properly the first time around. We have changed the text accordingly:

From these 26, we exclude Bangladesh, Bhutan, Italy, Japan, Nepal, Sri Lanka, Taiwan, and Turkey from further show limited or negligible trends in observations and predictions, and further analysis would lead to spurious results using the analysis method used here

Second, the information content of the bottom panel of Figs 7 and 8 is pretty small. Given that you have somewhere between 18 and 26 sites in total (depending on your response to the comment above) it seems like you could show all of these curves in one or other of Fig 7 or 8. Having said that, I've looked at the Figures in the supinfo and I don't know if those curves would be very informative (or readable) when plotted together. Perhaps it is best as you have it already. The motivation to include this data comes from L474-5, and particularly the phrase "captures the pattern" I'm not sure what you mean by this and I'd have to go and even if I did I would have to look in the SI to establish whether I agree. Can you come up with a description of what it means to capture the pattern then tell us the fraction of sites that meet that description? That might be enough rather than complicating the Figures, having everything in the SI as you do means people can go and look if they want to see the evidence.

- We thank the reviewer for the comment; we feel that we have covered this point in the text added in the last round of revisions – we explicitly explored the similarity between the patterns by looking at the standard deviations of the ratio of the two values, and discuss it in the text between lines 488 and 492. We are inclined to agree that adding more lines to our figures would clutter them up more than is helpful.

Comments on Authors' response to Reviewer #3.

In my view all of Reviewer #3's original comments have now been adequately addressed

Particular Concerns

The reviewer's third concern is that only a subset of the rainstorms that trigger landslides can be captured within the analysis performed here. This comment may have stemmed from a misunderstanding of the model that the authors use; and their response seeks primarily to clear up this misunderstanding. However, the reviewer's point about clarifying the subset of rainstorms that trigger landslides that can be captured in their method does deserve a more detailed treatment. I think this could be done easily within the introduction or methods by highlighting the range of rainfall durations responsible for the type of rapid catastrophic landslides studied here; then comparing these rainfall durations to the resolution of the rainfall data examined here.

We have added text to the discussion to explain that while it is theoretically possible for short duration events (sub-hourly microburst of rain) to trigger landsliding without exceeding the daily 95th percentile, analysis of intensity duration thresholds suggests that the intensity necessary to do so will generally exceed the 95th percentile of daily rainfall across most of the land areas around the world, suggesting that our model is sensitive to this kind of event. New text:

"In addition, extremely short duration (sub-hourly) rainfall events that exceed local intensity-duration thresholds that do not lead to total daily rainfall exceeding the historical 95th percentile will not be detected. However, the intensity of this kind of rainfall event necessary to trigger landsliding is likely on the order of 10-100mm/hr4, which is greater than the daily 95th percentile of historical rainfall across

most of the continental surface¹⁵. As such, if these events are captured by the satellite, they will likely lead to a model hazard nowcast"

This new text addresses the comment.

Minor Concerns

This comment had no response. I think that is just a straightforward oversight as it looks easy to address: This should be the South Asian summer monsoon (not India). Tropical cyclone seasonality - again, this is related to TCs in the NWP, the Indian Ocean and the Atlantic, primarily.

Changed 'Indian Summer Monsoon' to 'South Asian Summer Monsoon'

Adequately addressed

L154: This remains a concern (albeit a minor one) for me. R3 said: I can see the logic and strengths of aggregating to admin level, but does this cause problems? Many admin boundaries have no logic in topography or climate. Wont this approach lead to a level of confusion, for example where an admin area covers both a mountainous and a flat area of terrain?

The authors responded that: This is a good point, and we have elaborated upon this further in the text:

"Where admin regions include diverse landscapes (e.g., mountains and flat areas), it may be more difficult to pinpoint key locations for exposure, but we suggest that the admin-2 level offers the best balance of global reproducibility and small scale to allow for focus on local effects."

I think the authors could make the limitations clearer, R3 explains where this issue arises in their comment.

We have rephrased this section slightly, to help identify the potential issue more clearly:

"Many admin districts are not tied to topography or climate, and where admin regions include diverse landscapes (e.g., mountains and flat areas), it may be more difficult to pinpoint key locations for exposure. If exposure estimates are aggregated in such districts, a signal of a relationship between exposure and ENSO may be masked by noise or conflicting signals from a different part of the area. Smaller admin districts are less likely to merge conflicting signals, and we suggest that the admin-2 level offers the best balance of global reproducibility and small scale to allow for a focus on local effects."

This new text addresses the comment.

L165: I agree with the authors that this comment stemmed from a misunderstanding and they have now clarified the text. The authors could consider whether expressing their model structure as a flow diagram and/or as a series of equations in SI might further insure against this type of misunderstanding. Having said that I think a careful reading of the text in its current form is sufficient to understand what you have done.

In line with main comment #2 (above), we have added in a key explanatory equation and improvements to the text throughout to help with clarity, which hopefully helps address this point.

Adequately addressed.

Line comments:

L19-20: Not quite clear without reading the paper. Suggest: "than that due to seasonal rainfall variability". Trying to avoid ambiguity around what it is that is changing.

- Changed as suggested.

Line 23: not needed, "debris flows"

- Changed as suggested.

Line 46: particularly in developing

- Changed as suggested.

LL47-48: I suggest: "difficult to collect"

- Changed as suggested.

L60: suggest: "their cycles have long timescales that are not captured..."

- Changed as suggested.

LL81-82: suggest: changes over timescales on the order

- Changed as suggested.

L131: suggest: "are unchanged" if that is true.

- Changed as suggested.

L204: Perhaps here you could say:

"For each month, we can express the..." It would be useful somewhere here to indicate the time window over which E_j is being calculated.

Equation 1: I'm not very familiar with this sigma notation but I think it should be $\sigma(\text{subscript } i)$ rather than $\sigma(\text{subscript } j)$.

I would have gone for sigma with $i=1$ on the bottom and n on the top. Then I'd define n as the number of cells within the admin district.

Your notation might well be right, I'm just checking.

- Changed as suggested.

And: i should be subscript.

- Changed as suggested.

L 205: It might be worth adding either:

number of days in a given month or number of days in time period t then adding t as a subscript I think that would be:

$$E_{jt} = \sum(R_{it}, S_i, P_i)_j$$

- Added explanation that this is summed over an annual timespan

L240: An extremely minor point but if there is no other reason for the order it would be more intuitive to have Planadas, Kinnaur, Chenzhou from left to right to match their locations on the map.

This is only a suggestion not a requirement.

- Changed as suggested to flip the two around.

L253: suggest: "total monthly rainfall" if that is what this is.

- Changed as suggested.

L 257: Is this different from the total rainfall in 3A? If so, I'm not following the distinction here. If not then I think you can remove it from this list.

- See correction to previous comment; these two now agree.

L 280: A few more words would help here. Are we looking at the middle figures (Kinnaur) as an example where the slope is steep despite a weaker relationship or where the slope is gentler. It looks as though the left panel (Chengzhou) has a steeper gradient.

I'm looking for something like: "middle set of figures in Fig 2, where a unit increase in MVEI results in a reduction of 100,000 person days per month."

Again, a suggestion not requirement. Alternatively you might choose to simply cut this final clause. I'm not sure you need it given what follows.

- Changed to suggested sentence.

L 298: Suggest: "gradient" rather than magnitude.

- Changed as suggested.

L 304: Check Fig vs Figure for consistency

- Made sure all occurrences now read 'Figure'

L 307: Two full stops

- Fixed

L 321: suggest: "monthly average exposure"

- Changed as suggested.

LL325-326: suggest: "between MVEI value and each of: total rainfall, extreme rainfall, hazard nowcasts and exposure."

The double and at the end of the sentence might make it difficult to see that it is each of the first list items compared to MVEI.

- Changed as suggested.

L 387: The terminology here seems strange, though it is clear. Perhaps changing the first mention to United States of America would be enough. Alternatively you could cut the hyphenated clause and start the next sentence in particular.

- First entry says United States of America; remaining use shorter USA.

L392: Check for consistency: United States, USA, US.

- Fixed

L 393: This seems very broad, could you say:

"resolving these rainfall effects" i.e. of the type in the previous sentence. That would make things more focused.

However, it might be that you can't make this more specific (by adding these) and if so that's fine.

- Changed as suggested.

L 445: check consistency: F and P, F & P.

- Addressed; now read the same

L 460: I think labelling the top panel a and the bottom b would be helpful because of the way that you refer to them in the text.

- Changed as suggested.

L 460: Have these histograms been normalised so that their maximum frequency is equal to 1 or is that a coincidence? If you normalise I think you need to mention that above.

- Added text to explain that these are normalized.

L 462: suggest: "A y-axis value of 1 in panel b" or "A y-axis value of 1 in the lower panel"

- Changed to first suggestion

L 466: consider adding a percentage: e.g. "For most (55%) of" or "For 10 of the 18 analysed"

- Changed as suggested.

L 466: 'where'

- Changed as suggested.

LL 474-475: What do you mean by captures the pattern and by most? Again here you could add a percentage in brackets to evidence most.

- Rephrased to better explain that the patterns match one another.